



# Variability of the Brunt-Väisälä frequency at the OH-airglow layer height at low and mid latitudes

Sabine Wüst [1], Michael Bittner [1,2], Jeng-Hwa Yee [3] , Martin G. Mlynczak [4], James M. Russell III [5]

[1] Deutsches Fernerkundungsdatenzentrum, Deutsches Zentrum für Luft- und Raumfahrt, 82234 Oberpfaffenhofen, Germany

[2] Institut für Physik, Universität Augsburg, 86159 Augsburg, Germany

[3] Applied Physics Laboratory, The Johns Hopkins University, Laurel, Maryland, USA

[4] NASA Langley Research Center, Hampton, USA

[5] Center for Atmospheric Sciences, Hampton, USA

*Correspondence to*: Sabine Wüst (sabine.wuest@dlr.de)

**Abstract.** Airglow spectrometers as they are operated within the Network for the Detection of Mesospheric Change (NDMC, https://ndmc.dlr.de), for example, allow the derivation of rotational temperatures which are equivalent to the kinetic temperature, local thermodynamic equilibrium provided. Temperature variations at the height of the airglow layer are amongst others caused by gravity waves. However, airglow spectrometers do not deliver vertically-resolved temperature

information. This is an obstacle for the calculation of the density of gravity wave potential energy from these measurements. As Wüst et al. (2016) showed, the density of wave potential energy can be estimated from data of OH* airglow spectrometers if co-located TIMED-SABER (Thermosphere Ionosphere Mesosphere Energetics Dynamics, Sounding of the Atmosphere using Broadband Emission Radiometry) measurements are available since they allow the calculation of the Brunt-Väisälä frequency. If co-located measurements are not available, a climatology of the Brunt-Väisälä frequency is an

alternative. Based on 17 years of TIMED-SABER temperature data (2002–2018) such a climatology is provided here for the OH* airglow layer height and for a latitudinal longitudinal grid of 10° × 20° at mid and low latitudes. Additionally, climatologies of height and thickness of the OH* airglow layer are calculated.

**Key words:** Airglow, Hydroxyl, Brunt-Väisälä frequency, climatology, TIMED-SABER



# 1 Introduction

This is the succeeding publication to Wüst et al. (2017a) where the angular Brunt-Väisälä frequency (BV frequency) was calculated for the OH*-layer height between 43.93–48.09°N and 5.71–12.95°E using TIMED-SABER data from 2002 to 2015. The choice of the geographical region, which includes the Alps, was due to the location of the five NDMC-stations

Oberpfaffenhofen (48.09°N, 11.28°E), the observatory Hohenpeißenberg (47.8°N, 11.0°E), the Environmental Research Station Schneefernerhaus (47.42°N, 10.98°E), Germany, and the observatories Haute Provence (43.93°N, 5.71°E), France, and Sonnblick (47.05°N, 12.95°E), Austria. We described seasonal variations of the three parameters, height and full width at half maximum (FWHM) of the OH*-layer as well as the BV frequency weighted according to the parameters of the OH*-layer, and provided a climatology of the yearly course of this BV frequency.

Now, the data basis is extended to global TIMED-SABER (Thermosphere Ionosphere Mesosphere Energetics Dynamics, Sounding of the Atmosphere using Broadband Emission Radiometry) measurements. Three more years (2016–2018) are included in the analysis which changed slightly compared to Wüst et al. (2017a): instead of calculating the Gaussian-weighted BV frequency, the BV frequency is now weighted with the volume emission rate of the OH-B channel of SABER. Furthermore, the geographical position of the SABER measurements at 86 km height is taken into account (in our preceding

publication any part of the SABER profile needed to fit the geographical selection criteria).

The angular BV frequency $N$, which is for example needed for calculation of the density of gravity wave potential energy, varies with the temperature $T$ and its vertical gradient (e.g. Andrews, 2000):

$$N\left(T, \frac{dT}{dz}\right) = \sqrt{\frac{g}{T}\left(\frac{dT}{dz} - \Gamma_d\right)} \qquad (1)$$

where $\Gamma_d$ is the dry-adiabatic lapse rate defined as the vertical adiabatic temperature decrease. In most cases its value is given as 9.8 K/km. However, the acceleration due to gravity, g, is slightly height-dependent and determines together with the specific heat capacity at constant pressure, $c_p$, the dry-adiabatic lapse rate. g reaches a value of ca. 9.55 m/s² at 86 km height, therefore, the vertical adiabatic temperature decrease is ca. 9.5 K/km there. g also depends on the geographical position due to the fact that the Earth is not a perfect sphere but oblate. Since the variation in the Earth radius is less than 86 km (only

circa one quarter of it), this effect is of minor importance and therefore neglected here.

Measurement techniques which provide vertical temperature profiles allow therefore the direct calculation of the BV frequency and of further parameters such as the density of wave potential energy (see e.g. Kramer et al., 2015; Mzé et al., 2014; Rauthe et al., 2008 to mention just a few). OH* spectrometers, however, deliver information about temperature always vertically averaged over the OH*-layer. OH* imaging systems provide in most cases only brightness maps (e.g., Sedlak et al.

(2016) and Hannawald et al. (2016, 2019) who addressed a small part of the sky, Garcia et al. (1997) who operated an all-sky system). An exception is here Pautet et al. (2014) who worked with narrow-band filters and could derive temperature maps. At a worse horizontal resolution also scanning OH* spectrometers can deliver horizontally-resolved temperature information





(see e.g. Wachter et al., 2015; Wüst et al. 2018). However, also in these cases the temperature is vertically averaged over the OH\*-layer; the BV frequency cannot be calculated.

So, one needs to rely on temperature climatologies or on complementary temperature measurements. The latter should be of higher accuracy in most cases, if the coincidence in time and space of the complementary and the original measurement is

high enough (see Wendt et al., 2013 for the quantification of typical temperature differences due to mistime and misdistance). Since complementary measurements are rare and not available at every NDMC station and at every time, a climatology of the BV frequency based on global satellite-based measurements is very valuable. Ca. 85% of all spectrometers and photometers listed in the data basis of NDMC address at least one of the various OH emissions (Schmidt et al., 2018), thus TIMED-SABER OH-B channel and temperature measurements are used for the BV frequency

climatology. The OH-B channel covers the wavelength range from 1.56 to 1.72 μm, which includes mostly the OH (4-2) and OH (5-3) vibrational transition bands. The peak altitude difference for adjacent vibrational levels is ca. 500 m (e.g. Adler-Golden, 1997 and von Savigny et al., 2012) and therefore negligible compared to the FWHM. Of course, a climatology based on global satellite measurements always provides averaged information. Effects of processes which vary during one night, such as gravity waves, or which change significantly from year to year are not or only to a small extent included (due

to the thickness of the OH\*-layer, at least small-scale variations cancel out, see e.g. Wüst et al., 2016).



## 2 Data and analysis

We use TIMED-SABER temperature and OH-B channel data (volume emission rates, VER) in its latest version (2.0) for the years 2002 to 2018. It was downloaded from the SABER homepage (saber.gats-inc.com). In order not to duplicate
information, the reader is referred to our preceding publication Wüst et al. (2017a) and publications therein for more details about TIMED-SABER (Mertens et al., 2004; Mlynczak, 1997; Russell et al., 1999), the retrieval of kinetic temperatures (Dawkins et al., 2018; Garcia-Comas et al., 2008; Lopez-Puertas et al., 2004; Mertens et al., 2004 and 2008; Remsberg et al., 2008) and a comparison between SABER v2.0 temperature and ground-based lidar data (Dawkins et al., 2018). Since the OH*-spectrometers allow only measurements during night, we calculate the exact local time (by adding four minutes to UTC
for every longitudinal degree) and require SABER measurements between 6 p.m. and 6 a.m. (local time).

The squared BV frequency $N^2{}_i$ is computed for each SABER profile at every available height and weighted with the OH VER. The result is called the OH*-equivalent BV frequency in the following. From time to time, a maximum in the VER is observed around 40 km height and the respective profile shows strong oscillations. These profiles are excluded from further analysis steps.

Furthermore, information about the OH* layer is derived from the OH-VER profiles. The centroid height is denoted as the OH* height in the following and the FWHM is calculated by determining the maximum of the OH VER and subtracting the lower height from the upper height where half of the maximum is reached for the first time (starting at the height of maximal OH VER).

In the following, the OH* height, the FWHM and the OH*-equivalent BV frequency are mapped to a
20 20° (longitude) x 10° (latitude) grid. The data are ascribed to the mid points of the respective intervals. Here, the geographical position of the SABER measurement at 86 km height is taken into account. Then, the daily mean of each parameter is calculated for every grid cell. This is done for all years. It is assumed that every year is a leap year, this facilitates further calculations.

Due to the yaw cycle of SABER, data gaps are visible at higher latitudes and all investigations in this publication include the
25 latitudinal range of 60°S to 60°N.



# 3 Results and discussion

All three parameters, the OH\*-layer height, the FWHM, and the OH\*-equivalent BV frequency, vary with latitude,
longitude, and day of year (DoY). In this section, the results will be first described qualitatively based on some examples.
Then, the annual development of the OH\*-equivalent BV frequency will be approximated and the respective mathematical
function will be provided.

Since the BV frequency depends on temperature, which changes strongly with DoY and latitude, one would expect that the
BV frequency varies more with these two parameters than with longitude. Both, the latitudinal and the temporal dependence
of the temperature are strongly determined by the residual circulation (see e.g. Garcia and Solomon (1985) who give in their
introduction a concise overview about the development of our knowledge concerning the mean meridional circulation). The
residual circulation consists of horizontal and vertical movements. The higher the latitude the more important the vertical
movement and the less important the horizontal one becomes. The vertical movement influences the temperature through
adiabatic warming or cooling but also the downward transport of atomic oxygen (the dominating species for the formation of
OH\*) from heights above the OH\*-layer and therefore the OH\*-height and thickness (e.g. Shepherd et al., 2006): a
downward movement leads to a lower and brighter OH\*-layer and vice versa. On average, the OH\*-layer is thicker (thinner)
during a prevailing downward (upward) movement (e.g. Liu and Shepherd, 2006). Therefore, it is not surprising, that an
annual cycle is clearly visible in the temporal development of all three parameters at mid latitudes. It dominates the
development of the OH\* height and the OH\*-equivalent BV frequency during the year at all longitudes for 45° N (figure 1a),
for example. The FWHM additionally shows a period of ca. 60 d in every season but summer. At low latitudes the annual
cycle is less pronounced. At all longitudes for 5°N, for example, a semi-annual cycle and superimposed oscillations with
smaller periods of ca. 60 d (especially for the FWHM and the OH\*-equivalent BV frequency) gain importance for the
development of the three parameters during the year or even dominate it (figure 1b).

The annual development of the OH\*-layer height, the FWHM, and the OH\*-equivalent BV frequency varies to some extent
also with longitude. For the different longitudes the yearly latitudinal means over the three parameters range between ca.
85 km and 87 km, 7.0 km and 8.25 km, and 0.020 1/s and 0.023 1/s (figure 2). The longitudinal variability (peak to peak
difference) is at maximum ca. 1.5 km (ca. 2%) for the OH\*-layer height, ca. 1 km (ca. 13%) for the FWHM, and ca.
0.001 1/s (ca. 5%) for the OH\*-equivalent BV frequency (figure 2). The graphs referring to the different longitudes spread
more for 5°N than for 45°N (figure 1).

Therefore, the approximation of the annual development of the OH\*-equivalent BV frequency will be calculated on a
latitude longitude grid, which is $10° \times 20°$. The number of values per grid cell and year varies strongly. For high latitudes
(65–85 °N or °S), the OH\*-equivalent BV frequency can be provided for less than half a year due to the TIMED yaw cycle.
Thus, these latitudes are excluded from further investigations. For mid and low latitudes, data gaps exist only for individual





days. For the climatology of the OH*-equivalent BV frequency, which is based on 17 years of TIMED-SABER data, ca. 80–190 values are available per grid cell and day at maximum. The number of values and the variation in the number of values per grid cell over the year is higher for mid latitudes compared to low ones. The average number of values per grid cell and day ranges between ca. 45 (for low latitudes) and 85 (for mid latitudes).

In the following, we discuss the possible origin of the oscillations described above. Here, we have to discriminate between natural phenomena and possible artefacts due to the yaw cycle of TIMED in order to chose a correct mathematical approximation of the annual development of the BV frequency.

The overpass time of TIMED-SABER varies with DoY (figure 3): TIMED flies by a little bit earlier every day with respect

to a fixed geographical position and has a yaw cycle of 60 d, i.e., the viewing direction of the instrument changes every 60 d, and the overpass time at a specific geographical position is the same every 120 d for the same viewing direction (ascending or descending part of the orbit). If the observed parameter has a fixed daily cycle, an artificial 120 d oscillation can be generated in the respective time series. Zhang et al. (2006) showed such a periodicity in SABER temperature measurements at 86 km height in their figure 2a. If the viewing direction is neglected, the overpass time is identical every 60 d. In this case,

an artificial 60 d oscillation could be generated in the respective time series.

The BV frequency does not only depend on temperature but also on the vertical temperature gradient. Changes in temperature and its vertical gradient can affect the development of the BV frequency during the night but they do not necessarily need to (if both, temperature and vertical temperature gradient, increase (or decrease) simultaneously, their effect on the BV frequency can also cancel out, see equation 1). Approximating linearly the temporal dependence of

temperature and its vertical gradient during night for 5°N gives -0.94 K/h and -0.25 K/km/h (figure 4a and b, $R^2$ is 6% and 17%). For 45°N, the parameters increase on average by 0.81 K/h and 0.04 K/km/h during the night (figure 4c and d, $R^2$ is 3% and 0.4%). Assuming a linear behaviour of both, the temperature and its vertical gradient, an effect on the BV frequency cannot be derived for 45°N. For 5°N, the BV frequency shows a temporal dependence (figure 4e), which is the condition for a sensitivity to the yaw cycle. Even though the respective $R^2$ values are very low, the result is consistent with our

observations (figure 1c and f).

What leads to this temporal dependence of temperature and its vertical gradient? When comparing the dynamics of low and mid latitudes, dominating large-scale motions are a prominent difference. Offermann et al. (2009) showed in their figure 8 that tidal motions gain more and more importance in comparison to planetary waves at ca. 90 km height the lower the latitude is. At 20°N, tidal waves cause more variability than stationary planetary waves. Compared to travelling planetary

waves, tidal variations lie in the same range (Offermann et al., 2009). We can conclude that at low latitudes the effect of tidal motions on temperature is at least in the same order of magnitude as the influence of planetary waves. Concerning the temperature gradient we can argue as follows: tides are propagating faster in the vertical than planetary waves since their periods differ more than their vertical wavelengths (the mean vertical wavelength of the 5 d Rossby wave, for example, is ca. 50–60 km according to Pancheva et al. (2010) who investigated TIMED-SABER temperature measurements for six full





years from January 2002 to December 2007, while the vertical wavelengths of tides are mostly in the same range or slightly smaller and reach ca. 30 km at minimum as for example Zhang et al. (2006) showed for case studies and Forbes et al. (2008) who used TIMED-SABER temperature measurements from March 2002 to December 2006). That means they influence the temperature gradient stronger during one night than planetary waves. Since tides are more active at low latitudes compared

to mid latitudes, the temperature gradient should change more during one night at low latitudes. This agress qualitatively with the results shown in figure 4.

As shown by Silber et al. (2017) in their figure 9 for four years of GRIPS data (Ground based Infrared P-branch Spectrometer) at Tel Aviv (32.1°N, 34.8°E), the phase of the diurnal and the semi-diurnal tide is relatively stable (the tides are approximated by a cosine starting at 12 UT, the phases approach values near zero on average). So, tides can in principle

lead to an oscillation in the BV frequency.

However, for our analysis we use only nightime SABER data and do not distinguish between ascending and descending orbits. That means the observation time jumps for example around DoY 175 in figure 3 for 5°N from 6 p.m. to 6 a.m. If the BV shows a diurnal cycle due to the diurnal tide, a jump should be observed here, too. Since these discontinuities in observation time appear regularly, a kind of saw tooth pattern should be very prominent in figure 1. This is not the case even

though the existence of such a pattern can be disguised to some extent since the graphs in figure 1 are smoothed and since for selected time intervals data which are seperated by a few hours are averaged. The non-smoothed data exhibit indeed a slight saw tooth pattern (not shown here); in any case, we can therefore conclude that a semi-diurnal cycle must be prominently present in the data. This agrees with Silber et al. (2017) who showed in their figure 7 that the amplitudes of the semi-diurnal and the diurnal tide are comparable.

In the same figure, those authors exhibit that tidal amplitudes in general are relatively low during summer. This is captured by the strength of the 60 d oscillation in the OH*-equivalent BV frequency time series.

These effects can probably also explain the oscillation of ca. 60 d in the FWHM at 45°N during parts of the year. Compared to the calculation of the OH* layer height and of the BV frequency, the FWHM is not weighted by the VER (see section 2) and therefore it is more sensitive to individual variations of the OH VER profile. Only during selected time intervals (e.g.,

ca. DoY 90–120, 220–250 und 330–365, see figure 3), profiles sensed at different nighttimes (time difference ca. 4 h) are available. Comparing figure 3 and 1b, one can see for example around DoY 30–40 that the gradient of the FWHM changes its sign when the observation time jumps in this case from approximately 6 p.m. to 6 a.m.

However, oscillations with slightly shorter periods than 60 d (ca. 50 d) are also observed in measurements which are not affected by a 60 d yaw cycle as for example Rüfenacht et al. (2016) showed based on horizontal wind values derived from a

ground-based Doppler wind radiometer. Those measurements refer to the altitude range between the mid stratosphere (5 hPa) and the upper mesosphere (0.02 hPa); low, middle and high latitudes are adressed in the publication. The observed periods between 20 d and 50 d are subject to temporal variations. The reason for these oscillations is not clear. The authors discuss a link to solar forcing, however, they point out that solar forcing might influence the atmospheric wave pattern only





in an indirect way. Therefore, it is possible, that we see in our data a mixture between natural and artificial effects. However, we cannot distinguish between them.

A mentioned above, visual inspection of figure 1 depicts at least one additional oscillation with a semi-annual period besides the annual cycle and the 60 d variation. Semi- and even ter-annual periods are observed by other authors and in different parameters (for the semi-annual cycle in different parameters in mesosphere and higher but not specifically for airglow see, e.g., the introduction of Silber et al. (2016)). Based on WINDII (Wind Imaging Interferometer) measurements between 60°N and 60°S from 1991 to 1997, Shepherd et al (2006) showed in their figure 1 a semi-annual variation in the OH emission rate between ca. 20°N and 20°S with maxima in spring and autumn. The authors attribute this oscillation to the semi-annual variation of the downward mixing associated with the variation in amplitude of the diurnal tide. Liu and Shepherd (2006) pointed out that the column integrated emission rate is inversely related to the peak height for WINDII data between 40°S and 40°N and developed an empirical model for predicting the altitude of the peak of the OH nightglow emission. Here, they included amongst others sinusoidal annual and semi-annual variations. Mulligan et al. (2009) transferred this model to Longyearbyen (78°N, 16°E) and found amplitudes unequal to zero for the annual and semi-annual mode. For low latitudes, von Savigny (2015) showed minima in the OH* height in spring and autumn. Annual, semi- and ter-annual oscillations are also observed in temperature: as published by Höppner and Bittner (2007) in their figure 2 for Wuppertal, Germany (51°N, 7°E) 1993, the OH* temperature derived by the ground-based spectrometer GRIPS does not only show an annual cycle, it is charaterized by kind of plateaus at DoY 40–80 (February and March) and DoY 260–300 (September and October). Therefore, the authors approximate the overall yearly course a (quasi) annual, (quasi) semi-annual and (quasi) ter-annual sinusoidal (see also Bittner et al., 2000, 2002). Those plateaus can also be observed using GRIPS data at other stations, e.g. at Tel Aviv (see Wüst et al., 2017b) and if the temperature data are averaged over some years (tested for the Environmental Research Station Schneefernerhaus, UFS, and not shown here).

In order to provide qualitative results, the harmonic analysis (one or single step mode, see e.g. Wüst and Bittner, 2006) is applied to the time series of OH*-equivalent BV frequency averaged over each DoY and separated according to latitude and longitude. In order to avoid the approximation of the 60 d oscillation, which might be artificial as discussed above, the period range which the harmonic analysis uses is restricted to 180–366 d. The number of oscillations is chosen to be two. The results are summerized in table 1 and the quality of approximation is plotted versus latitude in figure 5. For many latitudes, longitudinal differences are clearly visible in the quality of approximation. However, when taking a +/- 10 % interval around the approximation all data are covered in nearly all cases (figure 6). As already shown above using the two latitudes of 5°N and 45°N, the importance of the annual and semi-annual mode (and therefore also the quality of approximation as well as the amplitudes of the different modes) decreases with decreasing latitude or inversly formulated the importance of the 60 d oscillation, which is not adapted, increases with decreasing latitude. The quality of approximation over all longitudes reaches its minimum near the equator (figure 5). Remarkable is the asymmetry in the quality of





approximation between the northern and the southern hemisphere. This agrees quite well with asymetries in tidal activity observed, for example, by Vincent et al. (1989). Those authors investigated radar wind observations which refer to 80–100 km height at Adelaide (35°S, 138°E) and Kyoto (35°N, 136°E), two places which are symmetrically located around the equator. The diurnal tidal winds at Adelaide have a larger amplitude than at Kyoto (factor 2–3). However, the reason for this

behaviour is not entirely clarified. There exist hemispheric differences in tidal forcing, but also differences in the middle atmosphere winds through which the tides must propagate, and finally differences in dissipation. Concerning the amplitudes of the semi-diurnal tide, the authors found out that they are in general smaller at both sites than the amplitudes of the diurnal tide. During local summer the amplitudes are larger at Adelaide.

As mentioned at the beginning, this is the succeeding manuscript to Wüst et al. (2017a) where the authors proposed an approximation of the OH*-equivalent BV frequency for the Alpine region more exactly for 43.93–48.09° N and 5.71–12.95° E based on three oscillations. The question, which naturally arises now, is how the two different approximations, the one of Wüst et al. (2017a) and one proposed here, compare. The following differences in methods and data basis exist: the pixel size of Wüst et al. (2017a) is ca. 5°×7° and therefore less than 50% of the pixel size applied here. Wüst et al. (2017a)

use three oscillations and SABER data from 2002–2015, we use two oscillations and SABER data from 2002–2018. Furthermore, instead of calculating the Gaussian-weighted BV frequency, the OH*-equivalent BV frequency is now weighted with the volume emission rate of the OH-B channel of SABER. Then, the geographical position of the SABER measurements at 86 km height is taken into account (in our preceding publication any part of the SABER profile needed to fit the geographical selection criteria). Finally, the dry-adiabatic lapse rate is assumed to be 9.8 K/km in Wüst et al. (2017a),

now it is 9.5 K/km since the height-dependence of g is taken into account here. As figure 7 shows, the two approximations agree for the majority of the year within an uncertainty of 5%. This is in the range of the natural variability (see table 1). There is an offset visible, which is due to the height-dependence of the dry-adiabatic lapse rate. Furthermore, the data disagree especially where the ter-annual oscillation used by Wüst et al. (2017a) has a maximum. This oscillation is not used here since tests showed that it appears very prominently at low latitudes in order to approximate at least in parts the

oscillation of ca. 60 d. Even though temperature data at mid-latitudes also show a slight ter-annual course as discussed above, it is not clear until which latitude the ter-annual oscillation might be real and from which latitude on it might be artificial. Therefore, we propose to use the values of Wüst et al. (2017a) for investigations which do not comprise a direct comparison with results from stations not within 43.93–48.09° N and 5.71–12.95° E. In any other case, the values of this manuscript should be applied.

Even though a climatology of the OH* layer height and its FWHM is not in the focus of this manuscript, it might be of interest for some scientific groups. Therefore, the harmonic analysis is applied to the time series of daily mean values of the OH* layer height and of the FWHM in the same way as it was used for the approximation of the BV frequency. The results are listed in the appendix (table 2 and table 3).

## 4 Summary and outlook

We provide a climatology of the OH*-equivalent BV frequency based on 17 years of TIMED-SABER data for mid and low latitudes on a $10° \times 20°$ grid. This is done in order to facilitate the estimation of the density of gravity wave potential energy

from airglow temperature measurements independent of co-located measurements which deliver vertical temperature profiles. This manuscript is the succeeding work of Wüst et al. (2017a) who published a climatology for the Alpine region only.

Especially at low latitudes, a prominent 60 d oscillation is present in the times series of daily OH*-equivalent BV frequency averaged over all years. This might be an artificial signal due to a combination of the yaw cycle of SABER and strong tidal

activity. Physical reasons for the annual and semi-annual oscillations, which are also present in the data, are known. Therefore, an analytical formulation for the approximation of the daily OH*-equivalent BV frequency based on the superposition of a mean value and an annual and a semi-annual oscillation is provided. It is estimated based on a harmonic analysis approach. Additional formulations for daily values of the OH* layer height and thickness are given.

## Author contribution

MGM and JMR were responsible for the TIMED-SABER data. JHY provided selected algorithms for the analysis of SABER data. SW formulated the research goals, analysed the data, wrote the paper and discussed it especially with MB and MGM.

## Acknowledgement

The work of Sabine Wüst was funded by the Bavarian State Ministry for the Environment and Consumer Protection (VoCaS-ALP, TKP01KPB-70581, and AlpEnDAC II, TUS01UFS-72184).
We thank Oleg Goussev, DLR, and Verena Wendt for their assistance in SABER data preparation.

## Competing interest

The authors declare that they have no conflict of interest.





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



**Figure captions**







**Figure 1** OH*-layer height (a and d), FWHM (b and e), and OH*-equivalent BV frequency (c and f) are shown for the latitudinal band of 45°N±5° (a-c) and 5°N±5° (d-f) depending on longitudinal bands (lon±10°, colour-coded) and DoY. The data are averaged for each DoY. Additionally, they are subject to a 31-points sliding mean. In order to facilitate comparison, the scales of the plots referring to 45°N±5° and 5°N±5° are identical.



**Figure 2** OH*-layer height (a), FWHM (b), and OH*-equivalent BV frequency (c) are averaged over all years and plotted for the latitudinal bands 55°S ± 5° to 55° N ±5° (colour-coded). Please be aware that the axes of the respective plots of figure 1 and 2 are not identical.





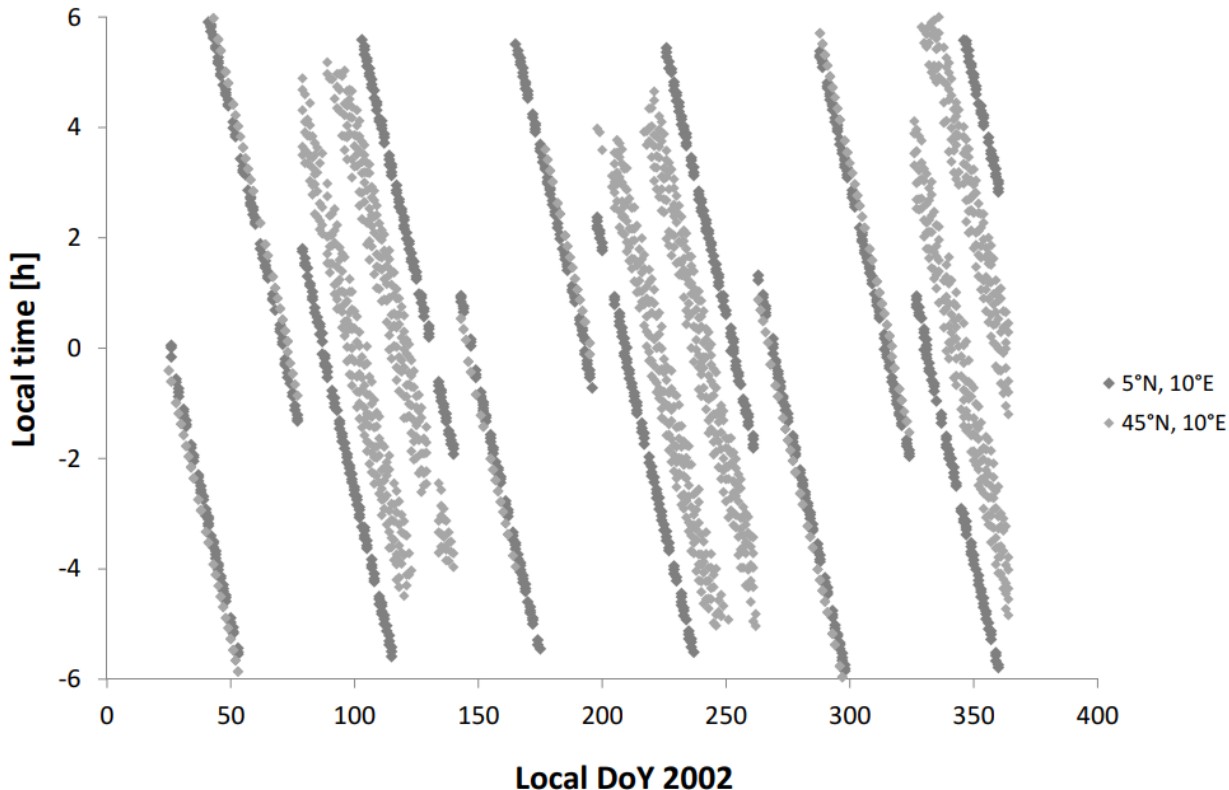

**Figure 3 Local overpass time of TIMED for the grid cells 5°N, 10°E (dark grey) and 45°N, 10°E (light grey) for the year 2002 during 6 p.m. and 6 a.m. A negative local time means that the respective profile was recorded before midnight (-2 LT = 10 p.m., -4 LT = 8 p.m., -6 LT = 6 p.m.). Daily averages are not computed, this plot shows the individual measurements (959 for 5°N and 1626 for 45°N).**





**Figure 4 Temperature and vertical temperature gradient, both VER weighted, for 5°N (a and b) and 45°N (c and d) for the year 2002. The nomenclature concerning the local time agrees with the one explained in the caption of figure 3. Subpanel e) shows the development of the BV-frequency during the night for 5°N, 10°E (grey) and 45°N, 10°E (black) based on the linear approximation of temperature and its vertical gradient.**





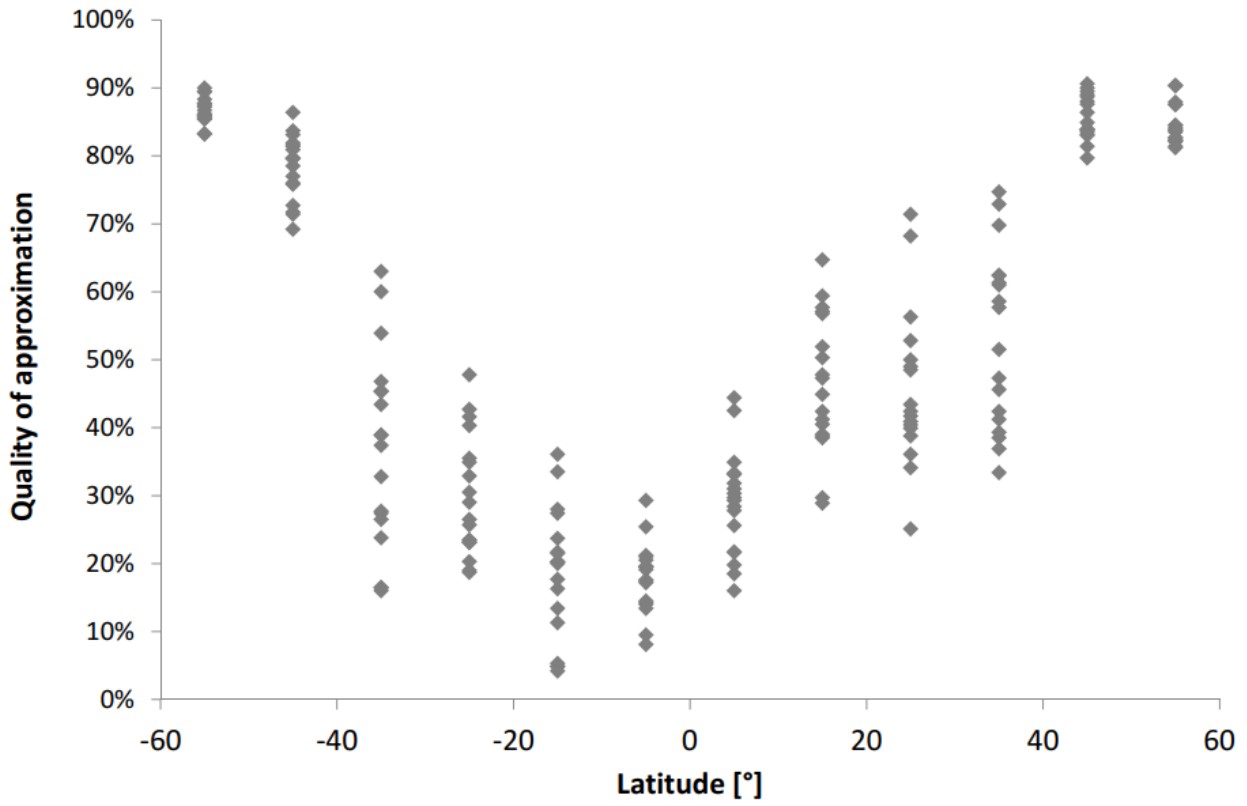

**Figure 5 Quality of approximation as shown in table 1 versus latitude. All longitudes are plotted but not separated by colour in order to keep the figure clear.**





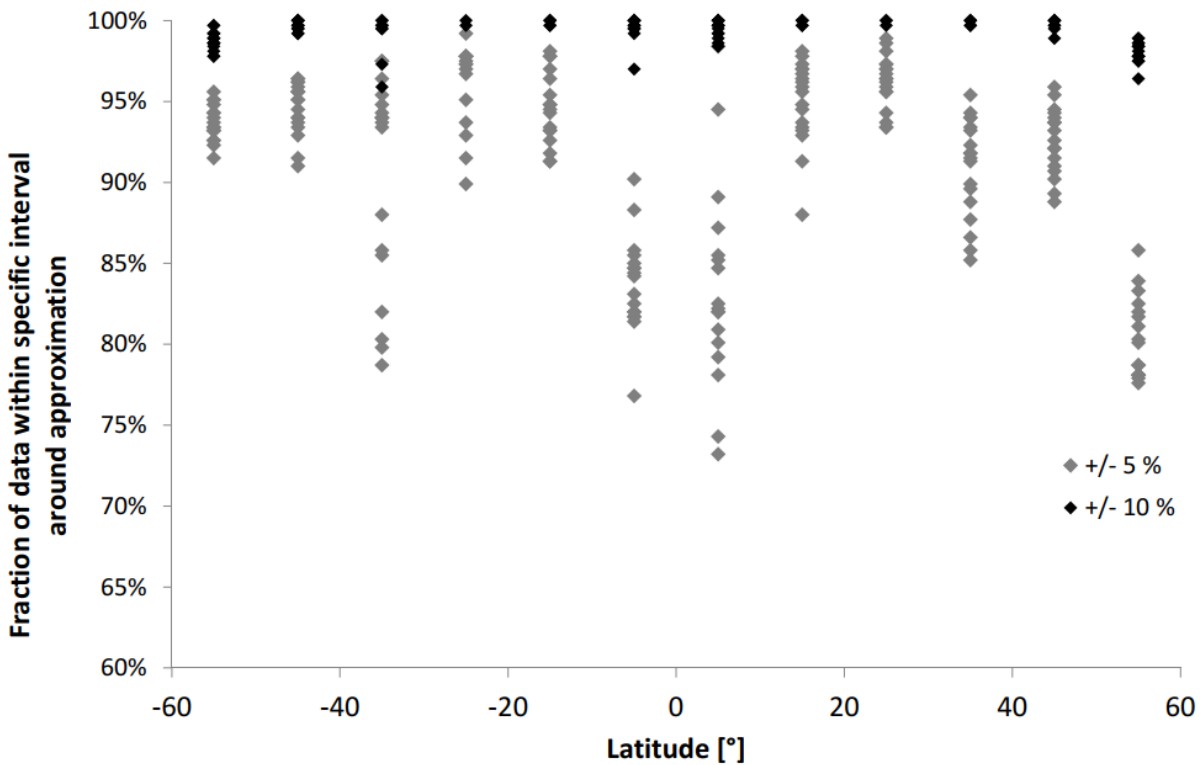

**Figure 6 Fraction of data within +/- 5 % and +/- 10 % intervals around approximation.**





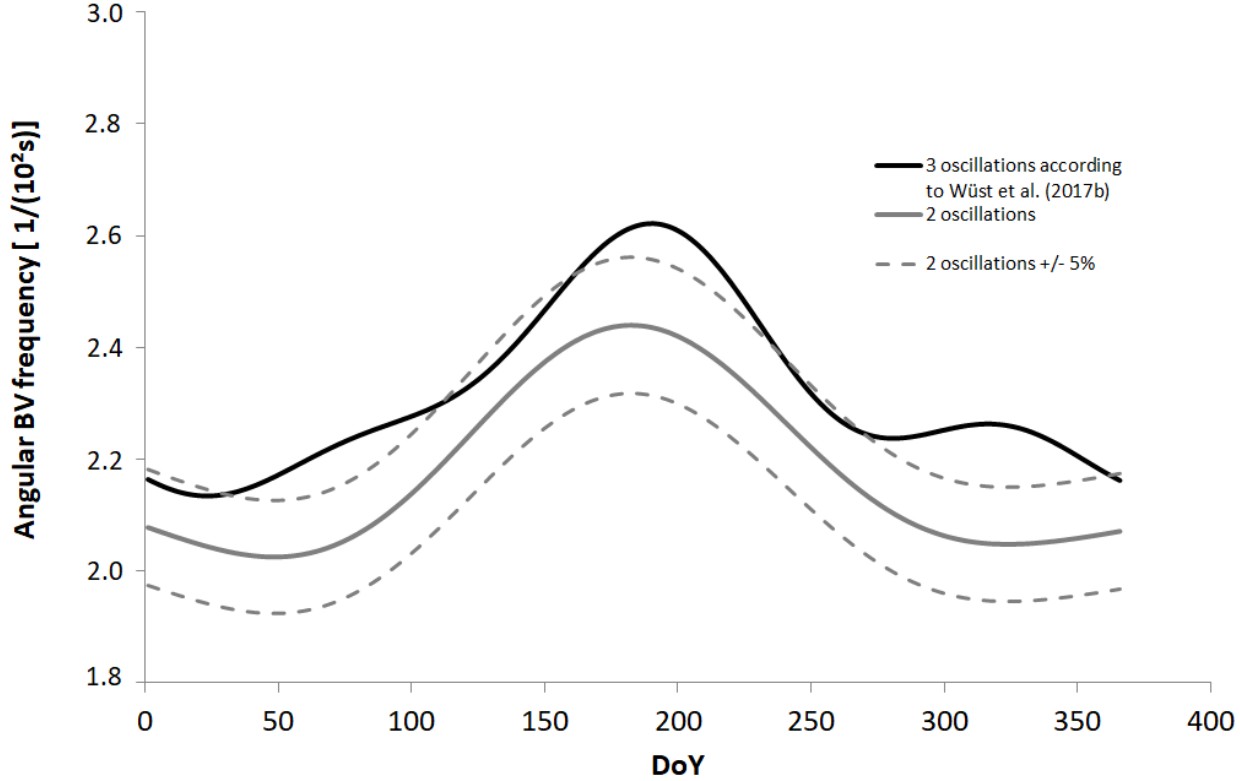

**Figure 7 Comparison of the approximation of the BV frequency in the Alpine region as it was proposed by Wüst et al. (2017a) based on three oscillations (black) and in this manuscript (gray, values of the pixel 45°N and 10°E are used).**





**Tables**

**Table 1**

5 **Period (T), amplitude (A) and phase (φ) of the two oscillations which explain the variability of the daily OH\*-equivalent BV frequency values (averaged over all years) best for a latitudinal and longitudinal gridding of 10° and 20°. They oscillate around the respective mean.**

**The OH\*-equivalent BV frequency [s⁻¹] can be estimated by mean $+ \sum_{i=1}^{2} A_i \, \sin\left(\frac{2\pi}{T_i} \cdot DoY - \varphi_i\right)$.**

**Due to leap years, the total amount of days for one year is set to 366, that means 1st March is DoY 61 for every year.**

**The harmonic oscillation explains the variability in the time series of daily OH\*-equivalent BV frequency values to a different**
10 **extent. The respective value is provided in the column "quality of approximation". Additionally, the fraction of data which lies within intervals of +/- 5 % or 10 % around the harmonic approximation is given.**

| Lat. | Lon. | Mean | 1st oscillation | | | 2nd oscillation | | | Quality of | Fraction of data | |
|---|---|---|---|---|---|---|---|---|---|---|---|
| [°] | [°] | [$10^{-2}$ s$^{-1}$] | Period [d] | Amp. [$10^{-2}$ s$^{-1}$] | Phase [rad] | Period [d] | Amp. [$10^{-2}$ s$^{-1}$] | Phase [rad] | approxi- mation | +/- 5% | +/- 10% |
| | | | | | | | | | | around approx. | |
| -55 | 10 | 2.19 | 366.00 | 0.24 | -1.56 | 192.59 | 0.13 | -1.68 | 0.87 | 0.92 | 0.99 |
| -55 | 30 | 2.18 | 366.00 | 0.23 | -1.61 | 189.32 | 0.12 | -1.62 | 0.87 | 0.95 | 0.99 |
| -55 | 50 | 2.18 | 366.00 | 0.22 | -1.62 | 187.92 | 0.11 | -1.51 | 0.85 | 0.93 | 0.99 |
| -55 | 70 | 2.17 | 366.00 | 0.21 | -1.61 | 188.86 | 0.10 | -1.60 | 0.83 | 0.93 | 0.99 |
| -55 | 90 | 2.18 | 366.00 | 0.22 | -1.59 | 190.26 | 0.09 | -1.59 | 0.87 | 0.94 | 0.99 |
| -55 | 110 | 2.18 | 366.00 | 0.22 | -1.58 | 188.39 | 0.10 | -1.56 | 0.83 | 0.93 | 0.98 |
| -55 | 130 | 2.17 | 366.00 | 0.24 | -1.56 | 189.79 | 0.12 | -1.62 | 0.88 | 0.95 | 0.99 |
| -55 | 150 | 2.15 | 366.00 | 0.22 | -1.58 | 190.26 | 0.11 | -1.58 | 0.86 | 0.94 | 1.00 |
| -55 | 170 | 2.15 | 366.00 | 0.22 | -1.59 | 189.32 | 0.11 | -1.71 | 0.86 | 0.93 | 0.98 |
| -55 | 190 | 2.15 | 366.00 | 0.23 | -1.58 | 188.86 | 0.12 | -1.69 | 0.88 | 0.93 | 0.98 |
| -55 | 210 | 2.15 | 366.00 | 0.25 | -1.58 | 188.39 | 0.13 | -1.67 | 0.89 | 0.93 | 0.99 |
| -55 | 230 | 2.14 | 366.00 | 0.24 | -1.58 | 186.06 | 0.10 | -1.61 | 0.90 | 0.96 | 0.99 |
| -55 | 250 | 2.15 | 366.00 | 0.25 | -1.54 | 189.32 | 0.10 | -1.66 | 0.90 | 0.95 | 0.99 |
| -55 | 270 | 2.16 | 366.00 | 0.24 | -1.54 | 189.32 | 0.11 | -1.69 | 0.86 | 0.93 | 0.99 |
| -55 | 290 | 2.17 | 366.00 | 0.23 | -1.56 | 188.39 | 0.11 | -1.77 | 0.86 | 0.92 | 0.99 |
| -55 | 310 | 2.17 | 366.00 | 0.22 | -1.56 | 188.86 | 0.11 | -1.77 | 0.90 | 0.94 | 1.00 |
| -55 | 330 | 2.17 | 366.00 | 0.22 | -1.55 | 189.32 | 0.11 | -1.76 | 0.88 | 0.94 | 0.98 |
| -55 | 350 | 2.18 | 366.00 | 0.22 | -1.54 | 191.19 | 0.11 | -1.66 | 0.86 | 0.94 | 0.99 |
| -45 | 10 | 2.18 | 366.00 | 0.16 | -1.52 | 201.91 | 0.05 | -1.90 | 0.72 | 0.91 | 0.99 |
| -45 | 30 | 2.18 | 366.00 | 0.17 | -1.55 | 195.38 | 0.05 | -1.77 | 0.80 | 0.93 | 1.00 |





| -45 | 50 | 2.18 | 366.00 | 0.15 | -1.56 | 195.38 | 0.04 | -1.53 | 0.76 | 0.94 | 0.99 |
|---|---|---|---|---|---|---|---|---|---|---|---|
| -45 | 70 | 2.17 | 366.00 | 0.14 | -1.50 | 204.71 | 0.03 | -1.80 | 0.69 | 0.92 | 1.00 |
| -45 | 90 | 2.17 | 366.00 | 0.15 | -1.49 | 192.12 | 0.03 | -2.04 | 0.80 | 0.96 | 1.00 |
| -45 | 110 | 2.18 | 366.00 | 0.16 | -1.50 | 194.45 | 0.04 | -1.81 | 0.84 | 0.96 | 1.00 |
| -45 | 130 | 2.18 | 366.00 | 0.18 | -1.51 | 193.52 | 0.05 | -1.82 | 0.86 | 0.96 | 1.00 |
| -45 | 150 | 2.16 | 366.00 | 0.16 | -1.50 | 193.05 | 0.04 | -1.71 | 0.80 | 0.95 | 1.00 |
| -45 | 170 | 2.16 | 366.00 | 0.15 | -1.51 | 191.65 | 0.04 | -1.93 | 0.81 | 0.96 | 1.00 |
| -45 | 190 | 2.16 | 366.00 | 0.16 | -1.46 | 193.52 | 0.05 | -1.85 | 0.82 | 0.96 | 1.00 |
| -45 | 210 | 2.16 | 366.00 | 0.17 | -1.48 | 193.98 | 0.05 | -1.76 | 0.83 | 0.96 | 1.00 |
| -45 | 230 | 2.16 | 366.00 | 0.15 | -1.51 | 190.26 | 0.04 | -1.85 | 0.81 | 0.96 | 1.00 |
| -45 | 250 | 2.17 | 366.00 | 0.14 | -1.45 | 192.12 | 0.03 | -1.84 | 0.77 | 0.95 | 1.00 |
| -45 | 270 | 2.18 | 366.00 | 0.13 | -1.44 | 186.06 | 0.04 | -1.92 | 0.73 | 0.94 | 1.00 |
| -45 | 290 | 2.18 | 366.00 | 0.15 | -1.47 | 184.20 | 0.06 | -1.97 | 0.79 | 0.94 | 1.00 |
| -45 | 310 | 2.16 | 366.00 | 0.15 | -1.52 | 191.19 | 0.05 | -2.03 | 0.82 | 0.96 | 1.00 |
| -45 | 330 | 2.16 | 366.00 | 0.14 | -1.50 | 195.38 | 0.04 | -2.14 | 0.76 | 0.95 | 1.00 |
| -45 | 350 | 2.18 | 366.00 | 0.15 | -1.51 | 200.05 | 0.04 | -2.06 | 0.71 | 0.93 | 1.00 |
| -35 | 10 | 2.22 | 365.53 | 0.07 | -1.12 | 180.00 | 0.04 | 1.93 | 0.27 | 0.80 | 0.96 |
| -35 | 30 | 2.22 | 315.65 | 0.08 | -0.72 | 180.00 | 0.04 | 1.79 | 0.43 | 0.86 | 1.00 |
| -35 | 50 | 2.22 | 255.98 | 0.08 | 0.17 | 180.00 | 0.03 | 1.16 | 0.39 | 0.80 | 1.00 |
| -35 | 70 | 2.21 | 228.95 | 0.05 | 0.17 | 366.00 | 0.04 | -0.51 | 0.27 | 0.82 | 1.00 |
| -35 | 90 | 2.21 | 328.24 | 0.05 | -0.59 | 180.00 | 0.04 | 1.97 | 0.33 | 0.88 | 1.00 |
| -35 | 110 | 2.22 | 350.62 | 0.07 | -1.06 | 180.00 | 0.03 | 2.13 | 0.47 | 0.94 | 1.00 |
| -35 | 130 | 2.23 | 354.35 | 0.09 | -1.21 | 180.00 | 0.02 | 2.62 | 0.60 | 0.98 | 1.00 |
| -35 | 150 | 2.22 | 323.58 | 0.06 | -0.76 | 180.00 | 0.02 | 1.77 | 0.39 | 0.94 | 1.00 |
| -35 | 170 | 2.20 | 275.10 | 0.04 | 0.30 | 180.00 | 0.02 | 1.72 | 0.24 | 0.95 | 1.00 |
| -35 | 190 | 2.21 | 333.37 | 0.07 | -0.71 | 180.00 | 0.02 | 2.15 | 0.45 | 0.95 | 1.00 |
| -35 | 210 | 2.23 | 366.00 | 0.09 | -1.16 | 180.00 | 0.01 | 1.71 | 0.63 | 0.98 | 1.00 |
| -35 | 230 | 2.23 | 312.39 | 0.05 | -0.67 | 180.00 | 0.01 | 1.74 | 0.37 | 0.96 | 1.00 |
| -35 | 250 | 2.24 | 199.11 | 0.03 | 1.28 | 366.00 | 0.02 | -0.19 | 0.17 | 0.94 | 1.00 |
| -35 | 270 | 2.23 | 325.91 | 0.03 | 0.16 | 180.00 | 0.02 | 3.03 | 0.16 | 0.93 | 1.00 |
| -35 | 290 | 2.20 | 350.62 | 0.06 | -0.88 | 180.00 | 0.03 | 3.11 | 0.45 | 0.94 | 1.00 |
| -35 | 310 | 2.19 | 342.69 | 0.08 | -1.04 | 180.00 | 0.03 | 2.44 | 0.54 | 0.94 | 1.00 |
| -35 | 330 | 2.19 | 222.89 | 0.05 | 0.61 | 366.00 | 0.02 | -0.90 | 0.28 | 0.86 | 1.00 |
| -35 | 350 | 2.21 | 215.89 | 0.04 | 0.73 | 366.00 | 0.02 | -1.28 | 0.16 | 0.79 | 0.97 |
| -25 | 10 | 2.20 | 180.00 | 0.04 | 1.97 | 366.00 | 0.03 | -0.71 | 0.23 | 0.92 | 1.00 |
| -25 | 30 | 2.20 | 245.73 | 0.05 | -0.13 | 366.00 | 0.03 | -0.33 | 0.43 | 0.98 | 1.00 |
| -25 | 50 | 2.21 | 213.56 | 0.06 | 0.55 | 366.00 | 0.04 | -0.40 | 0.42 | 0.97 | 1.00 |





| | | | | | | | | | | | |
|---|---|---|---|---|---|---|---|---|---|---|---|
| -25 | 70 | 2.21 | 180.00 | 0.05 | 1.63 | 366.00 | 0.03 | -0.45 | 0.35 | 0.94 | 1.00 |
| -25 | 90 | 2.19 | 180.00 | 0.04 | 1.69 | 366.00 | 0.02 | 0.04 | 0.26 | 0.95 | 1.00 |
| -25 | 110 | 2.20 | 334.30 | 0.03 | -0.50 | 180.00 | 0.02 | 1.65 | 0.29 | 0.99 | 1.00 |
| -25 | 130 | 2.22 | 354.81 | 0.06 | -1.17 | 180.00 | 0.02 | 1.44 | 0.48 | 0.98 | 1.00 |
| -25 | 150 | 2.21 | 207.97 | 0.03 | 0.70 | 366.00 | 0.02 | -0.77 | 0.19 | 0.97 | 1.00 |
| -25 | 170 | 2.19 | 180.00 | 0.04 | 1.56 | 292.35 | 0.03 | 1.61 | 0.31 | 0.98 | 1.00 |
| -25 | 190 | 2.19 | 366.00 | 0.03 | 0.42 | 188.86 | 0.01 | 1.27 | 0.19 | 0.98 | 1.00 |
| -25 | 210 | 2.22 | 366.00 | 0.05 | -0.79 | 221.49 | 0.01 | -1.06 | 0.36 | 0.97 | 1.00 |
| -25 | 230 | 2.24 | 265.77 | 0.04 | 0.43 | 180.00 | 0.02 | 0.57 | 0.20 | 0.98 | 1.00 |
| -25 | 250 | 2.25 | 366.00 | 0.04 | 1.45 | 192.12 | 0.03 | 1.50 | 0.27 | 0.97 | 1.00 |
| -25 | 270 | 2.22 | 366.00 | 0.05 | 1.28 | 180.00 | 0.03 | 2.74 | 0.40 | 0.97 | 1.00 |
| -25 | 290 | 2.19 | 324.05 | 0.03 | 0.74 | 180.00 | 0.02 | 2.86 | 0.23 | 0.98 | 1.00 |
| -25 | 310 | 2.18 | 335.23 | 0.04 | -0.47 | 180.00 | 0.02 | 2.48 | 0.33 | 0.98 | 1.00 |
| -25 | 330 | 2.19 | 180.00 | 0.04 | 1.83 | 366.00 | 0.02 | -0.97 | 0.24 | 0.93 | 1.00 |
| -25 | 350 | 2.21 | 180.00 | 0.06 | 2.15 | 238.74 | 0.03 | -1.79 | 0.23 | 0.90 | 1.00 |
| -15 | 10 | 2.13 | 180.00 | 0.01 | -1.49 | 363.67 | 0.01 | -2.97 | 0.04 | 0.95 | 1.00 |
| -15 | 30 | 2.12 | 366.00 | 0.03 | -1.50 | 180.00 | 0.01 | -1.13 | 0.18 | 0.93 | 1.00 |
| -15 | 50 | 2.13 | 303.07 | 0.03 | -0.58 | 180.00 | 0.01 | -0.04 | 0.20 | 0.98 | 1.00 |
| -15 | 70 | 2.13 | 212.17 | 0.04 | 1.54 | 180.00 | 0.04 | 0.02 | 0.13 | 0.97 | 1.00 |
| -15 | 90 | 2.11 | 366.00 | 0.01 | 2.15 | 180.00 | 0.01 | 0.17 | 0.05 | 0.93 | 1.00 |
| -15 | 110 | 2.11 | 180.00 | 0.02 | -0.67 | 245.73 | 0.01 | 0.26 | 0.05 | 0.92 | 1.00 |
| -15 | 130 | 2.13 | 366.00 | 0.06 | -1.41 | 197.71 | 0.03 | -0.59 | 0.36 | 0.92 | 1.00 |
| -15 | 150 | 2.13 | 180.00 | 0.04 | 0.38 | 366.00 | 0.03 | -1.93 | 0.27 | 0.95 | 1.00 |
| -15 | 170 | 2.12 | 366.00 | 0.04 | 2.47 | 180.00 | 0.02 | 1.00 | 0.24 | 0.94 | 1.00 |
| -15 | 190 | 2.10 | 361.34 | 0.03 | 1.55 | 232.68 | 0.01 | -1.46 | 0.16 | 0.91 | 1.00 |
| -15 | 210 | 2.13 | 184.20 | 0.04 | -0.95 | 366.00 | 0.03 | -0.91 | 0.22 | 0.93 | 1.00 |
| -15 | 230 | 2.15 | 366.00 | 0.03 | -1.79 | 180.00 | 0.02 | -1.22 | 0.20 | 0.95 | 1.00 |
| -15 | 250 | 2.15 | 366.00 | 0.05 | 2.15 | 236.41 | 0.01 | 0.98 | 0.28 | 0.98 | 1.00 |
| -15 | 270 | 2.14 | 366.00 | 0.06 | 1.78 | 233.14 | 0.01 | -0.89 | 0.34 | 0.95 | 1.00 |
| -15 | 290 | 2.11 | 307.73 | 0.02 | 2.68 | 180.00 | 0.01 | -1.37 | 0.05 | 0.93 | 1.00 |
| -15 | 310 | 2.11 | 361.80 | 0.03 | -1.17 | 231.28 | 0.01 | -1.19 | 0.11 | 0.91 | 1.00 |
| -15 | 330 | 2.13 | 274.63 | 0.04 | -1.26 | 187.46 | 0.01 | 2.33 | 0.20 | 0.96 | 1.00 |
| -15 | 350 | 2.14 | 348.75 | 0.03 | -3.09 | 204.71 | 0.01 | 0.74 | 0.22 | 0.98 | 1.00 |
| -5 | 10 | 2.06 | 180.00 | 0.04 | -0.97 | 366.00 | 0.04 | 2.47 | 0.21 | 0.82 | 1.00 |
| -5 | 30 | 2.05 | 180.00 | 0.04 | -1.16 | 366.00 | 0.01 | 2.80 | 0.15 | 0.82 | 1.00 |
| -5 | 50 | 2.04 | 180.00 | 0.03 | -1.42 | 366.00 | 0.01 | 2.06 | 0.08 | 0.83 | 1.00 |
| -5 | 70 | 2.04 | 306.33 | 0.03 | 2.46 | 180.00 | 0.02 | -1.17 | 0.13 | 0.81 | 1.00 |





| | | | | | | | | | | |
|---|---|---|---|---|---|---|---|---|---|---|
| -5 | 90 | 2.04 | 249.92 | 0.04 | 3.00 | 180.00 | 0.02 | -0.60 | 0.18 | 0.77 | 0.99 |
| -5 | 110 | 2.04 | 205.64 | 0.05 | -1.80 | 366.00 | 0.02 | 1.41 | 0.17 | 0.82 | 0.97 |
| -5 | 130 | 2.05 | 181.40 | 0.05 | -1.01 | 366.00 | 0.02 | -1.06 | 0.20 | 0.82 | 1.00 |
| -5 | 150 | 2.04 | 186.99 | 0.04 | -0.83 | 366.00 | 0.01 | -2.84 | 0.21 | 0.90 | 1.00 |
| -5 | 170 | 2.05 | 310.06 | 0.05 | -3.01 | 180.00 | 0.01 | 0.28 | 0.29 | 0.88 | 1.00 |
| -5 | 190 | 2.05 | 299.80 | 0.05 | 3.04 | 214.96 | 0.01 | 0.12 | 0.20 | 0.85 | 1.00 |
| -5 | 210 | 2.05 | 194.92 | 0.05 | -1.24 | 366.00 | 0.02 | -1.62 | 0.19 | 0.86 | 1.00 |
| -5 | 230 | 2.05 | 180.00 | 0.05 | -1.06 | 366.00 | 0.02 | -2.40 | 0.25 | 0.85 | 1.00 |
| -5 | 250 | 2.05 | 227.55 | 0.04 | -2.41 | 366.00 | 0.02 | 2.65 | 0.17 | 0.86 | 1.00 |
| -5 | 270 | 2.06 | 249.46 | 0.04 | -2.43 | 180.00 | 0.02 | -0.24 | 0.20 | 0.83 | 1.00 |
| -5 | 290 | 2.04 | 180.00 | 0.04 | -0.39 | 366.00 | 0.02 | 2.79 | 0.14 | 0.85 | 1.00 |
| -5 | 310 | 2.03 | 267.64 | 0.03 | -0.91 | 180.00 | 0.02 | -0.85 | 0.10 | 0.84 | 1.00 |
| -5 | 330 | 2.04 | 314.26 | 0.04 | -2.05 | 180.00 | 0.02 | -0.65 | 0.14 | 0.84 | 1.00 |
| -5 | 350 | 2.06 | 345.49 | 0.05 | 2.99 | 180.00 | 0.02 | -0.82 | 0.21 | 0.84 | 1.00 |
| 5 | 10 | 2.05 | 349.68 | 0.07 | 1.84 | 180.00 | 0.02 | -1.05 | 0.30 | 0.79 | 0.99 |
| 5 | 30 | 2.03 | 297.94 | 0.06 | 2.16 | 180.00 | 0.03 | -0.58 | 0.29 | 0.82 | 0.99 |
| 5 | 50 | 2.02 | 265.77 | 0.06 | 2.50 | 180.00 | 0.03 | -0.47 | 0.30 | 0.74 | 1.00 |
| 5 | 70 | 2.04 | 269.04 | 0.08 | 2.47 | 180.00 | 0.03 | -0.58 | 0.35 | 0.73 | 0.98 |
| 5 | 90 | 2.06 | 275.56 | 0.09 | 2.50 | 180.00 | 0.03 | -0.63 | 0.43 | 0.78 | 0.99 |
| 5 | 110 | 2.06 | 266.24 | 0.09 | 2.70 | 180.00 | 0.02 | -0.58 | 0.44 | 0.82 | 1.00 |
| 5 | 130 | 2.05 | 234.54 | 0.06 | -2.66 | 366.00 | 0.03 | 0.48 | 0.33 | 0.83 | 1.00 |
| 5 | 150 | 2.03 | 232.68 | 0.07 | -3.07 | 180.00 | 0.02 | 0.20 | 0.32 | 0.81 | 0.99 |
| 5 | 170 | 2.08 | 317.05 | 0.05 | 2.62 | 180.00 | 0.02 | -0.03 | 0.28 | 0.87 | 1.00 |
| 5 | 190 | 2.09 | 366.00 | 0.05 | 1.94 | 180.00 | 0.01 | 0.35 | 0.33 | 0.98 | 1.00 |
| 5 | 210 | 2.05 | 235.47 | 0.04 | -2.67 | 366.00 | 0.01 | 0.80 | 0.26 | 0.95 | 1.00 |
| 5 | 230 | 2.03 | 220.09 | 0.06 | -2.87 | 180.00 | 0.02 | 0.04 | 0.28 | 0.86 | 1.00 |
| 5 | 250 | 2.05 | 241.53 | 0.05 | 3.00 | 180.00 | 0.02 | -0.27 | 0.22 | 0.80 | 1.00 |
| 5 | 270 | 2.07 | 249.46 | 0.04 | 2.85 | 180.00 | 0.03 | 0.03 | 0.20 | 0.89 | 1.00 |
| 5 | 290 | 2.06 | 240.14 | 0.04 | 2.81 | 180.00 | 0.04 | -0.03 | 0.22 | 0.89 | 1.00 |
| 5 | 310 | 2.03 | 253.19 | 0.04 | 2.61 | 180.00 | 0.02 | -0.09 | 0.16 | 0.85 | 1.00 |
| 5 | 330 | 2.04 | 258.32 | 0.04 | 2.70 | 180.00 | 0.02 | -0.20 | 0.19 | 0.85 | 1.00 |
| 5 | 350 | 2.06 | 344.56 | 0.06 | 1.97 | 180.00 | 0.02 | -1.03 | 0.31 | 0.82 | 1.00 |
| 15 | 10 | 2.12 | 366.00 | 0.08 | 1.57 | 228.02 | 0.02 | 1.69 | 0.57 | 0.96 | 1.00 |
| 15 | 30 | 2.10 | 366.00 | 0.07 | 1.55 | 242.93 | 0.01 | 0.64 | 0.47 | 0.95 | 1.00 |
| 15 | 50 | 2.09 | 360.87 | 0.07 | 1.51 | 180.00 | 0.01 | -0.56 | 0.42 | 0.93 | 1.00 |
| 15 | 70 | 2.12 | 352.02 | 0.07 | 1.58 | 180.00 | 0.01 | -0.59 | 0.48 | 0.93 | 1.00 |
| 15 | 90 | 2.15 | 362.74 | 0.08 | 1.65 | 180.00 | 0.01 | -0.27 | 0.57 | 0.96 | 1.00 |





| | | | | | | | | | | | |
|---|---|---|---|---|---|---|---|---|---|---|---|
| 15 | 110 | 2.15 | 366.00 | 0.09 | 1.68 | 180.00 | 0.01 | 0.85 | 0.50 | 0.91 | 1.00 |
| 15 | 130 | 2.12 | 366.00 | 0.08 | 1.62 | 180.00 | 0.01 | 1.19 | 0.41 | 0.88 | 1.00 |
| 15 | 150 | 2.10 | 290.95 | 0.06 | 2.26 | 180.00 | 0.01 | -0.34 | 0.39 | 0.93 | 1.00 |
| 15 | 170 | 2.16 | 366.00 | 0.05 | 1.92 | 180.00 | 0.01 | 0.42 | 0.29 | 0.96 | 1.00 |
| 15 | 190 | 2.17 | 366.00 | 0.06 | 1.64 | 194.92 | 0.02 | 1.18 | 0.39 | 0.97 | 1.00 |
| 15 | 210 | 2.14 | 363.20 | 0.08 | 1.55 | 246.20 | 0.01 | 1.33 | 0.58 | 0.97 | 1.00 |
| 15 | 230 | 2.11 | 333.37 | 0.09 | 1.81 | 180.00 | 0.02 | -1.56 | 0.59 | 0.94 | 1.00 |
| 15 | 250 | 2.13 | 356.68 | 0.06 | 1.41 | 180.00 | 0.01 | -2.44 | 0.41 | 0.95 | 1.00 |
| 15 | 270 | 2.16 | 366.00 | 0.05 | 1.52 | 218.69 | 0.01 | 0.80 | 0.30 | 0.98 | 1.00 |
| 15 | 290 | 2.14 | 318.45 | 0.05 | 2.00 | 180.00 | 0.02 | -0.91 | 0.39 | 0.97 | 1.00 |
| 15 | 310 | 2.11 | 273.70 | 0.06 | 2.45 | 180.00 | 0.01 | -1.04 | 0.45 | 0.96 | 1.00 |
| 15 | 330 | 2.12 | 298.87 | 0.07 | 2.02 | 180.00 | 0.01 | -1.51 | 0.52 | 0.97 | 1.00 |
| 15 | 350 | 2.13 | 357.14 | 0.08 | 1.66 | 213.56 | 0.02 | 2.61 | 0.65 | 0.98 | 1.00 |
| 25 | 10 | 2.20 | 366.00 | 0.07 | 1.44 | 199.58 | 0.03 | 1.46 | 0.36 | 0.93 | 1.00 |
| 25 | 30 | 2.18 | 366.00 | 0.06 | 1.39 | 193.05 | 0.04 | 1.15 | 0.40 | 0.94 | 1.00 |
| 25 | 50 | 2.18 | 366.00 | 0.05 | 1.40 | 190.72 | 0.04 | 1.12 | 0.42 | 0.97 | 1.00 |
| 25 | 70 | 2.20 | 180.00 | 0.05 | 1.49 | 366.00 | 0.05 | 1.46 | 0.50 | 0.99 | 1.00 |
| 25 | 90 | 2.24 | 366.00 | 0.07 | 1.67 | 186.99 | 0.05 | 1.07 | 0.50 | 0.96 | 1.00 |
| 25 | 110 | 2.21 | 366.00 | 0.07 | 1.87 | 191.19 | 0.05 | 1.18 | 0.53 | 0.96 | 1.00 |
| 25 | 130 | 2.18 | 366.00 | 0.07 | 1.81 | 188.86 | 0.04 | 1.31 | 0.49 | 0.94 | 1.00 |
| 25 | 150 | 2.17 | 366.00 | 0.05 | 1.86 | 182.80 | 0.02 | 1.11 | 0.42 | 0.97 | 1.00 |
| 25 | 170 | 2.22 | 366.00 | 0.06 | 1.80 | 184.66 | 0.03 | 1.11 | 0.40 | 0.96 | 1.00 |
| 25 | 190 | 2.24 | 366.00 | 0.07 | 1.66 | 194.92 | 0.05 | 1.13 | 0.49 | 0.97 | 1.00 |
| 25 | 210 | 2.22 | 366.00 | 0.10 | 1.54 | 191.65 | 0.03 | 1.30 | 0.68 | 0.97 | 1.00 |
| 25 | 230 | 2.21 | 366.00 | 0.11 | 1.50 | 201.91 | 0.01 | 1.89 | 0.71 | 0.97 | 1.00 |
| 25 | 250 | 2.23 | 366.00 | 0.08 | 1.30 | 187.46 | 0.03 | 1.79 | 0.56 | 0.96 | 1.00 |
| 25 | 270 | 2.25 | 366.00 | 0.06 | 1.40 | 191.65 | 0.03 | 1.31 | 0.39 | 0.96 | 1.00 |
| 25 | 290 | 2.22 | 366.00 | 0.05 | 1.76 | 192.12 | 0.02 | 1.18 | 0.25 | 0.96 | 1.00 |
| 25 | 310 | 2.19 | 366.00 | 0.05 | 1.87 | 200.51 | 0.01 | 2.19 | 0.34 | 0.99 | 1.00 |
| 25 | 330 | 2.20 | 366.00 | 0.06 | 1.59 | 199.11 | 0.01 | 2.35 | 0.43 | 0.98 | 1.00 |
| 25 | 350 | 2.22 | 366.00 | 0.07 | 1.52 | 206.57 | 0.02 | 1.85 | 0.41 | 0.93 | 1.00 |
| 35 | 10 | 2.23 | 366.00 | 0.08 | 1.49 | 196.78 | 0.02 | 1.82 | 0.39 | 0.86 | 1.00 |
| 35 | 30 | 2.22 | 366.00 | 0.08 | 1.51 | 195.38 | 0.03 | 1.62 | 0.41 | 0.89 | 1.00 |
| 35 | 50 | 2.22 | 366.00 | 0.08 | 1.49 | 198.18 | 0.02 | 1.20 | 0.33 | 0.88 | 1.00 |
| 35 | 70 | 2.23 | 366.00 | 0.08 | 1.50 | 191.65 | 0.03 | 0.94 | 0.39 | 0.90 | 1.00 |
| 35 | 90 | 2.24 | 366.00 | 0.10 | 1.67 | 191.19 | 0.03 | 0.81 | 0.52 | 0.87 | 1.00 |
| 35 | 110 | 2.21 | 366.00 | 0.12 | 1.73 | 199.11 | 0.03 | 0.73 | 0.58 | 0.90 | 1.00 |



| | | | | | | | | | | | |
|---|---|---|---|---|---|---|---|---|---|---|---|
| 35 | 130 | 2.20 | 366.00 | 0.12 | 1.72 | 199.58 | 0.02 | 0.73 | 0.61 | 0.93 | 1.00 |
| 35 | 150 | 2.20 | 366.00 | 0.11 | 1.70 | 194.45 | 0.02 | 0.70 | 0.62 | 0.94 | 1.00 |
| 35 | 170 | 2.23 | 366.00 | 0.11 | 1.66 | 202.38 | 0.02 | 0.69 | 0.61 | 0.92 | 1.00 |
| 35 | 190 | 2.24 | 366.00 | 0.12 | 1.64 | 199.58 | 0.03 | 1.03 | 0.62 | 0.92 | 1.00 |
| 35 | 210 | 2.23 | 366.00 | 0.14 | 1.56 | 193.05 | 0.03 | 1.28 | 0.73 | 0.94 | 1.00 |
| 35 | 230 | 2.23 | 366.00 | 0.14 | 1.55 | 197.25 | 0.02 | 1.42 | 0.75 | 0.95 | 1.00 |
| 35 | 250 | 2.24 | 366.00 | 0.13 | 1.49 | 201.44 | 0.02 | 1.58 | 0.70 | 0.94 | 1.00 |
| 35 | 270 | 2.25 | 366.00 | 0.11 | 1.51 | 194.92 | 0.01 | 1.64 | 0.59 | 0.93 | 1.00 |
| 35 | 290 | 2.23 | 366.00 | 0.07 | 1.75 | 200.05 | 0.01 | 1.46 | 0.37 | 0.92 | 1.00 |
| 35 | 310 | 2.21 | 366.00 | 0.07 | 1.84 | 180.00 | 0.02 | 3.13 | 0.42 | 0.92 | 1.00 |
| 35 | 330 | 2.23 | 366.00 | 0.09 | 1.71 | 180.00 | 0.02 | -2.81 | 0.47 | 0.91 | 1.00 |
| 35 | 350 | 2.24 | 366.00 | 0.10 | 1.58 | 185.13 | 0.02 | 3.09 | 0.46 | 0.85 | 1.00 |
| 45 | 10 | 2.20 | 324.51 | 0.19 | 2.02 | 180.00 | 0.05 | -1.61 | 0.84 | 0.91 | 1.00 |
| 45 | 30 | 2.20 | 315.19 | 0.17 | 2.13 | 180.00 | 0.04 | -1.48 | 0.81 | 0.89 | 1.00 |
| 45 | 50 | 2.20 | 304.47 | 0.16 | 2.22 | 180.00 | 0.04 | -1.47 | 0.80 | 0.89 | 1.00 |
| 45 | 70 | 2.20 | 310.53 | 0.17 | 2.18 | 180.00 | 0.04 | -1.27 | 0.83 | 0.92 | 1.00 |
| 45 | 90 | 2.21 | 310.99 | 0.19 | 2.22 | 180.00 | 0.04 | -1.13 | 0.85 | 0.90 | 1.00 |
| 45 | 110 | 2.20 | 316.59 | 0.20 | 2.16 | 180.00 | 0.04 | -1.19 | 0.86 | 0.92 | 1.00 |
| 45 | 130 | 2.19 | 311.92 | 0.20 | 2.21 | 180.00 | 0.04 | -1.27 | 0.86 | 0.92 | 1.00 |
| 45 | 150 | 2.19 | 328.24 | 0.20 | 2.02 | 180.00 | 0.04 | -1.29 | 0.88 | 0.94 | 1.00 |
| 45 | 170 | 2.19 | 330.57 | 0.20 | 1.99 | 180.00 | 0.04 | -1.39 | 0.89 | 0.95 | 1.00 |
| 45 | 190 | 2.19 | 344.56 | 0.21 | 1.80 | 180.00 | 0.04 | -1.55 | 0.89 | 0.95 | 1.00 |
| 45 | 210 | 2.19 | 353.41 | 0.23 | 1.72 | 180.00 | 0.04 | -1.40 | 0.91 | 0.95 | 1.00 |
| 45 | 230 | 2.18 | 354.35 | 0.24 | 1.68 | 180.00 | 0.04 | -1.46 | 0.90 | 0.96 | 1.00 |
| 45 | 250 | 2.20 | 351.08 | 0.25 | 1.68 | 180.00 | 0.04 | -1.55 | 0.90 | 0.93 | 0.99 |
| 45 | 270 | 2.20 | 348.29 | 0.22 | 1.74 | 180.00 | 0.04 | -1.60 | 0.88 | 0.94 | 1.00 |
| 45 | 290 | 2.19 | 341.29 | 0.18 | 1.88 | 180.00 | 0.04 | -1.53 | 0.84 | 0.94 | 1.00 |
| 45 | 310 | 2.18 | 332.90 | 0.17 | 2.01 | 180.00 | 0.04 | -1.79 | 0.83 | 0.94 | 1.00 |
| 45 | 330 | 2.19 | 327.77 | 0.18 | 2.04 | 180.00 | 0.04 | -1.67 | 0.84 | 0.93 | 1.00 |
| 45 | 350 | 2.21 | 330.57 | 0.19 | 1.98 | 180.00 | 0.05 | -1.59 | 0.84 | 0.91 | 1.00 |
| 55 | 10 | 2.20 | 285.35 | 0.26 | 2.50 | 180.00 | 0.06 | -1.41 | 0.81 | 0.78 | 0.98 |
| 55 | 30 | 2.20 | 279.76 | 0.25 | 2.54 | 180.00 | 0.05 | -1.55 | 0.83 | 0.78 | 0.98 |
| 55 | 50 | 2.21 | 279.29 | 0.24 | 2.57 | 180.00 | 0.05 | -1.66 | 0.82 | 0.80 | 0.98 |
| 55 | 70 | 2.21 | 282.09 | 0.23 | 2.56 | 180.00 | 0.05 | -1.43 | 0.83 | 0.82 | 0.99 |
| 55 | 90 | 2.22 | 280.23 | 0.25 | 2.59 | 180.00 | 0.05 | -1.38 | 0.84 | 0.79 | 0.99 |
| 55 | 110 | 2.21 | 276.50 | 0.25 | 2.65 | 180.00 | 0.05 | -1.33 | 0.81 | 0.78 | 0.98 |
| 55 | 130 | 2.20 | 274.63 | 0.25 | 2.68 | 180.00 | 0.05 | -1.45 | 0.82 | 0.78 | 0.98 |





| 55 | 150 | 2.20 | 278.83 | 0.25 | 2.60 | 180.00 | 0.05 | -1.45 | 0.84 | 0.81 | 0.98 |
| 55 | 170 | 2.20 | 276.03 | 0.26 | 2.64 | 180.00 | 0.05 | -1.55 | 0.82 | 0.78 | 0.98 |
| 55 | 190 | 2.19 | 284.89 | 0.26 | 2.48 | 180.00 | 0.06 | -1.60 | 0.85 | 0.78 | 0.98 |
| 55 | 210 | 2.18 | 302.60 | 0.28 | 2.25 | 180.00 | 0.07 | -1.51 | 0.88 | 0.83 | 0.99 |
| 55 | 230 | 2.17 | 316.59 | 0.31 | 2.05 | 180.00 | 0.08 | -1.57 | 0.90 | 0.83 | 0.99 |
| 55 | 250 | 2.18 | 319.85 | 0.33 | 2.01 | 180.00 | 0.08 | -1.58 | 0.88 | 0.80 | 0.98 |
| 55 | 270 | 2.17 | 323.11 | 0.30 | 2.01 | 180.00 | 0.08 | -1.57 | 0.90 | 0.84 | 0.99 |
| 55 | 290 | 2.18 | 310.99 | 0.27 | 2.14 | 180.00 | 0.07 | -1.73 | 0.88 | 0.86 | 0.98 |
| 55 | 310 | 2.18 | 294.21 | 0.25 | 2.39 | 180.00 | 0.06 | -1.77 | 0.84 | 0.82 | 0.96 |
| 55 | 330 | 2.17 | 291.88 | 0.25 | 2.43 | 180.00 | 0.05 | -1.63 | 0.85 | 0.79 | 0.99 |
| 55 | 350 | 2.19 | 287.68 | 0.26 | 2.47 | 180.00 | 0.06 | -1.58 | 0.84 | 0.78 | 0.98 |





## Appendix

**Table 2** Same as table 1 but for the OH*-layer height. Please pay attention to the fact that OH*-layer height is less variable during the year and therefore the fraction of data around the approximation is provided for intervals of +/- 1 % and +/- 2 %.

| Lat. | Lon. | Mean | 1st oscillation | | | 2nd oscillation | | | Quality of | Fraction of data | |
| --- | --- | --- | --- | --- | --- | --- | --- | --- | --- | --- | --- |
| [°] | [°] | $[10^{-2}\ s^{-1}]$ | Period [d] | Amp. $[10^{-2}\ s^{-1}]$ | Phase [rad] | Period [d] | Amp. $[10^{-2}\ s^{-1}]$ | Phase [rad] | approxi-mation | +-1% | +/- 2% |
| | | | | | | | | | | around approx. | |
| -55 | 10 | 86.43 | 340.36 | 1.30 | -1.51 | 187.92 | 0.32 | 0.20 | 0.78 | 0.90 | 1.00 |
| -55 | 30 | 86.43 | 358.54 | 1.45 | -1.57 | 208.90 | 0.38 | -0.77 | 0.79 | 0.93 | 0.99 |
| -55 | 50 | 86.39 | 361.34 | 1.43 | -1.58 | 194.92 | 0.30 | -0.16 | 0.80 | 0.91 | 0.99 |
| -55 | 70 | 86.20 | 347.82 | 1.27 | -1.46 | 187.92 | 0.24 | 0.59 | 0.67 | 0.88 | 0.98 |
| -55 | 90 | 86.16 | 355.74 | 1.34 | -1.45 | 213.56 | 0.26 | -0.49 | 0.74 | 0.92 | 0.98 |
| -55 | 110 | 86.16 | 362.27 | 1.52 | -1.51 | 216.83 | 0.30 | -0.89 | 0.80 | 0.93 | 0.99 |
| -55 | 130 | 86.14 | 366.00 | 1.69 | -1.51 | 219.16 | 0.33 | -1.06 | 0.85 | 0.93 | 0.99 |
| -55 | 150 | 86.07 | 354.81 | 1.47 | -1.41 | 210.77 | 0.38 | -0.58 | 0.78 | 0.91 | 0.99 |
| -55 | 170 | 86.04 | 355.28 | 1.43 | -1.47 | 207.97 | 0.27 | -0.42 | 0.78 | 0.90 | 0.98 |
| -55 | 190 | 86.03 | 351.08 | 1.41 | -1.40 | 203.31 | 0.42 | -0.36 | 0.78 | 0.89 | 0.99 |
| -55 | 210 | 86.11 | 355.28 | 1.44 | -1.47 | 196.32 | 0.39 | -0.15 | 0.79 | 0.89 | 0.99 |
| -55 | 230 | 86.17 | 350.62 | 1.44 | -1.57 | 180.00 | 0.29 | 0.58 | 0.78 | 0.87 | 0.98 |
| -55 | 250 | 86.26 | 350.62 | 1.48 | -1.55 | 180.93 | 0.31 | 0.62 | 0.81 | 0.91 | 1.00 |
| -55 | 270 | 86.29 | 338.03 | 1.29 | -1.40 | 207.04 | 0.40 | -0.67 | 0.72 | 0.85 | 0.98 |
| -55 | 290 | 86.28 | 334.30 | 1.37 | -1.44 | 197.71 | 0.38 | -0.31 | 0.76 | 0.86 | 0.99 |
| -55 | 310 | 86.30 | 348.29 | 1.40 | -1.59 | 183.26 | 0.33 | 0.43 | 0.80 | 0.90 | 1.00 |
| -55 | 330 | 86.32 | 343.16 | 1.34 | -1.56 | 180.00 | 0.30 | 0.83 | 0.80 | 0.92 | 0.99 |
| -55 | 350 | 86.31 | 295.14 | 1.15 | -1.08 | 181.86 | 0.27 | 0.32 | 0.70 | 0.89 | 0.98 |
| -45 | 10 | 86.37 | 257.38 | 1.04 | -0.40 | 180.00 | 0.46 | 0.09 | 0.73 | 0.90 | 1.00 |
| -45 | 30 | 86.39 | 274.17 | 1.09 | -0.64 | 180.00 | 0.48 | 0.11 | 0.76 | 0.91 | 1.00 |
| -45 | 50 | 86.38 | 292.35 | 1.07 | -0.80 | 180.00 | 0.51 | 0.32 | 0.77 | 0.91 | 1.00 |
| -45 | 70 | 86.26 | 261.11 | 0.98 | -0.35 | 180.00 | 0.42 | 0.15 | 0.71 | 0.92 | 1.00 |
| -45 | 90 | 86.23 | 273.70 | 1.02 | -0.50 | 180.00 | 0.37 | 0.28 | 0.76 | 0.95 | 1.00 |
| -45 | 110 | 86.28 | 304.93 | 1.11 | -0.94 | 180.47 | 0.46 | 0.38 | 0.82 | 0.96 | 1.00 |
| -45 | 130 | 86.34 | 349.68 | 1.40 | -1.42 | 186.53 | 0.45 | 0.22 | 0.86 | 0.95 | 1.00 |
| -45 | 150 | 86.19 | 304.47 | 1.29 | -0.94 | 180.00 | 0.51 | 0.42 | 0.79 | 0.88 | 0.99 |
| -45 | 170 | 86.06 | 269.97 | 1.19 | -0.46 | 180.00 | 0.49 | 0.22 | 0.78 | 0.90 | 1.00 |
| -45 | 190 | 86.11 | 263.44 | 1.08 | -0.34 | 180.00 | 0.59 | 0.08 | 0.72 | 0.86 | 1.00 |
| -45 | 210 | 86.20 | 272.30 | 1.10 | -0.55 | 180.00 | 0.55 | 0.14 | 0.73 | 0.89 | 1.00 |



| | | | | | | | | | | | |
|---|---|---|---|---|---|---|---|---|---|---|---|
| -45 | 230 | 86.24 | 280.69 | 1.01 | -0.70 | 180.00 | 0.44 | 0.27 | 0.73 | 0.91 | 1.00 |
| -45 | 250 | 86.25 | 284.42 | 0.99 | -0.74 | 180.00 | 0.41 | 0.33 | 0.72 | 0.93 | 1.00 |
| -45 | 270 | 86.32 | 272.77 | 1.00 | -0.65 | 180.00 | 0.42 | 0.12 | 0.69 | 0.88 | 1.00 |
| -45 | 290 | 86.21 | 290.48 | 1.22 | -0.92 | 180.00 | 0.45 | 0.30 | 0.75 | 0.87 | 1.00 |
| -45 | 310 | 86.20 | 262.05 | 1.09 | -0.45 | 180.00 | 0.46 | 0.20 | 0.71 | 0.87 | 1.00 |
| -45 | 330 | 86.26 | 252.72 | 0.97 | -0.28 | 180.00 | 0.42 | 0.25 | 0.69 | 0.90 | 1.00 |
| -45 | 350 | 86.30 | 248.99 | 0.95 | -0.15 | 180.00 | 0.44 | 0.20 | 0.70 | 0.92 | 1.00 |
| -35 | 10 | 86.47 | 240.14 | 0.82 | -0.98 | 366.00 | 0.70 | -0.94 | 0.65 | 0.88 | 1.00 |
| -35 | 30 | 86.52 | 259.71 | 0.96 | -0.28 | 180.00 | 0.72 | -0.04 | 0.70 | 0.88 | 1.00 |
| -35 | 50 | 86.55 | 254.12 | 0.95 | -0.13 | 180.00 | 0.72 | -0.04 | 0.67 | 0.87 | 1.00 |
| -35 | 70 | 86.51 | 235.47 | 0.79 | -0.87 | 366.00 | 0.63 | -0.94 | 0.63 | 0.87 | 1.00 |
| -35 | 90 | 86.51 | 240.60 | 0.72 | -0.85 | 366.00 | 0.54 | -0.91 | 0.63 | 0.91 | 1.00 |
| -35 | 110 | 86.61 | 254.59 | 0.85 | -0.23 | 180.00 | 0.53 | -0.03 | 0.67 | 0.90 | 1.00 |
| -35 | 130 | 86.70 | 321.25 | 1.04 | -1.15 | 180.00 | 0.51 | 0.35 | 0.78 | 0.93 | 1.00 |
| -35 | 150 | 86.59 | 271.83 | 1.15 | -0.48 | 180.00 | 0.65 | 0.17 | 0.75 | 0.87 | 1.00 |
| -35 | 170 | 86.33 | 248.53 | 1.05 | 0.01 | 180.00 | 0.73 | 0.04 | 0.71 | 0.88 | 1.00 |
| -35 | 190 | 86.31 | 235.47 | 0.86 | 0.18 | 180.00 | 0.71 | -0.12 | 0.69 | 0.90 | 1.00 |
| -35 | 210 | 86.49 | 243.86 | 0.85 | -0.04 | 180.00 | 0.71 | -0.15 | 0.69 | 0.91 | 1.00 |
| -35 | 230 | 86.55 | 262.05 | 0.78 | -0.35 | 180.00 | 0.61 | -0.05 | 0.63 | 0.89 | 1.00 |
| -35 | 250 | 86.52 | 255.05 | 0.72 | -0.17 | 180.00 | 0.57 | -0.06 | 0.56 | 0.89 | 0.99 |
| -35 | 270 | 86.56 | 270.90 | 0.73 | -0.49 | 180.00 | 0.56 | -0.09 | 0.57 | 0.88 | 0.99 |
| -35 | 290 | 86.33 | 328.24 | 1.12 | -1.20 | 180.00 | 0.63 | 0.47 | 0.78 | 0.90 | 1.00 |
| -35 | 310 | 86.26 | 255.98 | 1.09 | -0.25 | 180.00 | 0.69 | 0.04 | 0.70 | 0.87 | 1.00 |
| -35 | 330 | 86.26 | 216.36 | 0.89 | -0.27 | 366.00 | 0.53 | -0.98 | 0.69 | 0.90 | 1.00 |
| -35 | 350 | 86.35 | 222.89 | 0.79 | -0.43 | 366.00 | 0.51 | -0.96 | 0.61 | 0.88 | 1.00 |
| -25 | 10 | 86.57 | 183.26 | 0.44 | -0.49 | 366.00 | 0.38 | -1.70 | 0.37 | 0.89 | 1.00 |
| -25 | 30 | 86.71 | 366.00 | 0.58 | -1.55 | 196.78 | 0.48 | -0.99 | 0.51 | 0.92 | 1.00 |
| -25 | 50 | 86.74 | 366.00 | 0.60 | -1.60 | 196.32 | 0.52 | -0.99 | 0.46 | 0.86 | 1.00 |
| -25 | 70 | 86.73 | 261.11 | 0.44 | -0.78 | 183.73 | 0.43 | -0.90 | 0.26 | 0.79 | 0.99 |
| -25 | 90 | 86.69 | 264.84 | 0.39 | -1.05 | 186.06 | 0.30 | -1.02 | 0.23 | 0.83 | 0.99 |
| -25 | 110 | 86.69 | 340.36 | 0.44 | -1.59 | 198.65 | 0.32 | -1.19 | 0.28 | 0.85 | 0.99 |
| -25 | 130 | 86.86 | 366.00 | 0.74 | -1.69 | 201.44 | 0.33 | -1.18 | 0.44 | 0.85 | 1.00 |
| -25 | 150 | 86.84 | 366.00 | 0.81 | -1.56 | 187.92 | 0.40 | -0.32 | 0.51 | 0.83 | 0.99 |
| -25 | 170 | 86.62 | 366.00 | 0.42 | -1.78 | 180.00 | 0.41 | 0.05 | 0.36 | 0.87 | 1.00 |
| -25 | 190 | 86.45 | 186.06 | 0.42 | -0.57 | 366.00 | 0.09 | -2.31 | 0.29 | 0.94 | 1.00 |
| -25 | 210 | 86.69 | 184.66 | 0.53 | -0.70 | 366.00 | 0.19 | -1.66 | 0.40 | 0.92 | 1.00 |
| -25 | 230 | 86.84 | 180.00 | 0.55 | -0.59 | 366.00 | 0.39 | -1.60 | 0.44 | 0.89 | 1.00 |





| | | | | | | | | | | | |
|---:|---:|---:|---:|---:|---:|---:|---:|---:|---:|---:|---:|
| -25 | 250 | 86.81 | 180.00 | 0.45 | -0.72 | 366.00 | 0.32 | -1.81 | 0.34 | 0.90 | 1.00 |
| -25 | 270 | 86.77 | 180.00 | 0.40 | -0.92 | 366.00 | 0.34 | -2.02 | 0.30 | 0.88 | 1.00 |
| -25 | 290 | 86.62 | 366.00 | 0.66 | -1.84 | 195.85 | 0.41 | -1.32 | 0.46 | 0.87 | 1.00 |
| -25 | 310 | 86.52 | 366.00 | 0.68 | -1.67 | 197.25 | 0.48 | -1.17 | 0.48 | 0.86 | 1.00 |
| -25 | 330 | 86.49 | 188.39 | 0.46 | -0.61 | 366.00 | 0.38 | -1.47 | 0.32 | 0.84 | 1.00 |
| -25 | 350 | 86.48 | 187.46 | 0.47 | -0.42 | 366.00 | 0.35 | -1.50 | 0.38 | 0.89 | 1.00 |
| -15 | 10 | 86.33 | 187.46 | 0.88 | -1.79 | 338.96 | 0.33 | -2.81 | 0.66 | 0.92 | 1.00 |
| -15 | 30 | 86.40 | 180.00 | 0.89 | -1.50 | 338.03 | 0.29 | -2.47 | 0.63 | 0.93 | 1.00 |
| -15 | 50 | 86.43 | 180.00 | 0.81 | -1.50 | 366.00 | 0.33 | -2.17 | 0.59 | 0.91 | 1.00 |
| -15 | 70 | 86.35 | 180.00 | 0.56 | -1.56 | 366.00 | 0.35 | -2.02 | 0.37 | 0.84 | 1.00 |
| -15 | 90 | 86.21 | 180.00 | 0.53 | -1.45 | 366.00 | 0.39 | -2.02 | 0.37 | 0.83 | 1.00 |
| -15 | 110 | 86.23 | 180.00 | 0.51 | -1.59 | 340.36 | 0.34 | -2.07 | 0.38 | 0.87 | 1.00 |
| -15 | 130 | 86.46 | 366.00 | 0.73 | -1.73 | 193.05 | 0.39 | -1.77 | 0.42 | 0.82 | 1.00 |
| -15 | 150 | 86.55 | 366.00 | 0.77 | -1.62 | 198.65 | 0.49 | -1.65 | 0.47 | 0.84 | 1.00 |
| -15 | 170 | 86.40 | 184.20 | 0.42 | -1.65 | 366.00 | 0.25 | -2.10 | 0.23 | 0.83 | 1.00 |
| -15 | 190 | 86.29 | 195.85 | 0.67 | -2.15 | 332.44 | 0.15 | -3.07 | 0.42 | 0.87 | 1.00 |
| -15 | 210 | 86.48 | 190.72 | 0.78 | -1.86 | 280.23 | 0.15 | -1.62 | 0.49 | 0.89 | 1.00 |
| -15 | 230 | 86.52 | 180.00 | 0.79 | -1.42 | 366.00 | 0.30 | -1.90 | 0.55 | 0.89 | 1.00 |
| -15 | 250 | 86.42 | 180.47 | 0.95 | -1.66 | 355.74 | 0.25 | -2.56 | 0.64 | 0.92 | 1.00 |
| -15 | 270 | 86.47 | 180.00 | 0.87 | -1.79 | 334.30 | 0.37 | -2.69 | 0.61 | 0.92 | 1.00 |
| -15 | 290 | 86.47 | 180.00 | 0.75 | -1.70 | 348.75 | 0.41 | -2.71 | 0.60 | 0.93 | 1.00 |
| -15 | 310 | 86.29 | 180.00 | 0.81 | -1.72 | 343.16 | 0.34 | -2.24 | 0.59 | 0.90 | 1.00 |
| -15 | 330 | 86.27 | 180.00 | 0.69 | -1.59 | 340.36 | 0.31 | -2.11 | 0.52 | 0.90 | 1.00 |
| -15 | 350 | 86.31 | 180.93 | 0.66 | -1.58 | 339.43 | 0.25 | -2.60 | 0.52 | 0.93 | 1.00 |
| -5 | 10 | 85.59 | 189.32 | 1.31 | -1.83 | 351.55 | 0.48 | -3.09 | 0.78 | 0.89 | 1.00 |
| -5 | 30 | 85.57 | 184.66 | 1.42 | -1.69 | 328.24 | 0.37 | -2.91 | 0.81 | 0.90 | 1.00 |
| -5 | 50 | 85.46 | 180.00 | 1.34 | -1.65 | 311.46 | 0.35 | -2.45 | 0.78 | 0.91 | 1.00 |
| -5 | 70 | 85.35 | 180.00 | 1.15 | -1.65 | 313.79 | 0.36 | -2.17 | 0.67 | 0.83 | 1.00 |
| -5 | 90 | 85.32 | 180.00 | 1.07 | -1.54 | 317.52 | 0.33 | -1.93 | 0.64 | 0.85 | 1.00 |
| -5 | 110 | 85.59 | 180.00 | 1.08 | -1.58 | 285.35 | 0.41 | -1.98 | 0.63 | 0.83 | 1.00 |
| -5 | 130 | 85.94 | 180.00 | 0.98 | -1.56 | 309.13 | 0.46 | -1.50 | 0.60 | 0.84 | 1.00 |
| -5 | 150 | 85.99 | 180.00 | 0.82 | -1.42 | 366.00 | 0.52 | -1.75 | 0.56 | 0.82 | 1.00 |
| -5 | 170 | 86.06 | 185.59 | 0.81 | -1.64 | 333.37 | 0.34 | -2.26 | 0.51 | 0.86 | 1.00 |
| -5 | 190 | 86.11 | 184.20 | 0.72 | -1.80 | 320.32 | 0.35 | -2.70 | 0.50 | 0.89 | 1.00 |
| -5 | 210 | 86.00 | 180.00 | 0.87 | -1.62 | 314.72 | 0.29 | -1.80 | 0.55 | 0.88 | 1.00 |
| -5 | 230 | 85.77 | 180.00 | 1.17 | -1.53 | 366.00 | 0.46 | -2.01 | 0.70 | 0.86 | 1.00 |
| -5 | 250 | 85.55 | 180.00 | 1.38 | -1.63 | 366.00 | 0.29 | -1.99 | 0.81 | 0.93 | 1.00 |





| | | | | | | | | | | | |
|---|---|---|---|---|---|---|---|---|---|---|---|
| -5 | 270 | 85.67 | 180.00 | 1.18 | -1.62 | 315.65 | 0.35 | -2.24 | 0.78 | 0.95 | 1.00 |
| -5 | 290 | 85.73 | 180.00 | 1.15 | -1.61 | 340.36 | 0.48 | -2.63 | 0.79 | 0.95 | 1.00 |
| -5 | 310 | 85.54 | 180.00 | 1.17 | -1.68 | 341.76 | 0.41 | -2.53 | 0.76 | 0.91 | 1.00 |
| -5 | 330 | 85.49 | 180.00 | 1.12 | -1.57 | 360.41 | 0.41 | -2.51 | 0.76 | 0.93 | 1.00 |
| -5 | 350 | 85.53 | 181.86 | 1.21 | -1.61 | 335.70 | 0.49 | -2.87 | 0.76 | 0.91 | 1.00 |
| 5 | 10 | 85.54 | 201.44 | 1.37 | -2.19 | 366.00 | 0.10 | 2.03 | 0.74 | 0.87 | 0.99 |
| 5 | 30 | 85.37 | 204.24 | 1.46 | -2.21 | 366.00 | 0.10 | 1.30 | 0.74 | 0.87 | 0.99 |
| 5 | 50 | 85.14 | 198.65 | 1.47 | -1.98 | 366.00 | 0.14 | 0.28 | 0.72 | 0.83 | 0.98 |
| 5 | 70 | 85.07 | 194.45 | 1.38 | -1.84 | 366.00 | 0.20 | -0.33 | 0.61 | 0.75 | 0.96 |
| 5 | 90 | 85.40 | 194.45 | 1.25 | -1.79 | 366.00 | 0.14 | -0.91 | 0.61 | 0.78 | 0.99 |
| 5 | 110 | 85.86 | 196.32 | 1.14 | -1.82 | 298.87 | 0.13 | -1.58 | 0.57 | 0.83 | 0.98 |
| 5 | 130 | 86.11 | 193.98 | 0.98 | -1.70 | 366.00 | 0.15 | -0.55 | 0.50 | 0.80 | 0.99 |
| 5 | 150 | 85.95 | 187.46 | 1.04 | -1.62 | 366.00 | 0.24 | -0.81 | 0.55 | 0.81 | 0.99 |
| 5 | 170 | 86.26 | 183.73 | 0.94 | -1.35 | 366.00 | 0.31 | -1.95 | 0.58 | 0.85 | 1.00 |
| 5 | 190 | 86.48 | 180.47 | 0.67 | -1.43 | 366.00 | 0.27 | -2.48 | 0.49 | 0.89 | 1.00 |
| 5 | 210 | 86.26 | 181.86 | 0.83 | -1.62 | 366.00 | 0.15 | -1.71 | 0.56 | 0.90 | 1.00 |
| 5 | 230 | 85.86 | 182.80 | 1.21 | -1.71 | 366.00 | 0.16 | -1.41 | 0.73 | 0.91 | 1.00 |
| 5 | 250 | 85.63 | 181.40 | 1.37 | -1.68 | 366.00 | 0.26 | -0.53 | 0.81 | 0.94 | 1.00 |
| 5 | 270 | 85.81 | 180.00 | 1.18 | -1.59 | 366.00 | 0.16 | -1.19 | 0.74 | 0.92 | 1.00 |
| 5 | 290 | 85.82 | 186.06 | 1.17 | -1.74 | 366.00 | 0.15 | -2.76 | 0.74 | 0.91 | 1.00 |
| 5 | 310 | 85.55 | 189.32 | 1.21 | -1.93 | 317.05 | 0.11 | -2.62 | 0.73 | 0.91 | 1.00 |
| 5 | 330 | 85.40 | 184.66 | 1.21 | -1.65 | 366.00 | 0.08 | -1.90 | 0.74 | 0.90 | 1.00 |
| 5 | 350 | 85.44 | 192.59 | 1.29 | -1.96 | 312.86 | 0.08 | -2.49 | 0.76 | 0.92 | 1.00 |
| 15 | 10 | 86.42 | 203.77 | 0.78 | -2.24 | 366.00 | 0.15 | 0.76 | 0.55 | 0.91 | 1.00 |
| 15 | 30 | 86.36 | 203.31 | 0.79 | -2.03 | 366.00 | 0.18 | 0.60 | 0.51 | 0.87 | 1.00 |
| 15 | 50 | 86.13 | 200.51 | 0.89 | -1.81 | 366.00 | 0.31 | 0.21 | 0.50 | 0.82 | 0.99 |
| 15 | 70 | 86.06 | 193.98 | 0.77 | -1.57 | 366.00 | 0.34 | -0.15 | 0.47 | 0.85 | 0.99 |
| 15 | 90 | 86.37 | 184.66 | 0.77 | -1.26 | 366.00 | 0.20 | -0.44 | 0.52 | 0.89 | 1.00 |
| 15 | 110 | 86.66 | 180.00 | 0.67 | -0.77 | 286.75 | 0.15 | 1.32 | 0.39 | 0.86 | 1.00 |
| 15 | 130 | 86.67 | 184.20 | 0.51 | -0.80 | 330.57 | 0.15 | 1.00 | 0.28 | 0.88 | 1.00 |
| 15 | 150 | 86.45 | 180.00 | 0.67 | -1.21 | 366.00 | 0.28 | -0.71 | 0.54 | 0.92 | 1.00 |
| 15 | 170 | 86.64 | 180.00 | 0.76 | -1.17 | 366.00 | 0.39 | -1.52 | 0.62 | 0.94 | 1.00 |
| 15 | 190 | 86.91 | 180.00 | 0.56 | -1.14 | 366.00 | 0.31 | -1.74 | 0.46 | 0.91 | 1.00 |
| 15 | 210 | 86.86 | 180.00 | 0.58 | -0.99 | 267.17 | 0.19 | 0.99 | 0.40 | 0.92 | 1.00 |
| 15 | 230 | 86.55 | 186.53 | 0.79 | -1.55 | 314.26 | 0.19 | 1.10 | 0.60 | 0.93 | 1.00 |
| 15 | 250 | 86.41 | 185.59 | 0.90 | -1.69 | 366.00 | 0.28 | 0.27 | 0.68 | 0.94 | 1.00 |
| 15 | 270 | 86.53 | 180.00 | 0.72 | -1.45 | 361.34 | 0.24 | -0.45 | 0.61 | 0.95 | 1.00 |



| 15 | 290 | 86.53 | 180.00 | 0.82 | -1.16 | 249.92 | 0.13 | 0.98 | 0.58 | 0.95 | 1.00 |
|----|-----|-------|--------|------|-------|--------|------|------|------|------|------|
| 15 | 310 | 86.32 | 181.40 | 0.75 | -1.24 | 257.85 | 0.06 | 1.02 | 0.52 | 0.93 | 1.00 |
| 15 | 330 | 86.16 | 184.66 | 0.82 | -1.50 | 366.00 | 0.16 | -0.66 | 0.56 | 0.91 | 1.00 |
| 15 | 350 | 86.34 | 192.12 | 0.78 | -1.82 | 332.44 | 0.12 | 0.55 | 0.62 | 0.95 | 1.00 |
| 25 | 10  | 86.83 | 180.00 | 0.45 | -0.36 | 366.00 | 0.30 | 1.26 | 0.31 | 0.86 | 1.00 |
| 25 | 30  | 86.79 | 180.00 | 0.51 | -0.20 | 326.38 | 0.20 | 1.08 | 0.29 | 0.86 | 1.00 |
| 25 | 50  | 86.70 | 180.00 | 0.48 | -0.19 | 358.08 | 0.17 | 0.70 | 0.29 | 0.89 | 1.00 |
| 25 | 70  | 86.60 | 180.00 | 0.43 | -0.28 | 366.00 | 0.23 | -0.04 | 0.35 | 0.95 | 1.00 |
| 25 | 90  | 86.75 | 180.00 | 0.52 | -0.23 | 336.63 | 0.16 | 0.13 | 0.35 | 0.91 | 1.00 |
| 25 | 110 | 86.84 | 180.00 | 0.67 | -0.09 | 259.71 | 0.26 | 0.40 | 0.39 | 0.89 | 1.00 |
| 25 | 130 | 86.81 | 180.00 | 0.52 | -0.02 | 283.02 | 0.28 | -0.28 | 0.35 | 0.89 | 1.00 |
| 25 | 150 | 86.63 | 366.00 | 0.48 | -1.27 | 180.00 | 0.44 | -0.51 | 0.49 | 0.92 | 1.00 |
| 25 | 170 | 86.71 | 180.00 | 0.56 | -0.59 | 356.68 | 0.39 | -1.21 | 0.52 | 0.94 | 1.00 |
| 25 | 190 | 86.96 | 180.00 | 0.61 | -0.49 | 298.41 | 0.22 | -0.46 | 0.42 | 0.91 | 1.00 |
| 25 | 210 | 87.04 | 180.00 | 0.57 | -0.31 | 297.94 | 0.10 | 1.10 | 0.40 | 0.93 | 1.00 |
| 25 | 230 | 86.95 | 180.00 | 0.60 | -0.58 | 355.74 | 0.35 | 1.17 | 0.56 | 0.95 | 1.00 |
| 25 | 250 | 86.83 | 189.32 | 0.56 | -1.05 | 366.00 | 0.37 | 0.60 | 0.54 | 0.95 | 1.00 |
| 25 | 270 | 86.79 | 180.00 | 0.52 | -0.35 | 366.00 | 0.18 | 0.12 | 0.42 | 0.93 | 1.00 |
| 25 | 290 | 86.76 | 180.00 | 0.63 | -0.25 | 338.96 | 0.13 | -0.94 | 0.44 | 0.91 | 1.00 |
| 25 | 310 | 86.71 | 180.00 | 0.52 | -0.18 | 358.54 | 0.20 | -1.49 | 0.43 | 0.96 | 1.00 |
| 25 | 330 | 86.68 | 180.00 | 0.46 | -0.43 | 366.00 | 0.14 | -0.94 | 0.35 | 0.94 | 1.00 |
| 25 | 350 | 86.77 | 180.00 | 0.51 | -0.57 | 352.02 | 0.25 | 1.25 | 0.34 | 0.89 | 1.00 |
| 35 | 10  | 86.58 | 180.00 | 0.68 | 0.57 | 366.00 | 0.63 | 1.32 | 0.65 | 0.93 | 1.00 |
| 35 | 30  | 86.56 | 180.00 | 0.72 | 0.69 | 366.00 | 0.48 | 1.29 | 0.63 | 0.94 | 1.00 |
| 35 | 50  | 86.54 | 180.00 | 0.66 | 0.62 | 366.00 | 0.40 | 1.23 | 0.55 | 0.93 | 1.00 |
| 35 | 70  | 86.44 | 180.00 | 0.63 | 0.58 | 366.00 | 0.31 | 0.98 | 0.50 | 0.92 | 1.00 |
| 35 | 90  | 86.49 | 180.00 | 0.73 | 0.57 | 366.00 | 0.29 | 1.17 | 0.55 | 0.91 | 1.00 |
| 35 | 110 | 86.52 | 180.00 | 0.77 | 0.63 | 366.00 | 0.13 | 1.16 | 0.55 | 0.91 | 1.00 |
| 35 | 130 | 86.57 | 180.00 | 0.71 | 0.59 | 291.41 | 0.22 | 0.39 | 0.50 | 0.94 | 1.00 |
| 35 | 150 | 86.50 | 180.00 | 0.59 | 0.47 | 317.05 | 0.27 | -0.18 | 0.41 | 0.93 | 1.00 |
| 35 | 170 | 86.58 | 180.00 | 0.62 | 0.37 | 312.86 | 0.23 | 0.37 | 0.41 | 0.92 | 1.00 |
| 35 | 190 | 86.72 | 180.00 | 0.56 | 0.39 | 335.70 | 0.21 | 1.14 | 0.37 | 0.90 | 1.00 |
| 35 | 210 | 86.83 | 180.00 | 0.60 | 0.38 | 366.00 | 0.39 | 1.14 | 0.50 | 0.92 | 1.00 |
| 35 | 230 | 86.82 | 180.00 | 0.60 | 0.27 | 366.00 | 0.51 | 1.17 | 0.58 | 0.95 | 1.00 |
| 35 | 250 | 86.72 | 366.00 | 0.55 | 1.04 | 180.00 | 0.56 | 0.14 | 0.62 | 0.94 | 1.00 |
| 35 | 270 | 86.63 | 180.00 | 0.58 | 0.23 | 366.00 | 0.40 | 0.89 | 0.51 | 0.94 | 1.00 |
| 35 | 290 | 86.52 | 180.00 | 0.72 | 0.49 | 366.00 | 0.21 | 0.39 | 0.54 | 0.92 | 1.00 |





| | | | | | | | | | | | |
|---|---|---|---|---|---|---|---|---|---|---|---|
| 35 | 310 | 86.45 | 180.00 | 0.73 | 0.55 | 366.00 | 0.25 | 0.67 | 0.60 | 0.94 | 1.00 |
| 35 | 330 | 86.53 | 180.00 | 0.64 | 0.54 | 366.00 | 0.37 | 1.03 | 0.59 | 0.95 | 1.00 |
| 35 | 350 | 86.64 | 180.00 | 0.62 | 0.46 | 366.00 | 0.59 | 1.31 | 0.62 | 0.93 | 1.00 |
| 45 | 10 | 86.34 | 366.00 | 0.97 | 1.35 | 180.00 | 0.72 | 0.80 | 0.78 | 0.94 | 1.00 |
| 45 | 30 | 86.36 | 366.00 | 0.86 | 1.33 | 180.93 | 0.72 | 0.77 | 0.78 | 0.96 | 1.00 |
| 45 | 50 | 86.37 | 366.00 | 0.79 | 1.34 | 182.33 | 0.66 | 0.82 | 0.74 | 0.96 | 1.00 |
| 45 | 70 | 86.35 | 180.00 | 0.67 | 0.91 | 365.07 | 0.65 | 1.35 | 0.74 | 0.96 | 1.00 |
| 45 | 90 | 86.35 | 180.00 | 0.69 | 0.79 | 366.00 | 0.65 | 1.40 | 0.72 | 0.96 | 1.00 |
| 45 | 110 | 86.30 | 180.00 | 0.71 | 0.86 | 364.14 | 0.61 | 1.44 | 0.67 | 0.94 | 1.00 |
| 45 | 130 | 86.29 | 180.00 | 0.59 | 0.89 | 361.34 | 0.55 | 1.27 | 0.63 | 0.96 | 1.00 |
| 45 | 150 | 86.25 | 366.00 | 0.55 | 1.21 | 180.00 | 0.53 | 1.00 | 0.57 | 0.93 | 1.00 |
| 45 | 170 | 86.24 | 366.00 | 0.65 | 1.31 | 180.00 | 0.59 | 0.97 | 0.65 | 0.91 | 1.00 |
| 45 | 190 | 86.27 | 366.00 | 0.88 | 1.37 | 180.00 | 0.67 | 0.99 | 0.73 | 0.90 | 1.00 |
| 45 | 210 | 86.33 | 366.00 | 1.08 | 1.39 | 180.47 | 0.71 | 0.88 | 0.78 | 0.92 | 1.00 |
| 45 | 230 | 86.35 | 366.00 | 1.12 | 1.38 | 181.86 | 0.72 | 0.82 | 0.80 | 0.92 | 1.00 |
| 45 | 250 | 86.31 | 366.00 | 0.96 | 1.30 | 180.47 | 0.69 | 0.81 | 0.80 | 0.94 | 1.00 |
| 45 | 270 | 86.27 | 366.00 | 0.73 | 1.21 | 180.00 | 0.70 | 0.89 | 0.74 | 0.96 | 1.00 |
| 45 | 290 | 86.24 | 180.00 | 0.74 | 0.85 | 358.08 | 0.58 | 1.15 | 0.70 | 0.95 | 1.00 |
| 45 | 310 | 86.14 | 180.00 | 0.69 | 0.88 | 354.35 | 0.65 | 1.21 | 0.66 | 0.95 | 1.00 |
| 45 | 330 | 86.24 | 366.00 | 0.83 | 1.23 | 180.00 | 0.63 | 0.78 | 0.75 | 0.95 | 1.00 |
| 45 | 350 | 86.35 | 366.00 | 0.90 | 1.32 | 180.00 | 0.69 | 0.73 | 0.80 | 0.96 | 1.00 |
| 55 | 10 | 86.38 | 366.00 | 1.20 | 1.34 | 180.00 | 0.48 | 0.98 | 0.76 | 0.91 | 1.00 |
| 55 | 30 | 86.40 | 366.00 | 1.11 | 1.33 | 180.00 | 0.46 | 0.96 | 0.73 | 0.89 | 1.00 |
| 55 | 50 | 86.44 | 366.00 | 1.00 | 1.34 | 182.80 | 0.44 | 0.99 | 0.72 | 0.91 | 1.00 |
| 55 | 70 | 86.43 | 366.00 | 0.88 | 1.34 | 181.40 | 0.45 | 1.08 | 0.69 | 0.93 | 0.99 |
| 55 | 90 | 86.41 | 366.00 | 0.92 | 1.40 | 180.00 | 0.40 | 0.98 | 0.69 | 0.92 | 0.99 |
| 55 | 110 | 86.28 | 366.00 | 0.91 | 1.35 | 180.00 | 0.42 | 1.06 | 0.68 | 0.93 | 1.00 |
| 55 | 130 | 86.18 | 366.00 | 0.96 | 1.40 | 180.00 | 0.36 | 1.25 | 0.68 | 0.89 | 1.00 |
| 55 | 150 | 86.10 | 366.00 | 1.08 | 1.39 | 180.00 | 0.35 | 1.27 | 0.70 | 0.89 | 0.99 |
| 55 | 170 | 86.04 | 366.00 | 1.31 | 1.43 | 180.00 | 0.36 | 1.32 | 0.75 | 0.88 | 0.99 |
| 55 | 190 | 86.02 | 366.00 | 1.53 | 1.39 | 180.00 | 0.46 | 1.34 | 0.80 | 0.85 | 0.99 |
| 55 | 210 | 85.97 | 366.00 | 1.74 | 1.43 | 180.00 | 0.50 | 1.27 | 0.83 | 0.88 | 1.00 |
| 55 | 230 | 85.99 | 366.00 | 1.70 | 1.42 | 180.00 | 0.55 | 1.11 | 0.84 | 0.87 | 1.00 |
| 55 | 250 | 86.04 | 366.00 | 1.45 | 1.43 | 180.00 | 0.53 | 1.25 | 0.81 | 0.89 | 0.99 |
| 55 | 270 | 86.08 | 366.00 | 1.21 | 1.38 | 180.00 | 0.54 | 1.30 | 0.73 | 0.88 | 0.99 |
| 55 | 290 | 86.15 | 366.00 | 1.10 | 1.26 | 180.00 | 0.47 | 1.33 | 0.70 | 0.89 | 0.99 |
| 55 | 310 | 86.09 | 366.00 | 1.15 | 1.26 | 180.00 | 0.36 | 1.16 | 0.70 | 0.89 | 0.99 |





| 55 | 330 | 86.16 | 366.00 | 1.18 | 1.28 | 180.00 | 0.35 | 1.03 | 0.74 | 0.89 | 0.99 |
| 55 | 350 | 86.27 | 366.00 | 1.15 | 1.32 | 180.00 | 0.45 | 0.83 | 0.74 | 0.90 | 1.00 |





**Table 3** Same as table 1 but for the FWHM. Please pay attention to the fact that FWHM is more variable during the year and therefore the fraction of data around the approximation is provided for intervals of +/- 7.5 % and +/- 15 %.

| Lat. | Lon. | Mean | 1st oscillation | | | 2nd oscillation | | | Quality of | Fraction of data | |
| | | | Period | Amp. | Phase | Period | Amp. | Phase | approxi- | +-7.5% | +/- 15% |
| [°] | [°] | [$10^{-2}$ s$^{-1}$] | [d] | [$10^{-2}$ s$^{-1}$] | [rad] | [d] | [$10^{-2}$ s$^{-1}$] | [rad] | mation | | |
| | | | | | | | | | | around approx. | |
| -55 | 10 | 7.50 | 366.00 | 0.74 | 1.78 | 191.65 | 0.60 | 1.22 | 0.51 | 0.68 | 0.97 |
| -55 | 30 | 7.48 | 366.00 | 0.82 | 1.63 | 191.65 | 0.60 | 1.23 | 0.60 | 0.71 | 0.96 |
| -55 | 50 | 7.54 | 366.00 | 0.68 | 1.58 | 189.32 | 0.47 | 1.37 | 0.48 | 0.70 | 0.94 |
| -55 | 70 | 7.57 | 366.00 | 0.67 | 1.61 | 193.98 | 0.38 | 1.20 | 0.32 | 0.68 | 0.96 |
| -55 | 90 | 7.53 | 366.00 | 0.67 | 1.69 | 194.45 | 0.36 | 1.25 | 0.36 | 0.63 | 0.95 |
| -55 | 110 | 7.51 | 366.00 | 0.79 | 1.67 | 193.52 | 0.48 | 1.23 | 0.55 | 0.69 | 0.96 |
| -55 | 130 | 7.58 | 366.00 | 0.77 | 1.69 | 193.98 | 0.45 | 1.13 | 0.49 | 0.66 | 0.95 |
| -55 | 150 | 7.65 | 366.00 | 0.72 | 1.60 | 193.52 | 0.46 | 1.16 | 0.46 | 0.67 | 0.94 |
| -55 | 170 | 7.69 | 366.00 | 0.75 | 1.61 | 191.65 | 0.51 | 1.15 | 0.53 | 0.68 | 0.98 |
| -55 | 190 | 7.66 | 366.00 | 0.78 | 1.58 | 191.19 | 0.60 | 1.23 | 0.58 | 0.72 | 0.97 |
| -55 | 210 | 7.68 | 366.00 | 0.85 | 1.63 | 190.72 | 0.61 | 1.08 | 0.62 | 0.73 | 0.97 |
| -55 | 230 | 7.76 | 366.00 | 0.80 | 1.68 | 189.32 | 0.48 | 1.22 | 0.65 | 0.80 | 0.98 |
| -55 | 250 | 7.71 | 366.00 | 0.94 | 1.73 | 189.79 | 0.49 | 1.37 | 0.66 | 0.76 | 0.96 |
| -55 | 270 | 7.70 | 366.00 | 0.97 | 1.73 | 188.86 | 0.53 | 1.19 | 0.64 | 0.67 | 0.95 |
| -55 | 290 | 7.75 | 366.00 | 0.95 | 1.70 | 188.39 | 0.53 | 1.03 | 0.66 | 0.71 | 0.96 |
| -55 | 310 | 7.66 | 366.00 | 0.87 | 1.67 | 193.52 | 0.59 | 0.73 | 0.65 | 0.77 | 0.96 |
| -55 | 330 | 7.59 | 366.00 | 0.76 | 1.72 | 188.86 | 0.51 | 0.96 | 0.61 | 0.76 | 0.98 |
| -55 | 350 | 7.57 | 366.00 | 0.68 | 1.93 | 189.32 | 0.40 | 1.26 | 0.46 | 0.69 | 0.94 |
| -45 | 10 | 7.55 | 366.00 | 0.43 | 1.69 | 190.72 | 0.33 | 0.84 | 0.28 | 0.64 | 0.95 |
| -45 | 30 | 7.57 | 366.00 | 0.51 | 1.55 | 193.52 | 0.33 | 1.05 | 0.36 | 0.72 | 0.98 |
| -45 | 50 | 7.57 | 366.00 | 0.50 | 1.52 | 199.58 | 0.29 | 1.09 | 0.32 | 0.74 | 0.98 |
| -45 | 70 | 7.66 | 366.00 | 0.45 | 1.54 | 199.11 | 0.24 | 0.80 | 0.25 | 0.68 | 0.96 |
| -45 | 90 | 7.61 | 366.00 | 0.51 | 1.75 | 200.51 | 0.18 | 0.68 | 0.29 | 0.71 | 0.96 |
| -45 | 110 | 7.58 | 366.00 | 0.62 | 1.76 | 203.31 | 0.24 | 0.92 | 0.39 | 0.75 | 0.98 |
| -45 | 130 | 7.55 | 366.00 | 0.63 | 1.75 | 199.58 | 0.28 | 0.93 | 0.45 | 0.75 | 0.98 |
| -45 | 150 | 7.72 | 366.00 | 0.47 | 1.85 | 204.24 | 0.16 | 1.10 | 0.29 | 0.73 | 0.99 |
| -45 | 170 | 7.76 | 366.00 | 0.44 | 1.77 | 201.91 | 0.16 | 0.82 | 0.27 | 0.80 | 0.97 |
| -45 | 190 | 7.67 | 366.00 | 0.55 | 1.84 | 196.78 | 0.24 | 0.88 | 0.40 | 0.80 | 0.98 |
| -45 | 210 | 7.69 | 366.00 | 0.61 | 1.74 | 196.32 | 0.30 | 0.88 | 0.47 | 0.79 | 0.99 |
| -45 | 230 | 7.71 | 366.00 | 0.52 | 1.79 | 198.65 | 0.22 | 0.67 | 0.37 | 0.80 | 0.98 |
| -45 | 250 | 7.66 | 366.00 | 0.58 | 1.92 | 199.11 | 0.22 | 0.64 | 0.38 | 0.74 | 0.97 |


| | | | | | | | | | | | |
|---|---|---|---|---|---|---|---|---|---|---|---|
| -45 | 270 | 7.66 | 366.00 | 0.55 | 1.86 | 192.12 | 0.37 | 0.77 | 0.39 | 0.70 | 0.96 |
| -45 | 290 | 7.76 | 366.00 | 0.62 | 1.82 | 180.00 | 0.32 | 0.82 | 0.51 | 0.76 | 0.99 |
| -45 | 310 | 7.76 | 366.00 | 0.46 | 1.69 | 187.46 | 0.34 | 0.86 | 0.42 | 0.79 | 0.98 |
| -45 | 330 | 7.65 | 366.00 | 0.45 | 1.69 | 188.86 | 0.36 | 0.60 | 0.40 | 0.77 | 0.99 |
| -45 | 350 | 7.58 | 366.00 | 0.41 | 1.79 | 193.52 | 0.26 | 0.70 | 0.26 | 0.72 | 0.96 |
| -35 | 10 | 7.39 | 366.00 | 0.25 | 2.07 | 180.00 | 0.21 | -0.38 | 0.10 | 0.60 | 0.90 |
| -35 | 30 | 7.41 | 366.00 | 0.28 | 1.90 | 180.00 | 0.21 | -0.94 | 0.13 | 0.63 | 0.92 |
| -35 | 50 | 7.43 | 366.00 | 0.27 | 2.05 | 180.00 | 0.26 | -1.33 | 0.15 | 0.69 | 0.90 |
| -35 | 70 | 7.47 | 180.00 | 0.24 | -1.09 | 366.00 | 0.20 | 1.97 | 0.10 | 0.68 | 0.91 |
| -35 | 90 | 7.48 | 366.00 | 0.21 | 1.84 | 180.00 | 0.15 | -0.67 | 0.07 | 0.62 | 0.92 |
| -35 | 110 | 7.40 | 366.00 | 0.31 | 1.72 | 180.00 | 0.12 | -0.76 | 0.13 | 0.66 | 0.93 |
| -35 | 130 | 7.33 | 366.00 | 0.32 | 1.83 | 180.00 | 0.16 | -0.17 | 0.16 | 0.69 | 0.94 |
| -35 | 150 | 7.42 | 331.04 | 0.27 | 2.54 | 180.00 | 0.17 | -1.31 | 0.14 | 0.69 | 0.95 |
| -35 | 170 | 7.54 | 180.00 | 0.25 | -1.01 | 366.00 | 0.21 | 2.79 | 0.15 | 0.74 | 0.94 |
| -35 | 190 | 7.44 | 366.00 | 0.37 | 2.33 | 180.00 | 0.17 | -1.05 | 0.24 | 0.76 | 0.96 |
| -35 | 210 | 7.30 | 366.00 | 0.42 | 2.13 | 180.00 | 0.03 | -0.63 | 0.29 | 0.78 | 0.97 |
| -35 | 230 | 7.36 | 351.08 | 0.18 | 2.24 | 180.00 | 0.10 | -1.13 | 0.08 | 0.76 | 0.97 |
| -35 | 250 | 7.38 | 322.18 | 0.19 | -2.90 | 180.00 | 0.15 | -0.71 | 0.09 | 0.73 | 0.95 |
| -35 | 270 | 7.49 | 180.00 | 0.29 | 0.30 | 366.00 | 0.24 | 3.06 | 0.16 | 0.66 | 0.93 |
| -35 | 290 | 7.69 | 359.94 | 0.43 | 2.11 | 180.00 | 0.27 | 0.06 | 0.33 | 0.74 | 0.96 |
| -35 | 310 | 7.73 | 366.00 | 0.32 | 1.85 | 180.00 | 0.17 | -0.31 | 0.18 | 0.72 | 0.95 |
| -35 | 330 | 7.67 | 180.00 | 0.29 | -0.60 | 366.00 | 0.15 | 1.96 | 0.12 | 0.69 | 0.93 |
| -35 | 350 | 7.45 | 180.00 | 0.26 | -0.23 | 366.00 | 0.15 | 1.37 | 0.08 | 0.60 | 0.89 |
| -25 | 10 | 7.48 | 180.00 | 0.15 | 0.02 | 366.00 | 0.11 | 2.18 | 0.05 | 0.68 | 0.96 |
| -25 | 30 | 7.53 | 366.00 | 0.24 | 2.21 | 180.00 | 0.11 | -1.54 | 0.11 | 0.71 | 0.96 |
| -25 | 50 | 7.48 | 366.00 | 0.19 | 2.41 | 180.00 | 0.13 | -1.55 | 0.07 | 0.67 | 0.97 |
| -25 | 70 | 7.49 | 180.00 | 0.18 | -1.13 | 366.00 | 0.15 | 2.38 | 0.07 | 0.66 | 0.94 |
| -25 | 90 | 7.59 | 366.00 | 0.15 | 1.61 | 180.00 | 0.10 | -1.37 | 0.04 | 0.65 | 0.95 |
| -25 | 110 | 7.50 | 363.67 | 0.25 | 1.43 | 239.67 | 0.10 | 0.96 | 0.09 | 0.69 | 0.96 |
| -25 | 130 | 7.34 | 365.53 | 0.35 | 1.49 | 180.00 | 0.12 | -1.49 | 0.17 | 0.61 | 0.95 |
| -25 | 150 | 7.46 | 180.00 | 0.16 | -1.25 | 366.00 | 0.14 | 1.70 | 0.05 | 0.65 | 0.93 |
| -25 | 170 | 7.55 | 180.00 | 0.17 | -0.51 | 366.00 | 0.13 | -2.77 | 0.07 | 0.71 | 0.96 |
| -25 | 190 | 7.47 | 366.00 | 0.22 | 2.75 | 180.00 | 0.11 | 0.18 | 0.12 | 0.78 | 0.98 |
| -25 | 210 | 7.20 | 366.00 | 0.26 | 2.27 | 180.00 | 0.09 | 0.73 | 0.18 | 0.81 | 0.98 |
| -25 | 230 | 7.17 | 238.74 | 0.12 | -1.71 | 180.00 | 0.07 | 0.88 | 0.05 | 0.72 | 0.97 |
| -25 | 250 | 7.33 | 263.91 | 0.21 | -0.48 | 180.00 | 0.20 | -0.30 | 0.13 | 0.72 | 0.96 |
| -25 | 270 | 7.51 | 219.62 | 0.27 | 0.05 | 366.00 | 0.12 | -3.00 | 0.14 | 0.68 | 0.96 |





| | | | | | | | | | | | |
|---|---|---|---|---|---|---|---|---|---|---|---|
| -25 | 290 | 7.69 | 359.94 | 0.21 | 2.87 | 180.00 | 0.17 | 0.58 | 0.13 | 0.75 | 0.98 |
| -25 | 310 | 7.68 | 366.00 | 0.30 | 2.34 | 236.41 | 0.09 | 0.81 | 0.12 | 0.75 | 0.98 |
| -25 | 330 | 7.59 | 180.00 | 0.25 | -0.52 | 225.22 | 0.19 | 1.56 | 0.04 | 0.69 | 0.95 |
| -25 | 350 | 7.47 | 180.00 | 0.37 | -0.14 | 244.80 | 0.27 | 1.19 | 0.10 | 0.67 | 0.95 |
| -15 | 10 | 7.73 | 180.00 | 0.19 | 1.94 | 365.53 | 0.18 | 1.34 | 0.18 | 0.85 | 1.00 |
| -15 | 30 | 7.75 | 366.00 | 0.21 | 1.69 | 186.53 | 0.20 | 1.85 | 0.18 | 0.84 | 0.99 |
| -15 | 50 | 7.74 | 180.00 | 0.13 | 2.50 | 346.89 | 0.12 | 2.16 | 0.07 | 0.79 | 0.99 |
| -15 | 70 | 7.75 | 366.00 | 0.13 | 1.82 | 180.00 | 0.13 | 2.54 | 0.07 | 0.75 | 0.98 |
| -15 | 90 | 7.94 | 180.00 | 0.20 | 1.99 | 366.00 | 0.14 | 1.56 | 0.11 | 0.76 | 0.98 |
| -15 | 110 | 7.88 | 180.00 | 0.23 | 2.16 | 361.34 | 0.21 | 1.43 | 0.19 | 0.80 | 0.98 |
| -15 | 130 | 7.79 | 366.00 | 0.38 | 1.33 | 197.25 | 0.26 | 1.96 | 0.28 | 0.75 | 1.00 |
| -15 | 150 | 7.84 | 292.35 | 0.26 | 1.13 | 180.00 | 0.13 | 2.85 | 0.16 | 0.78 | 0.98 |
| -15 | 170 | 8.02 | 334.77 | 0.18 | 0.36 | 180.00 | 0.05 | 2.75 | 0.08 | 0.79 | 1.00 |
| -15 | 190 | 7.92 | 366.00 | 0.19 | 1.63 | 198.65 | 0.17 | 2.12 | 0.15 | 0.84 | 0.99 |
| -15 | 210 | 7.65 | 366.00 | 0.38 | 1.63 | 204.71 | 0.20 | 1.70 | 0.31 | 0.83 | 1.00 |
| -15 | 230 | 7.62 | 366.00 | 0.18 | 1.13 | 184.66 | 0.14 | 1.27 | 0.12 | 0.83 | 0.99 |
| -15 | 250 | 7.75 | 305.86 | 0.14 | -0.82 | 180.00 | 0.03 | 1.34 | 0.07 | 0.83 | 1.00 |
| -15 | 270 | 7.77 | 366.00 | 0.17 | -2.64 | 186.53 | 0.15 | 2.69 | 0.14 | 0.86 | 1.00 |
| -15 | 290 | 7.85 | 366.00 | 0.22 | 2.42 | 180.93 | 0.16 | 2.68 | 0.18 | 0.84 | 1.00 |
| -15 | 310 | 7.81 | 181.86 | 0.25 | 2.88 | 366.00 | 0.22 | 1.78 | 0.25 | 0.83 | 0.99 |
| -15 | 330 | 7.72 | 274.17 | 0.21 | 1.78 | 180.93 | 0.11 | 2.45 | 0.18 | 0.90 | 1.00 |
| -15 | 350 | 7.71 | 356.68 | 0.23 | 0.67 | 180.00 | 0.16 | 1.92 | 0.18 | 0.84 | 1.00 |
| -5 | 10 | 7.73 | 280.23 | 0.11 | 1.90 | 180.00 | 0.11 | -2.84 | 0.06 | 0.72 | 0.99 |
| -5 | 30 | 7.74 | 180.00 | 0.14 | -2.86 | 254.59 | 0.09 | -2.60 | 0.04 | 0.78 | 0.98 |
| -5 | 50 | 7.73 | 366.00 | 0.10 | -1.62 | 180.00 | 0.06 | -3.00 | 0.02 | 0.73 | 0.98 |
| -5 | 70 | 7.72 | 303.07 | 0.26 | -1.50 | 180.00 | 0.17 | 2.63 | 0.16 | 0.78 | 0.98 |
| -5 | 90 | 7.82 | 283.49 | 0.37 | -1.40 | 180.00 | 0.19 | 2.41 | 0.23 | 0.75 | 0.97 |
| -5 | 110 | 7.81 | 238.74 | 0.40 | -0.49 | 180.00 | 0.25 | 2.87 | 0.24 | 0.75 | 0.97 |
| -5 | 130 | 7.91 | 211.70 | 0.39 | 0.94 | 366.00 | 0.19 | -2.81 | 0.28 | 0.77 | 0.99 |
| -5 | 150 | 8.05 | 228.02 | 0.43 | 0.19 | 180.00 | 0.32 | -2.90 | 0.32 | 0.79 | 0.98 |
| -5 | 170 | 8.09 | 249.46 | 0.28 | 0.26 | 180.00 | 0.26 | -2.59 | 0.23 | 0.79 | 1.00 |
| -5 | 190 | 8.13 | 180.00 | 0.35 | 3.14 | 366.00 | 0.08 | 0.76 | 0.21 | 0.81 | 0.99 |
| -5 | 210 | 8.09 | 200.05 | 0.39 | 2.56 | 366.00 | 0.24 | 1.08 | 0.29 | 0.74 | 0.98 |
| -5 | 230 | 8.03 | 180.00 | 0.37 | -3.07 | 366.00 | 0.19 | 0.72 | 0.24 | 0.75 | 0.99 |
| -5 | 250 | 7.90 | 180.00 | 0.46 | -2.68 | 366.00 | 0.19 | -1.23 | 0.32 | 0.77 | 0.97 |
| -5 | 270 | 7.70 | 180.00 | 0.46 | -2.77 | 366.00 | 0.13 | -1.64 | 0.32 | 0.77 | 0.98 |
| -5 | 290 | 7.70 | 180.00 | 0.34 | -2.76 | 304.00 | 0.14 | -2.01 | 0.18 | 0.74 | 0.98 |





| -5 | 310 | 7.70 | 184.66 | 0.29 | -2.80 | 289.55 | 0.04 | 2.71 | 0.17 | 0.80 | 0.98 |
|---|---|---|---|---|---|---|---|---|---|---|---|
| -5 | 330 | 7.73 | 180.00 | 0.23 | -2.48 | 366.00 | 0.15 | 0.77 | 0.13 | 0.75 | 0.99 |
| -5 | 350 | 7.74 | 276.50 | 0.14 | 1.44 | 180.00 | 0.13 | -2.76 | 0.08 | 0.72 | 0.98 |
| 5 | 10 | 7.85 | 366.00 | 0.30 | -0.94 | 180.00 | 0.28 | -2.58 | 0.22 | 0.75 | 0.98 |
| 5 | 30 | 7.85 | 180.00 | 0.25 | -2.60 | 366.00 | 0.24 | -1.22 | 0.18 | 0.72 | 0.97 |
| 5 | 50 | 7.77 | 366.00 | 0.35 | -1.38 | 180.00 | 0.16 | -3.02 | 0.18 | 0.69 | 0.96 |
| 5 | 70 | 7.74 | 325.44 | 0.49 | -1.36 | 180.00 | 0.16 | 2.54 | 0.29 | 0.68 | 0.95 |
| 5 | 90 | 7.77 | 329.64 | 0.60 | -1.38 | 180.00 | 0.23 | 2.53 | 0.41 | 0.67 | 0.97 |
| 5 | 110 | 7.86 | 332.44 | 0.68 | -1.57 | 180.00 | 0.24 | 2.57 | 0.52 | 0.77 | 0.99 |
| 5 | 130 | 8.00 | 338.50 | 0.58 | -1.76 | 180.00 | 0.24 | 2.55 | 0.54 | 0.85 | 1.00 |
| 5 | 150 | 8.02 | 333.83 | 0.60 | -1.60 | 180.00 | 0.34 | 2.82 | 0.52 | 0.81 | 0.99 |
| 5 | 170 | 7.89 | 364.14 | 0.51 | -1.63 | 180.00 | 0.28 | -2.91 | 0.48 | 0.83 | 1.00 |
| 5 | 190 | 7.99 | 348.75 | 0.46 | -1.70 | 180.00 | 0.19 | 2.88 | 0.39 | 0.83 | 1.00 |
| 5 | 210 | 8.19 | 245.26 | 0.34 | -0.21 | 180.00 | 0.30 | -2.90 | 0.30 | 0.85 | 1.00 |
| 5 | 230 | 8.12 | 180.00 | 0.33 | -3.10 | 366.00 | 0.27 | -1.37 | 0.27 | 0.80 | 0.99 |
| 5 | 250 | 7.93 | 180.00 | 0.42 | -2.81 | 366.00 | 0.27 | -1.32 | 0.35 | 0.78 | 0.98 |
| 5 | 270 | 7.79 | 180.00 | 0.44 | -2.89 | 366.00 | 0.29 | -1.52 | 0.38 | 0.78 | 0.99 |
| 5 | 290 | 7.77 | 366.00 | 0.39 | -1.64 | 180.00 | 0.34 | -3.04 | 0.39 | 0.78 | 0.99 |
| 5 | 310 | 7.70 | 366.00 | 0.20 | -1.84 | 180.00 | 0.20 | -2.76 | 0.15 | 0.79 | 0.98 |
| 5 | 330 | 7.74 | 180.00 | 0.23 | -2.73 | 366.00 | 0.12 | -1.03 | 0.11 | 0.72 | 0.98 |
| 5 | 350 | 7.77 | 366.00 | 0.23 | -1.08 | 180.00 | 0.20 | -2.66 | 0.16 | 0.74 | 0.98 |
| 15 | 10 | 7.86 | 351.08 | 0.36 | -1.36 | 180.00 | 0.08 | 1.70 | 0.24 | 0.79 | 0.98 |
| 15 | 30 | 7.94 | 359.01 | 0.43 | -1.45 | 184.66 | 0.12 | 3.01 | 0.33 | 0.77 | 0.99 |
| 15 | 50 | 7.90 | 352.02 | 0.43 | -1.47 | 180.47 | 0.18 | 2.86 | 0.38 | 0.84 | 1.00 |
| 15 | 70 | 7.82 | 352.95 | 0.52 | -1.57 | 180.00 | 0.11 | 2.27 | 0.41 | 0.80 | 1.00 |
| 15 | 90 | 7.78 | 354.35 | 0.50 | -1.41 | 180.00 | 0.18 | 3.05 | 0.40 | 0.79 | 0.99 |
| 15 | 110 | 7.89 | 366.00 | 0.55 | -1.55 | 180.00 | 0.17 | -2.91 | 0.36 | 0.72 | 0.98 |
| 15 | 130 | 7.96 | 363.67 | 0.50 | -1.65 | 180.47 | 0.15 | 3.06 | 0.28 | 0.71 | 0.96 |
| 15 | 150 | 7.87 | 331.04 | 0.37 | -1.50 | 180.00 | 0.08 | 2.31 | 0.30 | 0.85 | 0.99 |
| 15 | 170 | 7.54 | 366.00 | 0.37 | -1.41 | 180.00 | 0.05 | -2.52 | 0.26 | 0.81 | 0.99 |
| 15 | 190 | 7.60 | 366.00 | 0.55 | -1.51 | 221.49 | 0.07 | -1.57 | 0.40 | 0.76 | 0.98 |
| 15 | 210 | 7.83 | 350.62 | 0.60 | -1.57 | 180.00 | 0.13 | 1.77 | 0.45 | 0.76 | 0.99 |
| 15 | 230 | 7.85 | 348.29 | 0.47 | -1.43 | 180.00 | 0.13 | 1.87 | 0.38 | 0.82 | 0.99 |
| 15 | 250 | 7.78 | 366.00 | 0.19 | -2.21 | 205.64 | 0.07 | -2.86 | 0.08 | 0.80 | 0.99 |
| 15 | 270 | 7.74 | 180.00 | 0.09 | 2.90 | 366.00 | 0.04 | -1.95 | 0.02 | 0.77 | 0.98 |
| 15 | 290 | 7.78 | 282.09 | 0.29 | -0.62 | 180.00 | 0.15 | 2.88 | 0.24 | 0.83 | 0.99 |
| 15 | 310 | 7.87 | 331.50 | 0.41 | -1.38 | 180.00 | 0.14 | 2.02 | 0.39 | 0.88 | 1.00 |





| 15 | 330 | 7.78 | 310.53 | 0.41 | -1.22 | 180.00 | 0.13 | 1.53 | 0.40 | 0.86 | 1.00 |
|----|-----|------|--------|------|-------|--------|------|------|------|------|------|
| 15 | 350 | 7.77 | 344.56 | 0.43 | -1.33 | 180.00 | 0.13 | 1.20 | 0.39 | 0.85 | 0.99 |
| 25 | 10  | 7.54 | 366.00 | 0.15 | -1.79 | 206.11 | 0.07 | -1.43 | 0.03 | 0.62 | 0.91 |
| 25 | 30  | 7.57 | 366.00 | 0.23 | -1.81 | 189.32 | 0.12 | -2.24 | 0.07 | 0.68 | 0.92 |
| 25 | 50  | 7.55 | 366.00 | 0.24 | -1.92 | 198.65 | 0.11 | -2.29 | 0.08 | 0.66 | 0.95 |
| 25 | 70  | 7.46 | 180.00 | 0.19 | -1.81 | 366.00 | 0.14 | -2.08 | 0.09 | 0.71 | 0.95 |
| 25 | 90  | 7.44 | 366.00 | 0.21 | -1.00 | 180.00 | 0.21 | -2.02 | 0.11 | 0.67 | 0.93 |
| 25 | 110 | 7.57 | 366.00 | 0.30 | -0.82 | 185.13 | 0.24 | -1.89 | 0.18 | 0.70 | 0.93 |
| 25 | 130 | 7.64 | 366.00 | 0.36 | -1.08 | 187.92 | 0.12 | -1.61 | 0.19 | 0.75 | 0.96 |
| 25 | 150 | 7.48 | 366.00 | 0.26 | -0.80 | 198.18 | 0.10 | -1.66 | 0.14 | 0.79 | 0.99 |
| 25 | 170 | 7.21 | 366.00 | 0.37 | -0.95 | 206.11 | 0.21 | -1.63 | 0.18 | 0.66 | 0.95 |
| 25 | 190 | 7.20 | 366.00 | 0.49 | -1.19 | 199.11 | 0.24 | -1.44 | 0.32 | 0.71 | 0.96 |
| 25 | 210 | 7.35 | 366.00 | 0.55 | -1.45 | 199.11 | 0.12 | -1.24 | 0.37 | 0.69 | 0.99 |
| 25 | 230 | 7.39 | 366.00 | 0.49 | -1.53 | 192.12 | 0.12 | -0.96 | 0.32 | 0.70 | 0.96 |
| 25 | 250 | 7.36 | 180.00 | 0.21 | -0.88 | 366.00 | 0.20 | -2.23 | 0.11 | 0.67 | 0.94 |
| 25 | 270 | 7.36 | 202.38 | 0.20 | -1.60 | 365.53 | 0.04 | -2.00 | 0.05 | 0.63 | 0.93 |
| 25 | 290 | 7.50 | 366.00 | 0.14 | -0.72 | 208.44 | 0.09 | -1.22 | 0.03 | 0.67 | 0.95 |
| 25 | 310 | 7.56 | 240.60 | 0.19 | -0.71 | 366.00 | 0.17 | -0.68 | 0.12 | 0.74 | 0.98 |
| 25 | 330 | 7.49 | 243.40 | 0.29 | -0.21 | 180.00 | 0.11 | 0.70 | 0.16 | 0.65 | 0.96 |
| 25 | 350 | 7.44 | 276.03 | 0.20 | -0.72 | 180.00 | 0.11 | 0.75 | 0.08 | 0.63 | 0.92 |
| 35 | 10  | 7.37 | 366.00 | 0.19 | -1.41 | 204.71 | 0.17 | -1.16 | 0.06 | 0.61 | 0.93 |
| 35 | 30  | 7.38 | 366.00 | 0.24 | -1.46 | 200.51 | 0.18 | -1.21 | 0.09 | 0.67 | 0.92 |
| 35 | 50  | 7.33 | 366.00 | 0.19 | -1.41 | 204.24 | 0.12 | -1.54 | 0.05 | 0.62 | 0.91 |
| 35 | 70  | 7.28 | 366.00 | 0.22 | -1.26 | 197.71 | 0.20 | -1.70 | 0.08 | 0.64 | 0.92 |
| 35 | 90  | 7.28 | 366.00 | 0.32 | -0.96 | 191.65 | 0.25 | -1.67 | 0.15 | 0.61 | 0.92 |
| 35 | 110 | 7.40 | 366.00 | 0.48 | -0.93 | 186.99 | 0.22 | -1.30 | 0.30 | 0.72 | 0.95 |
| 35 | 130 | 7.47 | 366.00 | 0.48 | -0.88 | 180.00 | 0.21 | -0.58 | 0.35 | 0.75 | 0.96 |
| 35 | 150 | 7.39 | 366.00 | 0.46 | -0.83 | 192.59 | 0.17 | -1.22 | 0.30 | 0.73 | 0.96 |
| 35 | 170 | 7.22 | 366.00 | 0.45 | -0.92 | 201.91 | 0.19 | -1.56 | 0.26 | 0.67 | 0.96 |
| 35 | 190 | 7.19 | 366.00 | 0.58 | -1.10 | 200.98 | 0.22 | -1.44 | 0.34 | 0.68 | 0.96 |
| 35 | 210 | 7.24 | 366.00 | 0.65 | -1.27 | 195.38 | 0.23 | -1.50 | 0.44 | 0.71 | 0.97 |
| 35 | 230 | 7.25 | 366.00 | 0.66 | -1.30 | 198.18 | 0.24 | -1.53 | 0.43 | 0.70 | 0.96 |
| 35 | 250 | 7.26 | 366.00 | 0.48 | -1.43 | 199.58 | 0.28 | -1.63 | 0.27 | 0.65 | 0.95 |
| 35 | 270 | 7.30 | 181.86 | 0.24 | -1.08 | 366.00 | 0.18 | -1.17 | 0.13 | 0.71 | 0.95 |
| 35 | 290 | 7.45 | 180.00 | 0.20 | -0.58 | 355.74 | 0.14 | -0.43 | 0.09 | 0.72 | 0.96 |
| 35 | 310 | 7.42 | 366.00 | 0.27 | -1.05 | 180.00 | 0.20 | 0.07 | 0.17 | 0.73 | 0.96 |
| 35 | 330 | 7.30 | 366.00 | 0.32 | -1.20 | 180.00 | 0.18 | 0.59 | 0.17 | 0.65 | 0.94 |



| 35 | 350 | 7.25 | 366.00 | 0.29 | -1.30 | 198.18 | 0.16 | -0.46 | 0.11 | 0.57 | 0.92 |
|---|---|---|---|---|---|---|---|---|---|---|---|
| 45 | 10 | 7.47 | 337.10 | 0.61 | -1.08 | 180.00 | 0.29 | 1.17 | 0.48 | 0.78 | 0.97 |
| 45 | 30 | 7.44 | 339.89 | 0.53 | -1.10 | 180.00 | 0.30 | 1.11 | 0.42 | 0.72 | 0.97 |
| 45 | 50 | 7.42 | 260.18 | 0.43 | -0.82 | 366.00 | 0.31 | -0.73 | 0.38 | 0.70 | 0.97 |
| 45 | 70 | 7.38 | 320.32 | 0.52 | -0.82 | 180.00 | 0.25 | 1.24 | 0.39 | 0.74 | 0.96 |
| 45 | 90 | 7.35 | 314.26 | 0.60 | -0.69 | 180.00 | 0.26 | 1.24 | 0.43 | 0.74 | 0.94 |
| 45 | 110 | 7.46 | 311.46 | 0.60 | -0.63 | 180.00 | 0.24 | 1.05 | 0.45 | 0.74 | 0.98 |
| 45 | 130 | 7.50 | 294.68 | 0.56 | -0.35 | 180.00 | 0.25 | 0.86 | 0.45 | 0.78 | 0.97 |
| 45 | 150 | 7.51 | 334.30 | 0.63 | -0.83 | 180.00 | 0.28 | 1.06 | 0.53 | 0.80 | 0.98 |
| 45 | 170 | 7.43 | 322.18 | 0.59 | -0.77 | 180.00 | 0.31 | 1.14 | 0.52 | 0.78 | 0.97 |
| 45 | 190 | 7.41 | 345.95 | 0.73 | -1.11 | 180.00 | 0.32 | 1.09 | 0.59 | 0.79 | 0.98 |
| 45 | 210 | 7.43 | 362.74 | 0.83 | -1.29 | 180.00 | 0.27 | 1.23 | 0.62 | 0.73 | 0.99 |
| 45 | 230 | 7.49 | 366.00 | 0.96 | -1.39 | 180.00 | 0.23 | 1.03 | 0.63 | 0.72 | 0.98 |
| 45 | 250 | 7.53 | 366.00 | 0.93 | -1.48 | 180.00 | 0.19 | 0.78 | 0.58 | 0.69 | 0.96 |
| 45 | 270 | 7.67 | 366.00 | 0.67 | -1.31 | 206.57 | 0.21 | -1.30 | 0.44 | 0.72 | 0.98 |
| 45 | 290 | 7.66 | 366.00 | 0.54 | -1.26 | 188.39 | 0.20 | -0.33 | 0.39 | 0.75 | 0.99 |
| 45 | 310 | 7.59 | 366.00 | 0.62 | -1.27 | 180.00 | 0.24 | 0.70 | 0.45 | 0.73 | 0.98 |
| 45 | 330 | 7.52 | 359.01 | 0.70 | -1.22 | 180.00 | 0.33 | 1.10 | 0.53 | 0.73 | 0.99 |
| 45 | 350 | 7.44 | 349.22 | 0.67 | -1.20 | 180.00 | 0.30 | 1.32 | 0.48 | 0.68 | 0.97 |
| 55 | 10 | 7.40 | 262.51 | 0.93 | -0.16 | 180.00 | 0.31 | 0.82 | 0.58 | 0.64 | 0.95 |
| 55 | 30 | 7.38 | 249.46 | 0.73 | -0.39 | 366.00 | 0.37 | -0.88 | 0.57 | 0.66 | 0.93 |
| 55 | 50 | 7.36 | 253.19 | 0.59 | -0.40 | 366.00 | 0.34 | -0.63 | 0.47 | 0.69 | 0.92 |
| 55 | 70 | 7.27 | 248.53 | 0.73 | 0.16 | 180.00 | 0.28 | 0.73 | 0.49 | 0.54 | 0.80 |
| 55 | 90 | 7.25 | 245.26 | 0.59 | -0.06 | 366.00 | 0.27 | -0.69 | 0.41 | 0.64 | 0.93 |
| 55 | 110 | 7.31 | 232.21 | 0.57 | 0.26 | 366.00 | 0.20 | -0.80 | 0.41 | 0.71 | 0.94 |
| 55 | 130 | 7.32 | 230.81 | 0.65 | 0.30 | 366.00 | 0.21 | -0.81 | 0.47 | 0.68 | 0.95 |
| 55 | 150 | 7.34 | 230.81 | 0.62 | 0.26 | 366.00 | 0.22 | -0.71 | 0.46 | 0.70 | 0.95 |
| 55 | 170 | 7.31 | 227.55 | 0.71 | 0.29 | 366.00 | 0.25 | -0.85 | 0.55 | 0.71 | 0.95 |
| 55 | 190 | 7.32 | 234.54 | 0.76 | 0.03 | 366.00 | 0.32 | -0.93 | 0.57 | 0.67 | 0.94 |
| 55 | 210 | 7.41 | 259.25 | 0.88 | -0.11 | 180.00 | 0.29 | 1.10 | 0.58 | 0.63 | 0.94 |
| 55 | 230 | 7.54 | 290.48 | 1.00 | -0.64 | 180.00 | 0.37 | 1.21 | 0.67 | 0.63 | 0.95 |
| 55 | 250 | 7.63 | 327.31 | 1.07 | -1.16 | 180.00 | 0.37 | 1.27 | 0.68 | 0.69 | 0.95 |
| 55 | 270 | 7.65 | 347.82 | 1.05 | -1.36 | 180.00 | 0.42 | 0.86 | 0.66 | 0.69 | 0.95 |
| 55 | 290 | 7.69 | 347.35 | 0.91 | -1.28 | 180.00 | 0.38 | 0.93 | 0.61 | 0.73 | 0.95 |
| 55 | 310 | 7.61 | 329.64 | 0.95 | -1.00 | 180.00 | 0.45 | 1.03 | 0.66 | 0.70 | 0.96 |
| 55 | 330 | 7.60 | 317.52 | 0.86 | -0.82 | 180.00 | 0.43 | 1.12 | 0.60 | 0.64 | 0.95 |
| 55 | 350 | 7.44 | 266.71 | 0.97 | -0.24 | 180.00 | 0.35 | 0.86 | 0.61 | 0.62 | 0.94 |