# Peer review of "Variability of the Brunt-Väisälä frequency at the OH-airglow layer height at low and mid latitudes"

_Atmospheric Measurement Techniques, 2020_

## Referee Comment (RC1) · Anonymous Referee #1 · 29 May 2020

In their paper, the authors derive a climatology of the Brunt-Vaisala Frequency (BVF) at the OH airglow layer height using TIMED-SABER observations from satellite. The purpose of this climatology is to complement observations of OH airglow spectrometers that provide temperatures only averaged over the OH layer. With the help of the BVF climatology potential energies can be calculated from temperature fluctuations caused by gravity waves. A BVF climatology is therefore a useful tool for these kind of ground based observations.

Overall, the paper is well written and of interest for the readership of AMT. The paper is recommended for publication in AMT after addressing my major comment and my

additional comments.

Major comment:

It is well-known that atmospheric tides in the MLT can have strong influence on the BVF. Potential effects of tides should therefore be mentioned in the introduction, perhaps on p.3, after l.15. The first occurrence of the word "tides" is on p.6, which is way too late. In their paper, the authors investigate possible effects of tides, however effects of tides are not included in their BVF climatology. This should be mentioned in the abstract and the summary of the paper.

Several papers on tides observed with TIMED-SABER are relevant for the current study, but not considered. These papers should be mentioned in the introduction and some discussion be added, where appropriate:

Mukhtarov, P., Pancheva, D., and Andonov, B.: Global structure and seasonal and inter-annual variability of the migrating diurnal tide seen in the SABER/TIMED temperatures between 20 and 120 km, J. Geophys. Res., 114, A02309, doi:10.1029/2008JA013759, 2009.

This paper shows that near the mesopause the DW1 amplitude can exceed 10K at the equator and 5K at midlatitudes. Vertical wavelength is usually short (20-25km). Therefore the DW1 should have effect on the BVF and OH layer FWHM.

Pancheva, D., Mukhtarov, P., and Andonov, B.: Global structure, seasonal and interannual variability of the migrating semidiurnal tide seen in the SABER/TIMED temperatures (2002-2007), Ann. Geophys., 27, 687-703, 2009.

This paper shows that near the mesopause the SW2 amplitude is up to 10K at midlatitudes. Also this tidal mode should affect the BVF and OH layer FWHM.

There is also a climatology of eastward propagating tides:

Pancheva, D., Mukhtarov, P., and Andonov, B.: Global structure, seasonal and interannual variability of the eastward propagating tides seen in the SABER/TIMED temperatures (2002-2007), Advances in Space Research, 46, 257-274, 2010.

showing that near the mesopause eastward propagating tides should have smaller amplitudes and smaller effect on the BVF and OH layer FWHM than the DW1 and the SW2.

Additional comments:

p.2, Eq.1: Please state explicitly that gamma=g/cp

p.3, l.12: Please state a typical value of the OH layer FWHM

p.4, l.24: Here you use a latitude range of 60S to 60N. However, TIMED-SABER covers only 50S to 50N continuously. Therefore the information should be included that latitudes 50 to 60 deg are an extrapolation when TIMED-SABER views toward the other hemisphere.

p.5, l.4-7: Please state here that also local time is relevant!

p.5, l.20: Please mention that also tides should have effect on the FWHM of the OH layer (due to temperature variation and secondary circulation induced by tides). Therefore the 60d oscillations could be related to the yaw cycle of the TIMED satellite and to changes in the local time of TIMED-SABER observations.

p.5, l.33: again: TIMED-SABER observes continuously only between 50S and 50N!

p.6, l.9: Please state that only nighttime TIMED overpassings are considered in Fig. 3.

p.6, l.9: Please clarify whether these overpass times are the satellite overpassings, or the local time of TIMED-SABER observations. As TIMED-SABER views sideward, there should be a considerable difference!

p.6, l.20: what is $R^2$ ? The linear correlation coefficient squared? However, it should be pointed out that the annual cycle was not taken out, and this will increase the scatter in Fig. 4. Consequently, the linear relations shown in Fig.4 could be much more significant than suggested by the correlation coefficients. As amplitudes of tides vary strongly during one year, the linear relations should also be time dependent.

p.6, l.24: Again, not clear whether low $R^2$ values are meaningful! The correlation could be better if seasonal variations are accounted for.

p.6, l.26: This question is somehow out of place as it suggests some surprising effect! It is however well-known that tides have significant effect on the MLT!

p.6, l.26-30: There are several papers that quantify tides derived from TIMED-SABER temperatures. See my major comment.

p.7, l.19: See the paper by Pancheva et al. (2009) for effects of semidiurnal tides.

p.8, l.26: "artificial oscillation" is not a good expression here! The oscillation arises from sampling tides at different phase. State clearly that tidal effects are neglected and not included in the climatology.

p.8, l.28: how is the "quality of approximation" defined? Please explain!

p.10, l.9/10: It should be stated clearly that tidal waves will have some effect on the BVF, and this effect is not considered in the climatology.

Other comments:

p.3, l.5: high enough -> good enough

p.3, l.5: mistime -> miss-time

p.3, l.6: misdistance -> miss-distance

p.6, l.18 simoultaneously -> simultaneously

p.7, l.5: agress -> agrees

p.7, l.16: seperated -> separated

p.8, l.18: charaterized -> characterized

p.8, l.28: summerized -> summarized

p.9, l.1: asymetries -> asymmetries

———————————————————

---

## Referee Comment (RC2) · Anonymous Referee #2 · 3 Jun 2020

OH airglow spectrometers can provide information about atmospheric temperature and its variability from the height of the airglow layer, which is roughly centered around 87 km. In order to evaluate gravity wave parameters, the gravity wave potential energy density (GWPED) is the more meaningful parameter. GWPED calculation requires the knowledge of Brunt-Vaisala-frequency N, which can not be directly derived from airglow data. The paper by Wüst et al. provides a data set of BV-frequency on a nightly mean basis for a broad latitude range including low and mid latitudes based on a temperature climatology by TIMED/SABER. Annual and semiannual variability of N are quantified, and reasons for higher-order variations are discussed. By intention, due to the choice of methods, the paper is solely relevant for evaluation of OH airglow data.

information about the "true" N around 87 km nor information about other airglow layer heights is provided. However, the paper will be of interest for a large community. I have several comments on the structure of the manuscript as well as on the depth of the results that I like to see addressed before the paper can be published.

Major comments:

- A major concern is the missing analysis of tidal effects of the results. Fig. 3 shows that SABER observes at a local time that is slowly precessing with time. That means that for a number of consecutive days the measurements happen at a distinct phase of the tides, producing a systematic offset of temperature and temperature gradient compared to the nightly mean. This becomes obvious when the data flip from ascending to descending node or vice versa. The authors somewhat discuss this effect for the apparent 60-d-variation of N, but I think the different potential biases of the results due to tidal waves needs to be further elaborated.

- P7L20-P7L25: I agree with the authors that R2 is very low in these cases. As a consequence, the linear equation explains only a very small fraction of the relation between temperature (or T gradient) and local time. I wonder, why I should expect a linear relation at all. A sinusoidal relation due to tides might be even more likely. Even if R2 is such small, in Fig. 4e a very precise linear local time dependence of N evolves. I would like to see some more discussion of this, from my point of view, surprising result presented in Fig. 4e. Minor topic: Please add an "x" in the best fit equation for 45°N in Fig. 4e.

- P9 (Harmonic Analysis): What is the reason for allowing the fit two arbitrary frequencies between 180 and 366 d? The authors argue that they fit an annual and semiannual variation (which is reasonable), but the periods deviate partly by some tens of days from the particular annual/semi-annual periods (see tables 1-3). For example, in Table 1 at -35° latitude a period of 256 days (side note: I think, two decimals are far beyond physically reasonable) is given for the annual variation at 50° E, while in AMTD
the next longitude bin 229 days is interpreted as semi-annual. Furthermore, partly the period of one of the oscillations is exactly 366 d or 180 d for several adjacent regions. This seems to be somewhat artificial, like if the fit would "prefer" some period outside the limits. I suggest fitting some fixed periods of 183 days and 366 days for the whole data set. Regarding the results of the harmonic analysis, the authors acknowledge a low quality of the fits at low latitudes. Unfortunately, they focus in the discussion mainly on the longitudinal differences. Please discuss the consequences of the low quality in more detail. How representative is the climatology if the fit quality is low? I further suggest showing the fit results together with the original data for at least one or two representative examples.

Minor comments:

- P5L15: I see the range "60°S to 60°N" somewhat misleading and overambitious, if true whole-year data coverage is only between 52°S and 52°N. P5L10 says that data refer to the mid of the interval, i.e. 60°S would mean 55°S-65°S. Even if the last interval is centered around 55°S/N, as Fig. 2 suggests, this interval would in fact only contain data of 50°S/N-52°S/N.

- P6L28: What is the reason for giving a percentage variation of the OH layer height? How can I interpret a 1% change?

- P6L33: As mentioned above, the yaw cycle affects data coverage already poleward of  ${\sim}52^{\circ}.$

- P7-8: I suggest making the structure of the results sections more obvious to the reader. E.g., I realized quite lately that a large fraction of pages 7 and 8 explains the reason for the 60-d-oscillation as an artefact of the yaw-cycle in relation with tidal variations. Sub-section headings, e.g., could help the reader to follow the line of arguments.

- P7L32-33: I am sorry, but I do not understand this argument. Additionally, I suggest removing the brackets around "the mean vertical wavelength ...".
- P8L9-10: I suggest adding some text that tides not necessarily always have stable phases (there is a lot of observational evidence for "unstable" tides). But in this case, arguing for potential artefacts, the chance of stable phases is sufficient.

- P8L11: I suggest removing "However", as the following sentence is not in contrast to the previous.

- P8L17: I am sorry, but I do not understand why you can conclude, "a semi-diurnal tide must be present". Please try to improve.

- P9L2: If I understand the paper correctly, you may add "and, therefore, ignore the 60-d-oscillation in our BV climatology."

- P9L28: What is the measure for the quality of approximation? R2, again?

- P10L14: The  $5^{\circ}x7^{\circ}$  pixel has less than 25% of the size (area) of the  $10^{\circ}x20^{\circ}$  pixel.

- Fig. 3: I suggest using different open and filled symbols. The two gray colors are hard to distinguish. (Please apply to Fig. 5 and 6 accordingly.)

- Fig 4: The caption gives the wrong color coding for Subpanel e).

---

## Author Response (AR1)

We would like to thank the anonymous reviewer for the valuable comments. Our answers are inserted in orange in the text below.

**Anonymous Referee #1**

In their paper, the authors derive a climatology of the Brunt-Vaisala Frequency (BVF) at the OH airglow layer height
20  using TIMED-SABER observations from satellite. The purpose of this climatology is to complement observations of OH airglow spectrometers that provide temperatures only averaged over the OH layer. With the help of the BVF climatology potential energies can be calculated from temperature fluctuations caused by gravity waves. A BVF climatology is therefore a useful tool for these kind of ground based observations.

Overall, the paper is well written and of interest for the readership of AMT. The paper is recommended for publication in
25  AMT after addressing my major comment and my additional comments.

**Major comment:**

It is well-known that atmospheric tides in the MLT can have strong influence on the BVF. Potential effects of tides should
30  therefore be mentioned in the introduction, perhaps on p.3, after l.15. Done

The first occurrence of the word "tides" is on p.6, which is way too late. In their paper, the authors investigate possible effects of tides, however effects of tides are not included in their BVF climatology. This should be mentioned in the abstract and the summary of the paper. I provided an estimation of tidal effects in the section results and discussion, subsection
35  3.3. Therefore, I did not change the abstract, but mentioned the estimation of tidal effects in the summary.

Several papers on tides observed with TIMED-SABER are relevant for the current study, but not considered. These papers should be mentioned in the introduction and some discussion be added, where appropriate:

Thank you for these hints. I inserted them in the introduction and also in section 3 where appropriate. Due to this comment and comments from the other reviewer, I re-arranged this part of the discussion (now 3.2, 60 d oscillation).

Mukhtarov, P., Pancheva, D., and Andonov, B.: Global structure and seasonal and inter- annual variability of the migrating diurnal tide seen in the SABER/TIMED temperatures between 20 and 120 km, J. Geophys. Res., 114, A02309, doi:10.1029/2008JA013759, 2009.

This paper shows that near the mesopause the DW1 amplitude can exceed 10K at the equator and 5K at midlatitudes. Vertical wavelength is usually short (20-25km). Therefore the DW1 should have effect on the BVF and OH layer FWHM.

Pancheva, D., Mukhtarov, P., and Andonov, B.: Global structure, seasonal and interannual variability of the migrating semidiurnal tide seen in the SABER/TIMED tempera- tures (2002-2007), Ann. Geophys., 27, 687-703, 2009.

This paper shows that near the mesopause the SW2 amplitude is up to 10K at midlatitudes. Also this tidal mode should affect the BVF and OH layer FWHM.

There is also a climatology of eastward propagating tides:

Pancheva, D., Mukhtarov, P., and Andonov, B.: Global structure, seasonal and interan nual variability of the eastward propagating tides seen in the SABER/TIMED tempera- tures (2002-2007), Advances in Space Research, 46, 257-274, 2010.

showing that near the mesopause eastward propagating tides should have smaller amplitudes and smaller effect on the BVF and OH layer FWHM than the DW1 and the SW2.

**Additional comments:**

p.2, Eq.1: Please state explicitly that gamma=g/cp Done

p.3, l.12: Please state a typical value of the OH layer FWHM Done

p.4, l.24: Here you use a latitude range of 60°S to 60°N. However, TIMED-SABER covers only 50S to 50N continuously. Therefore the information should be included that latitudes 50 to 60 deg are an extrapolation when TIMED-SABER views toward the other hemisphere. I clarified that SABER delivers data between 52°S and 52°N the whole year.

p.5, l.4-7: Please state here that also local time is relevant! Done

p.5, l.20: Please mention that also tides should have effect on the FWHM of the OH layer (due to temperature variation and secondary circulation induced by tides). Therefore the 60d oscillations could be related to the yaw cycle of the TIMED satellite and to changes in the local time of TIMED-SABER observations. I mention the possible effect of tides on the FWHM later in the paper (page 8 ll. 12). At this stage, I have already explained the 60d effect in the BV frequency which is probably also related to the yaw cycle and the changes in local observation time.

p.5, l.33: again: TIMED-SABER observes continuously only between 50S and 50N! Clarified

p.6, l.9: Please state that only nighttime TIMED overpassings are considered in Fig. 3. Done

p.6, l.9: Please clarify whether these overpass times are the satellite overpassings, or the local time of TIMED-SABER observations. As TIMED-SABER views sideward, there should be a considerable difference! The parameter "time" in the SABER files is given in UTC. When I calculate the overpass time I do this by is using the tangent point (and not the spacecraft) longitude. So, the local time I mention is the local time of the tangent point and not of the spacecraft.

p.6, l.20: what is R^2? The linear correlation coefficient squared? Yes. However, it should be pointed out that the annual cycle was not taken out, and this will increase the scatter in Fig. 4. Consequently, the linear relations shown in Fig.4 could be much more significant than suggested by the correlation coefficients. As amplitudes of tides vary strongly during one year, the linear relations should also be time dependent. I inserted some discussion concerning these points.

p.6, l.24: Again, not clear whether low Rˆ2 values are meaningful! The correlation could be better if seasonal variations are accounted for. I inserted some discussion concerning these points.

p.6, l.26: This question is somehow out of place as it suggests some surprising effect! It is however well-known that tides have significant effect on the MLT! Question deleted.

p.6, l.26-30: There are several papers that quantify tides derived from TIMED-SABER temperatures. See my major comment. Yes that's true and I included them in other parts of the paper, especially a little bit later in the same section. Here, I would like to stick to Offermann et al. (2009) because those authors show a very nice plot in the publication (which I mention explicitly) and this publication is only used for the generalized information that "at low latitudes the effect of tidal motions on temperature is at least in the same order of magnitude as the influence of planetary waves" in the context of the comparison tides versus planetary waves.

p.7, l.19: See the paper by Pancheva et al. (2009) for effects of semidiurnal tides. Thank you, included

p.8, l.26: "artificial oscillation" is not a good expression here! Done The oscillation arises from sampling tides at different phase. State clearly that tidal effects are neglected and not included in the climatology. I changed the text to: "In order to avoid the approximation of the 60 d oscillation, which might be due to sampling tides at different phases as discussed above, the period range which the harmonic analysis uses is restricted to 180–366 d. That means tidal effects are not included here."

p.8, l.28: how is the "quality of approximation" defined? Please explain! I inserted in brackets after quality of approximation … that means $1 - \sigma_{res}^2/\sigma^2$ where $\sigma^2$ is the variance of the original time series and $\sigma_{res}^2$ is the variance of the residual time series, so original time series minus approximation …

p.10, l.9/10: It should be stated clearly that tidal waves will have some effect on the BVF, and this effect is not considered in the climatology. See comment above referring to the second major point.

**Other comments:**
p.3, l.5: high enough -> good enough Corrected

p.3, l.5: mistime -> miss-time Corrected

p.3, l.6: misdistance -> miss-distance Corrected

p.6, l.18 simoultaneously -> simultaneously Corrected

p.7, l.5: agress -> agrees Corrected

p.7, l.16: seperated -> separated Corrected

p.8, l.18: charaterized -> characterized Corrected

p.8, l.28: summerized -> summarized Corrected

p.9, l.1: asymetries -> asymmetries Corrected

Atmos. Meas. Tech. Discuss., doi:10.5194/amt-2020-73-RC2, 2020

[Figure]

**Answer to**

*Interactive comment on* **"Variability of the Brunt-Väisälä frequency at**
10 **the OH-airglow layer**
**height at low and mid latitudes"** *by* **Sabine Wüst et al.**

We would like to thank the anonymous reviewer for the valuable comments. Our answers are inserted in orange in the text
below.

15 From the page and line references in the minor comments I conclude that the reviewer used the original version of the
manuscript and not the one after the quick review. The changes during the quick review process were marginal in most
parts of the manuscript (most of changes referred to section 2) and the points on which the reviewer comments are still
part of the manuscript, so I just answered them and changed the manuscript where needed.

20 **Anonymous Referee #2**

OH airglow spectrometers can provide information about atmospheric temperature and its variability from the height of the
airglow layer, which is roughly centered around 87 km. In order to evaluate gravity wave parameters, the gravity wave
25 potential energy density (GWPED) is the more meaningful parameter. GWPED calculation requires the knowledge of Brunt-
Vaisala-frequency N, which can not be directly derived from airglow data. The paper by Wüst et al. provides a data set of BV-
frequency on a nightly mean basis for a broad latitude range including low and mid latitudes based on a temperature
climatology by TIMED/SABER. Annual and semiannual variability of N are quantified, and reasons for higher-order
variations are discussed. By intention, due to the choice of methods, the paper is solely relevant for evaluation of OH
30 airglow data. Neither information about the "true" N around 87 km nor information about other airglow layer heights is
provided. However, the paper will be of interest for a large community. I have several comments on the structure of the
manuscript as well as on the depth of the results that I like to see addressed before the paper can be published.

**Major comments:**

A major concern is the missing analysis of tidal effects of the results. Fig. 3 shows that SABER observes at a local time that is slowly precessing with time. That means that for a number of consecutive days the measurements happen at a distinct phase of the tides, producing a systematic offset of temperature and temperature gradient compared to the nightly mean. This becomes obvious when the data flip from ascending to descending node or vice versa. The authors
5 somewhat discuss this effect for the apparent 60-d-variation of N, but I think the different potential biases of the results due to tidal waves needs to be further elaborated.
I included an approximation of the uncertainty in N during one night due to tidal effects in the discussion: "As mentioned above, tidal effects are not included in the approximation of the OH*-equivalent BV frequency. However, an uncertainty range of the OH*-equivalent BV frequency due to these effects is an additional useful information. In order to estimate it, SABER
10 profiles which refer to the same night (between 6 p.m. and 6 a.m.) and are separated by six hours at minimum are collected for all years (2002–2018) at each grid point. Since tides have periods in the range of six hours and more (the dominating ones have periods of 12 h and 24 h), we argue that the difference of the OH*-equivalent BV frequency within these hours during one night is mainly due to tides. Of course, also gravity waves still play a role in this period range. Since our gridding is relatively coarse for gravity waves, we assume that gravity waves increase and decrease the OH*-equivalent BV frequency in
15 one pixel so that their effect cancels out over time. This is not true for larger scale phenomena. The mean difference per hour over all years is calculated at each grid point. The results are averaged over latitude afterwards (table 2). The results are averaged over latitude afterwards (table 2). The number of data per latitude is in the range of some 10 000. As mentioned above, tidal activity varies during the year. Therefore, the provision of monthly values would make more sense. However, especially at mid and high latitudes, SABER profiles which refer to the same night separated by six hours at least are not
20 evenly distributed over the year or not available at every month. The uncertainty range provided in table 2 can therefore be regarded as a rather rough estimate. With respect to an OH*-equivalent BV frequency of 0.02 1/s, the results are in the range of ca. 1–2%. For a night of twelve hours tidal effects sum up to ca. 21 % at maximum (that means for low latitudes). We can assume that the approximation of the OH*-equivalent BV frequency refers to midnight, so the tidal effects can be approximated by ±11 % for the whole night in this case." This procedure is described in the section results and discussion
25 on page 9.
Furthermore, I re-arranged the discussion of the 60d oscillation.

P7L20-P7L25: I agree with the authors that $R^2$ is very low in these cases. As a consequence, the linear equation explains only a very small fraction of the relation between temperature (or T gradient) and local time. I wonder, why I
30 should expect a linear relation at all. A sinusoidal relation due to tides might be even more likely. Even if $R^2$ is such small, in Fig. 4e a very precise linear local time dependence of N evolves. I would like to see some more discussion of this, from my point of view, surprising result presented in Fig. 4e. I inserted some discussion concerning these points. Figure 4(e) shows that even in the case that the vertical temperature gradient and the temperature were behaving strictly linearly (so decrease or increase linearly with time), the BV frequency could stay constant.
35 Minor topic: Please add an "x" in the best fit equation for 45˚N in Fig. 4e.Done

P9 (Harmonic Analysis): What is the reason for allowing the fit two arbitrary frequencies between 180 and 366 d? The reason is that we would like to achieve the best approximation and this can be done if the frequencies are chosen freely by the analysis. The analysis was allowed to search for two periods between 180 and 366 d.
40 The authors argue that they fit an annual and semi- annual variation (which is reasonable), but the periods deviate partly by some tens of days from the particular annual/semi-annual periods (see tables 1-3). We do not fit an annual and semi-annual oscillation, we make the analysis search for the two oscillations between 180 d and 366 d which approximate the BV frequency best. In the majority of cases, the analysis finds approximately an annual and a semi-annual variation.
For example, in Table 1 at -35˚ latitude a period of 256 days (side note: I think, two decimals are far beyond physically
45 reasonable changed also in table 2 and 3, which are now table 3 and 4) is given for the annual variation at 50˚ E, while in the next longitude bin 229 days is interpreted as semi-annual.
I deleted the part in the text where annual and semi-annual are associated with the first and the second fitted oscillation. Furthermore, partly the period of one of the oscillations is exactly 366 d or 180 d for several adjacent regions. This seems to be somewhat artificial, like if the fit would "prefer" some period outside the limits. I suggest fitting some fixed
50 periods of 183 days and 366 days for the whole data set. I inserted the case you mentioned above for (-35°N, 50°E) in

the paper (new figure 5). I think it becomes clear that the combination of oscillation (256 & 180 days) is chosen by the analysis since the autumn maximum is flatter than the one in spring. I can repeat the analysis for fixed periods if you wish but I would like to know your opinion about this approximation.

Regarding the results of the harmonic analysis, the authors acknowledge a low quality of the fits at low latitudes. Unfortunately, they focus in the discussion mainly on the longitudinal differences. Please discuss the consequences of the low quality in more detail. How representative is the climatology if the fit quality is low?

It seems that the order of sentences we chose was not optimal. The discussion is about the latitudinal variation of the quality of approximation and the reasons for it. The longitudinal variation is only mentioned in one sentence. I shifted the sentence referring to the longitudinal variation to the end of this paragraph, in order to improve the structure.

I further suggest showing the fit results together with the original data for at least one or two representative examples. I provided these examples in the new figure 5. I chose the one which you mentioned above (-35°N and 50°E) and one mid-European one.

**Minor comments:**

P5L15: I see the range "60°S to 60°N" somewhat misleading and overambitious, if true whole-year data coverage is only between 52°S and 52°N. Changed P5L10 says that data refer to the mid of the interval, i.e. 60°S would mean 55°S-65°S. Even if the last interval is centered around 55°S/N, as Fig. 2 suggests, this interval would in fact only contain data of 50°S/N-52°S/N. Changed in the figure caption and also in the tables.

P6L28: What is the reason for giving a percentage variation of the OH layer height? How can I interpret a 1% change? The sense behind the percentage variations was only to put the variations in a context. For example, a variation of 0.001 1/s in the BV frequency does not sound much, but is a 5% effect relative to the mean BV frequency. I can leave it out. At the moment, it is still in and I clarified that this percentage value is calculated relative to the mean value of the respective variable.

P6L33: As mentioned above, the yaw cycle affects data coverage already poleward of 52°. Changed

P7-8: I suggest making the structure of the results sections more obvious to the reader. E.g., I realized quite lately that a large fraction of pages 7 and 8 explains the reason for the 60-d-oscillation as an artefact of the yaw-cycle in relation with tidal variations. Sub-section headings, e.g., could help the reader to follow the line of arguments. Ok, done.

P7L32-33: I am sorry, but I do not understand this argument. Additionally, I suggest removing the brackets around "the mean vertical wavelength . . .". I removed the brackets and re-formulated the argument: the influence of waves on the temperature gradient depends on the vertical wavelength (the larger the wavelength, the less the influence for wave with the same amplitude) and the amplitude of the wave (the larger the amplitude, the greater the influence for waves with the same vertical wavelength). Additionally I made some changes in the following text.

P8L9-10: I suggest adding some text that tides not necessarily always have stable phases (there is a lot of observational evidence for "unstable" tides). But in this case, arguing for potential artefacts, the chance of stable phases is sufficient. Thank you for this correction. I toned down the statement and provided further information about the phase variations in other publications.

P8L11: I suggest removing "However", as the following sentence is not in contrast to the previous. Corrected.

P8L17: I am sorry, but I do not understand why you can conclude, "a semi-diurnal tide must be present". Please try to improve. I re-arranged this part of the manuscript due to the comments of both reviewers on tidal effects. Doing this, I realized that the argument I gave is only valid if the tide either influences the temperature gradient or the temperature. If both variables are influenced it will not work. Since I think that the argumentation is not necessary after the re-arrangement any more I would prefer to leave it out.

P9L2: If I understand the paper correctly, you may add "and, therefore, ignore the 60-d-oscillation in our BV climatology." Done but I changed "in our BV climatology" to "for our climatology".

P9L28: What is the measure for the quality of approximation? R^2, again? No, it is not R^2. It refers to the harmonic analysis and is calculated as follows: $1 - \sigma_{res}^2/\sigma^2$ where $\sigma^2$ is the variance of the original time series and $\sigma_{res}^2$ is the variance of the residual time series, so original one minus approximation. I gave this information in brackets in the manuscript.

P10L14: The 5°x7° pixel has less than 25% of the size (area) of the 10°x20° pixel. That is also true and I changed it.

Fig. 3: I suggest using different open and filled symbols. The two gray colors are hard to distinguish. (Please apply to Fig. 5 and 6 accordingly. I changed it for figure 3 and 5, in figure 6 only one gray color is used, so nothing to change.

Fig 4: The caption gives the wrong color coding for Subpanel e). Corrected
* * *

[revised manuscript text omitted]

**(a)**

VER-weighted temperature [K]

Line of best fit:
y=-0.94·x+200.03

Local time [h]

**(b)**

VER-weighted temperature gradient [K/km]

Line of best fit:
y=-0.25·x+0.95

Local time [h]

**(c)**

VER-weighted temperature [K]

Line of best fit:
y=-0.78·x+195.98

Local time [h]

**(d)**

VER-weighted temperature gradient [K/km]

Line of best fit:
y=-0.04·x+0.26

Local time [h]

**(e)**

BV angular frequency [1/(10²s)]

Line of best fit:
$y = (-4.0 \cdot 10^{-6} + 2.2) \cdot 10^{-2}$

• 5°N, 10°E
• 45°N, 10°E

Line of best fit:
$y = (-0.02 \cdot x + 2.06) \cdot 10^{-2}$

Local time [h]

[Figure]

[Figure]

**Figure 4 Temperature and vertical temperature gradient, both VER weighted, for 5°N (a and b) and 45°N (c and d) for the year 2002. The nomenclature concerning the local time agrees with the one explained in the caption of figure 3. Subpanel e) shows the development of the BV-frequency during the night for 5°N, 10°E (black) and 45°N, 10°E (black) based on the linear approximation of temperature and its vertical gradient.**

[Figure]

Figure 5 Comparison of the approximation of the BV frequency (solid line) with the approximated values for two different bins. The dashed line refers to the ±10%-interval around the approximation

[Figure]

[Figure]

**Figure 65 Quality of approximation as shown in table 1 versus latitude. All longitudes are plotted but not separated by colour in order to keep the figure clear.**

[Figure]

**Figure 76 Fraction of data within +/- 5 % and +/- 10 % intervals around approximation.**

[Figure]

Figure 87 Comparison of the approximation of the BV frequency in the Alpine region as it was proposed by Wüst et al. (2017 a) based on three oscillations (black) and in this manuscript (gray, values of the pixel 45°N and 10°E are used).

**Tables**

**Table 1**

Period (T), amplitude (A) and phase (φ) of the two oscillations which explain the variability of the daily OH\*-equivalent BV frequency values (averaged over all years) best for a latitudinal and longitudinal gridding of 10° and 20°. They oscillate around the respective mean.

The OH\*-equivalent BV frequency $[s^{-1}]$ can be estimated by $mean + \sum_{i=1}^{2} A_i \ sin\left(\frac{2\pi}{T_i} \cdot DoY - \varphi_i\right)$.

Due to leap years, the total amount of days for one year is set to 366, that means 1st March is DoY 61 for every year.

The harmonic oscillation explains the variability in the time series of daily OH\*-equivalent BV frequency values to a different extent. The respective value is provided in the column "quality of approximation". Additionally, the fraction of data which lies within intervals of +/- 5 % or 10 % around the harmonic approximation is given.

| Lat. [°] | Lon. [°] | Mean $[10^{-2}\ s^{-1}]$ | 1st oscillation | | | 2nd oscillation | | | Quality of approxi-mation | Fraction of data | |
|---|---|---|---|---|---|---|---|---|---|---|---|
| | | | Period [d] | Amp. $[10^{-2}\ s^{-1}]$ | Phase [rad] | Period [d] | Amp. $[10^{-2}\ s^{-1}]$ | Phase [rad] | | +/- 5% | +/- 10% |
| | | | | | | | | | | around approx. | |
| -51 | 10 | 2.19 | 366.0 | 0.24 | -1.56 | 192.6 | 0.13 | -1.68 | 0.87 | 0.92 | 0.99 |
| -51 | 30 | 2.18 | 366.0 | 0.23 | -1.61 | 189.3 | 0.12 | -1.62 | 0.87 | 0.95 | 0.99 |
| -51 | 50 | 2.18 | 366.0 | 0.22 | -1.62 | 187.9 | 0.11 | -1.51 | 0.85 | 0.93 | 0.99 |
| -51 | 70 | 2.17 | 366.0 | 0.21 | -1.61 | 188.9 | 0.10 | -1.60 | 0.83 | 0.93 | 0.99 |
| -51 | 90 | 2.18 | 366.0 | 0.22 | -1.59 | 190.3 | 0.09 | -1.59 | 0.87 | 0.94 | 0.99 |
| -51 | 110 | 2.18 | 366.0 | 0.22 | -1.58 | 188.4 | 0.10 | -1.56 | 0.83 | 0.93 | 0.98 |
| -51 | 130 | 2.17 | 366.0 | 0.24 | -1.56 | 189.8 | 0.12 | -1.62 | 0.88 | 0.95 | 0.99 |
| -51 | 150 | 2.15 | 366.0 | 0.22 | -1.58 | 190.3 | 0.11 | -1.58 | 0.86 | 0.94 | 1.00 |
| -51 | 170 | 2.15 | 366.0 | 0.22 | -1.59 | 189.3 | 0.11 | -1.71 | 0.86 | 0.93 | 0.98 |
| -51 | 190 | 2.15 | 366.0 | 0.23 | -1.58 | 188.9 | 0.12 | -1.69 | 0.88 | 0.93 | 0.98 |
| -51 | 210 | 2.15 | 366.0 | 0.25 | -1.58 | 188.4 | 0.13 | -1.67 | 0.89 | 0.93 | 0.99 |
| -51 | 230 | 2.14 | 366.0 | 0.24 | -1.58 | 186.1 | 0.10 | -1.61 | 0.90 | 0.96 | 0.99 |
| - | 250 | 2.15 | 366.0 | 0.25 | -1.54 | 189.3 | 0.10 | -1.66 | 0.90 | 0.95 | 0.99 |

| | | | | | | | | | | |
|---|---|---|---|---|---|---|---|---|---|---|
| 51 | | |  | | |  | | | | |
| -51 | 270 | 2.16 | 366.0 | 0.24 | -1.54 | 189.3 | 0.11 | -1.69 | 0.86 | 0.93 | 0.99 |
| -51 | 290 | 2.17 | 366.0 | 0.23 | -1.56 | 188.4 | 0.11 | -1.77 | 0.86 | 0.92 | 0.99 |
| 51 | 310 | 2.17 | 366.0 | 0.22 | -1.56 | 188.9 | 0.11 | -1.77 | 0.90 | 0.94 | 1.00 |
| 51 | 330 | 2.17 | 366.0 | 0.22 | -1.55 | 189.3 | 0.11 | -1.76 | 0.88 | 0.94 | 0.98 |
| -51 | 350 | 2.18 | 366.0 | 0.22 | -1.54 | 191.2 | 0.11 | -1.66 | 0.86 | 0.94 | 0.99 |
| -45 | 10 | 2.18 | 366.0 | 0.16 | -1.52 | 201.9 | 0.05 | -1.90 | 0.72 | 0.91 | 0.99 |
| -45 | 30 | 2.18 | 366.0 | 0.17 | -1.55 | 195.4 | 0.05 | -1.77 | 0.80 | 0.93 | 1.00 |
| -45 | 50 | 2.18 | 366.0 | 0.15 | -1.56 | 195.4 | 0.04 | -1.53 | 0.76 | 0.94 | 0.99 |
| -45 | 70 | 2.17 | 366.0 | 0.14 | -1.50 | 204.7 | 0.03 | -1.80 | 0.69 | 0.92 | 1.00 |
| -45 | 90 | 2.17 | 366.0 | 0.15 | -1.49 | 192.1 | 0.03 | -2.04 | 0.80 | 0.96 | 1.00 |
| -45 | 110 | 2.18 | 366.0 | 0.16 | -1.50 | 194.5 | 0.04 | -1.81 | 0.84 | 0.96 | 1.00 |
| -45 | 130 | 2.18 | 366.0 | 0.18 | -1.51 | 193.5 | 0.05 | -1.82 | 0.86 | 0.96 | 1.00 |
| -45 | 150 | 2.16 | 366.0 | 0.16 | -1.50 | 193.1 | 0.04 | -1.71 | 0.80 | 0.95 | 1.00 |
| -45 | 170 | 2.16 | 366.0 | 0.15 | -1.51 | 191.7 | 0.04 | -1.93 | 0.81 | 0.96 | 1.00 |
| -45 | 190 | 2.16 | 366.0 | 0.16 | -1.46 | 193.5 | 0.05 | -1.85 | 0.82 | 0.96 | 1.00 |
| -45 | 210 | 2.16 | 366.0 | 0.17 | -1.48 | 194.0 | 0.05 | -1.76 | 0.83 | 0.96 | 1.00 |
| -45 | 230 | 2.16 | 366.0 | 0.15 | -1.51 | 190.3 | 0.04 | -1.85 | 0.81 | 0.96 | 1.00 |
| -45 | 250 | 2.17 | 366.0 | 0.14 | -1.45 | 192.1 | 0.03 | -1.84 | 0.77 | 0.95 | 1.00 |
| -45 | 270 | 2.18 | 366.0 | 0.13 | -1.44 | 186.1 | 0.04 | -1.92 | 0.73 | 0.94 | 1.00 |
| -45 | 290 | 2.18 | 366.0 | 0.15 | -1.47 | 184.2 | 0.06 | -1.97 | 0.79 | 0.94 | 1.00 |
| -45 | 310 | 2.16 | 366.0 | 0.15 | -1.52 | 191.2 | 0.05 | -2.03 | 0.82 | 0.96 | 1.00 |
| -45 | 330 | 2.16 | 366.0 | 0.14 | -1.50 | 195.4 | 0.04 | -2.14 | 0.76 | 0.95 | 1.00 |
| -45 | 350 | 2.18 | 366.0 | 0.15 | -1.51 | 200.1 | 0.04 | -2.06 | 0.71 | 0.93 | 1.00 |
| -35 | 10 | 2.22 | 365.5 | 0.07 | -1.12 | 180.0 | 0.04 | 1.93 | 0.27 | 0.80 | 0.96 |

| | | | | | | | | | | | |
|---|---|---|---|---|---|---|---|---|---|---|---|
| | | | 3 | | | 0 | | | | | |
| -35 | 30 | 2.22 | 315.7315.65 | 0.08 | -0.72 | 180.0180.00 | 0.04 | 1.79 | 0.43 | 0.86 | 1.00 |
| -35 | 50 | 2.22 | 256.0255.98 | 0.08 | 0.17 | 180.0180.00 | 0.03 | 1.16 | 0.39 | 0.80 | 1.00 |
| -35 | 70 | 2.21 | 229.0228.95 | 0.05 | 0.17 | 366.0366.00 | 0.04 | -0.51 | 0.27 | 0.82 | 1.00 |
| -35 | 90 | 2.21 | 328.2328.24 | 0.05 | -0.59 | 180.0180.00 | 0.04 | 1.97 | 0.33 | 0.88 | 1.00 |
| -35 | 110 | 2.22 | 350.6350.62 | 0.07 | -1.06 | 180.0180.00 | 0.03 | 2.13 | 0.47 | 0.94 | 1.00 |
| -35 | 130 | 2.23 | 354.4354.35 | 0.09 | -1.21 | 180.0180.00 | 0.02 | 2.62 | 0.60 | 0.98 | 1.00 |
| -35 | 150 | 2.22 | 323.6323.58 | 0.06 | -0.76 | 180.0180.00 | 0.02 | 1.77 | 0.39 | 0.94 | 1.00 |
| -35 | 170 | 2.20 | 275.1275.10 | 0.04 | 0.30 | 180.0180.00 | 0.02 | 1.72 | 0.24 | 0.95 | 1.00 |
| -35 | 190 | 2.21 | 333.4333.37 | 0.07 | -0.71 | 180.0180.00 | 0.02 | 2.15 | 0.45 | 0.95 | 1.00 |
| -35 | 210 | 2.23 | 366.0366.00 | 0.09 | -1.16 | 180.0180.00 | 0.01 | 1.71 | 0.63 | 0.98 | 1.00 |
| -35 | 230 | 2.23 | 312.4312.39 | 0.05 | -0.67 | 180.0180.00 | 0.01 | 1.74 | 0.37 | 0.96 | 1.00 |
| -35 | 250 | 2.24 | 199.1199.11 | 0.03 | 1.28 | 366.0366.00 | 0.02 | -0.19 | 0.17 | 0.94 | 1.00 |
| -35 | 270 | 2.23 | 325.9325.91 | 0.03 | 0.16 | 180.0180.00 | 0.02 | 3.03 | 0.16 | 0.93 | 1.00 |
| -35 | 290 | 2.20 | 350.6350.62 | 0.06 | -0.88 | 180.0180.00 | 0.03 | 3.11 | 0.45 | 0.94 | 1.00 |
| -35 | 310 | 2.19 | 342.7342.69 | 0.08 | -1.04 | 180.0180.00 | 0.03 | 2.44 | 0.54 | 0.94 | 1.00 |
| -35 | 330 | 2.19 | 222.9222.89 | 0.05 | 0.61 | 366.0366.00 | 0.02 | -0.90 | 0.28 | 0.86 | 1.00 |
| -35 | 350 | 2.21 | 215.9215.89 | 0.04 | 0.73 | 366.0366.00 | 0.02 | -1.28 | 0.16 | 0.79 | 0.97 |
| -25 | 10 | 2.20 | 180.0180.00 | 0.04 | 1.97 | 366.0366.00 | 0.03 | -0.71 | 0.23 | 0.92 | 1.00 |
| -25 | 30 | 2.20 | 245.7245.73 | 0.05 | -0.13 | 366.0366.00 | 0.03 | -0.33 | 0.43 | 0.98 | 1.00 |
| -25 | 50 | 2.21 | 213.6213.56 | 0.06 | 0.55 | 366.0366.00 | 0.04 | -0.40 | 0.42 | 0.97 | 1.00 |
| -25 | 70 | 2.21 | 180.0180.00 | 0.05 | 1.63 | 366.0366.00 | 0.03 | -0.45 | 0.35 | 0.94 | 1.00 |
| -25 | 90 | 2.19 | 180.0180.00 | 0.04 | 1.69 | 366.0366.00 | 0.02 | 0.04 | 0.26 | 0.95 | 1.00 |
| -25 | 110 | 2.20 | 334.3334.30 | 0.03 | -0.50 | 180.0180.00 | 0.02 | 1.65 | 0.29 | 0.99 | 1.00 |
| -25 | 130 | 2.22 | 354.8354.8 | 0.06 | -1.17 | 180.0180.0 | 0.02 | 1.44 | 0.48 | 0.98 | 1.00 |

| | | | | | | | | | | | |
|---|---|---|---|---|---|---|---|---|---|---|---|
| | | |  | | |  | | | | | |
| -25 | 150 | 2.21 | 208.0  | 0.03 | 0.70 | 366.0  | 0.02 | -0.77 | 0.19 | 0.97 | 1.00 |
| -25 | 170 | 2.19 | 180.0  | 0.04 | 1.56 | 292.4  | 0.03 | 1.61 | 0.31 | 0.98 | 1.00 |
| -25 | 190 | 2.19 | 366.0  | 0.03 | 0.42 | 188.9  | 0.01 | 1.27 | 0.19 | 0.98 | 1.00 |
| -25 | 210 | 2.22 | 366.0  | 0.05 | -0.79 | 221.5  | 0.01 | -1.06 | 0.36 | 0.97 | 1.00 |
| -25 | 230 | 2.24 | 265.8  | 0.04 | 0.43 | 180.0  | 0.02 | 0.57 | 0.20 | 0.98 | 1.00 |
| -25 | 250 | 2.25 | 366.0  | 0.04 | 1.45 | 192.1  | 0.03 | 1.50 | 0.27 | 0.97 | 1.00 |
| -25 | 270 | 2.22 | 366.0  | 0.05 | 1.28 | 180.0  | 0.03 | 2.74 | 0.40 | 0.97 | 1.00 |
| -25 | 290 | 2.19 | 324.1  | 0.03 | 0.74 | 180.0  | 0.02 | 2.86 | 0.23 | 0.98 | 1.00 |
| -25 | 310 | 2.18 | 335.2  | 0.04 | -0.47 | 180.0  | 0.02 | 2.48 | 0.33 | 0.98 | 1.00 |
| -25 | 330 | 2.19 | 180.0  | 0.04 | 1.83 | 366.0  | 0.02 | -0.97 | 0.24 | 0.93 | 1.00 |
| -25 | 350 | 2.21 | 180.0  | 0.06 | 2.15 | 238.7  | 0.03 | -1.79 | 0.23 | 0.90 | 1.00 |
| -15 | 10 | 2.13 | 180.0  | 0.01 | -1.49 | 363.7  | 0.01 | -2.97 | 0.04 | 0.95 | 1.00 |
| -15 | 30 | 2.12 | 366.0  | 0.03 | -1.50 | 180.0  | 0.01 | -1.13 | 0.18 | 0.93 | 1.00 |
| -15 | 50 | 2.13 | 303.1  | 0.03 | -0.58 | 180.0  | 0.01 | -0.04 | 0.20 | 0.98 | 1.00 |
| -15 | 70 | 2.13 | 212.2  | 0.04 | 1.54 | 180.0  | 0.04 | 0.02 | 0.13 | 0.97 | 1.00 |
| -15 | 90 | 2.11 | 366.0  | 0.01 | 2.15 | 180.0  | 0.01 | 0.17 | 0.05 | 0.93 | 1.00 |
| -15 | 110 | 2.11 | 180.0  | 0.02 | -0.67 | 245.7  | 0.01 | 0.26 | 0.05 | 0.92 | 1.00 |
| -15 | 130 | 2.13 | 366.0  | 0.06 | -1.41 | 197.7  | 0.03 | -0.59 | 0.36 | 0.92 | 1.00 |
| -15 | 150 | 2.13 | 180.0  | 0.04 | 0.38 | 366.0  | 0.03 | -1.93 | 0.27 | 0.95 | 1.00 |
| -15 | 170 | 2.12 | 366.0  | 0.04 | 2.47 | 180.0  | 0.02 | 1.00 | 0.24 | 0.94 | 1.00 |
| -15 | 190 | 2.10 | 361.3  | 0.03 | 1.55 | 232.7  | 0.01 | -1.46 | 0.16 | 0.91 | 1.00 |
| -15 | 210 | 2.13 | 184.2  | 0.04 | -0.95 | 366.0  | 0.03 | -0.91 | 0.22 | 0.93 | 1.00 |
| -15 | 230 | 2.15 | 366.0  | 0.03 | -1.79 | 180.0  | 0.02 | -1.22 | 0.20 | 0.95 | 1.00 |
| -15 | 250 | 2.15 | 366.0  | 0.05 | 2.15 | 236.4  | 0.01 | 0.98 | 0.28 | 0.98 | 1.00 |

| | | | | | | | | | | | |
|---|---|---|---|---|---|---|---|---|---|---|---|
| | | | 0 | | | 1 | | | | | |
| -15 | 270 | 2.14 | 366.0366.00 | 0.06 | 1.78 | 233.12233.14 | 0.01 | -0.89 | 0.34 | 0.95 | 1.00 |
| -15 | 290 | 2.11 | 307.7307.73 | 0.02 | 2.68 | 180.0180.00 | 0.01 | -1.37 | 0.05 | 0.93 | 1.00 |
| -15 | 310 | 2.11 | 361.8361.80 | 0.03 | -1.17 | 231.3231.28 | 0.01 | -1.19 | 0.11 | 0.91 | 1.00 |
| -15 | 330 | 2.13 | 274.6274.63 | 0.04 | -1.26 | 187.5187.46 | 0.01 | 2.33 | 0.20 | 0.96 | 1.00 |
| -15 | 350 | 2.14 | 348.8348.75 | 0.03 | -3.09 | 204.7204.71 | 0.01 | 0.74 | 0.22 | 0.98 | 1.00 |
| -5 | 10 | 2.06 | 180.0180.00 | 0.04 | -0.97 | 366.0366.00 | 0.04 | 2.47 | 0.21 | 0.82 | 1.00 |
| -5 | 30 | 2.05 | 180.0180.00 | 0.04 | -1.16 | 366.0366.00 | 0.01 | 2.80 | 0.15 | 0.82 | 1.00 |
| -5 | 50 | 2.04 | 180.0180.00 | 0.03 | -1.42 | 366.0366.00 | 0.01 | 2.06 | 0.08 | 0.83 | 1.00 |
| -5 | 70 | 2.04 | 306.3306.33 | 0.03 | 2.46 | 180.0180.00 | 0.02 | -1.17 | 0.13 | 0.81 | 1.00 |
| -5 | 90 | 2.04 | 249.9249.92 | 0.04 | 3.00 | 180.0180.00 | 0.02 | -0.60 | 0.18 | 0.77 | 0.99 |
| -5 | 110 | 2.04 | 205.6205.64 | 0.05 | -1.80 | 366.0366.00 | 0.02 | 1.41 | 0.17 | 0.82 | 0.97 |
| -5 | 130 | 2.05 | 181.4181.40 | 0.05 | -1.01 | 366.0366.00 | 0.02 | -1.06 | 0.20 | 0.82 | 1.00 |
| -5 | 150 | 2.04 | 187.0186.99 | 0.04 | -0.83 | 366.0366.00 | 0.01 | -2.84 | 0.21 | 0.90 | 1.00 |
| -5 | 170 | 2.05 | 310.1310.06 | 0.05 | -3.01 | 180.0180.00 | 0.01 | 0.28 | 0.29 | 0.88 | 1.00 |
| -5 | 190 | 2.05 | 299.8299.80 | 0.05 | 3.04 | 215.0214.96 | 0.01 | 0.12 | 0.20 | 0.85 | 1.00 |
| -5 | 210 | 2.05 | 194.9194.92 | 0.05 | -1.24 | 366.0366.00 | 0.02 | -1.62 | 0.19 | 0.86 | 1.00 |
| -5 | 230 | 2.05 | 180.0180.00 | 0.05 | -1.06 | 366.0366.00 | 0.02 | -2.40 | 0.25 | 0.85 | 1.00 |
| -5 | 250 | 2.05 | 227.6227.55 | 0.04 | -2.41 | 366.0366.00 | 0.02 | 2.65 | 0.17 | 0.86 | 1.00 |
| -5 | 270 | 2.06 | 249.5249.46 | 0.04 | -2.43 | 180.0180.00 | 0.02 | -0.24 | 0.20 | 0.83 | 1.00 |
| -5 | 290 | 2.04 | 180.0180.00 | 0.04 | -0.39 | 366.0366.00 | 0.02 | 2.79 | 0.14 | 0.85 | 1.00 |
| -5 | 310 | 2.03 | 267.6267.64 | 0.03 | -0.91 | 180.0180.00 | 0.02 | -0.85 | 0.10 | 0.84 | 1.00 |
| -5 | 330 | 2.04 | 314.3314.26 | 0.04 | -2.05 | 180.0180.00 | 0.02 | -0.65 | 0.14 | 0.84 | 1.00 |
| -5 | 350 | 2.06 | 345.5345.49 | 0.05 | 2.99 | 180.0180.00 | 0.02 | -0.82 | 0.21 | 0.84 | 1.00 |
| 5 | 10 | 2.05 | 349.7349.6 | 0.07 | 1.84 | 180.0180.0 | 0.02 | -1.05 | 0.30 | 0.79 | 0.99 |

| | | | | | | | | | | | |
|---|---|---|---|---|---|---|---|---|---|---|---|
| | | |  | | |  | | | | | |
| 5 | 30 | 2.03 | 297.9 | 0.06 | 2.16 | 180.0 | 0.03 | -0.58 | 0.29 | 0.82 | 0.99 |
| 5 | 50 | 2.02 | 265.8 | 0.06 | 2.50 | 180.0 | 0.03 | -0.47 | 0.30 | 0.74 | 1.00 |
| 5 | 70 | 2.04 | 269.0 | 0.08 | 2.47 | 180.0 | 0.03 | -0.58 | 0.35 | 0.73 | 0.98 |
| 5 | 90 | 2.06 | 275.6 | 0.09 | 2.50 | 180.0 | 0.03 | -0.63 | 0.43 | 0.78 | 0.99 |
| 5 | 110 | 2.06 | 266.2 | 0.09 | 2.70 | 180.0 | 0.02 | -0.58 | 0.44 | 0.82 | 1.00 |
| 5 | 130 | 2.05 | 234.5 | 0.06 | -2.66 | 366.0 | 0.03 | 0.48 | 0.33 | 0.83 | 1.00 |
| 5 | 150 | 2.03 | 232.7 | 0.07 | -3.07 | 180.0 | 0.02 | 0.20 | 0.32 | 0.81 | 0.99 |
| 5 | 170 | 2.08 | 317.1 | 0.05 | 2.62 | 180.0 | 0.02 | -0.03 | 0.28 | 0.87 | 1.00 |
| 5 | 190 | 2.09 | 366.0 | 0.05 | 1.94 | 180.0 | 0.01 | 0.35 | 0.33 | 0.98 | 1.00 |
| 5 | 210 | 2.05 | 235.5 | 0.04 | -2.67 | 366.0 | 0.01 | 0.80 | 0.26 | 0.95 | 1.00 |
| 5 | 230 | 2.03 | 220.1 | 0.06 | -2.87 | 180.0 | 0.02 | 0.04 | 0.28 | 0.86 | 1.00 |
| 5 | 250 | 2.05 | 241.5 | 0.05 | 3.00 | 180.0 | 0.02 | -0.27 | 0.22 | 0.80 | 1.00 |
| 5 | 270 | 2.07 | 249.5 | 0.04 | 2.85 | 180.0 | 0.03 | 0.03 | 0.20 | 0.89 | 1.00 |
| 5 | 290 | 2.06 | 240.1 | 0.04 | 2.81 | 180.0 | 0.04 | -0.03 | 0.22 | 0.89 | 1.00 |
| 5 | 310 | 2.03 | 253.2 | 0.04 | 2.61 | 180.0 | 0.02 | -0.09 | 0.16 | 0.85 | 1.00 |
| 5 | 330 | 2.04 | 258.3 | 0.04 | 2.70 | 180.0 | 0.02 | -0.20 | 0.19 | 0.85 | 1.00 |
| 5 | 350 | 2.06 | 344.6 | 0.06 | 1.97 | 180.0 | 0.02 | -1.03 | 0.31 | 0.82 | 1.00 |
| 15 | 10 | 2.12 | 366.0 | 0.08 | 1.57 | 228.0 | 0.02 | 1.69 | 0.57 | 0.96 | 1.00 |
| 15 | 30 | 2.10 | 366.0 | 0.07 | 1.55 | 242.9 | 0.01 | 0.64 | 0.47 | 0.95 | 1.00 |
| 15 | 50 | 2.09 | 360.9 | 0.07 | 1.51 | 180.0 | 0.01 | -0.56 | 0.42 | 0.93 | 1.00 |
| 15 | 70 | 2.12 | 352.0 | 0.07 | 1.58 | 180.0 | 0.01 | -0.59 | 0.48 | 0.93 | 1.00 |
| 15 | 90 | 2.15 | 362.7 | 0.08 | 1.65 | 180.0 | 0.01 | -0.27 | 0.57 | 0.96 | 1.00 |
| 15 | 110 | 2.15 | 366.0 | 0.09 | 1.68 | 180.0 | 0.01 | 0.85 | 0.50 | 0.91 | 1.00 |
| 15 | 130 | 2.12 | 366.0 | 0.08 | 1.62 | 180.0 | 0.01 | 1.19 | 0.41 | 0.88 | 1.00 |

| | | | | | | | | | | | |
|---|---|---|---|---|---|---|---|---|---|---|---|
| | | |  | | |  | | | | | |
| 15 | 150 | 2.10 | 291.0 | 0.06 | 2.26 | 180.0 | 0.01 | -0.34 | 0.39 | 0.93 | 1.00 |
| 15 | 170 | 2.16 | 366.0 | 0.05 | 1.92 | 180.0 | 0.01 | 0.42 | 0.29 | 0.96 | 1.00 |
| 15 | 190 | 2.17 | 366.0 | 0.06 | 1.64 | 194.9 | 0.02 | 1.18 | 0.39 | 0.97 | 1.00 |
| 15 | 210 | 2.14 | 363.2 | 0.08 | 1.55 | 246.2 | 0.01 | 1.33 | 0.58 | 0.97 | 1.00 |
| 15 | 230 | 2.11 | 333.4 | 0.09 | 1.81 | 180.0 | 0.02 | -1.56 | 0.59 | 0.94 | 1.00 |
| 15 | 250 | 2.13 | 356.7 | 0.06 | 1.41 | 180.0 | 0.01 | -2.44 | 0.41 | 0.95 | 1.00 |
| 15 | 270 | 2.16 | 366.0 | 0.05 | 1.52 | 218.7 | 0.01 | 0.80 | 0.30 | 0.98 | 1.00 |
| 15 | 290 | 2.14 | 318.5 | 0.05 | 2.00 | 180.0 | 0.02 | -0.91 | 0.39 | 0.97 | 1.00 |
| 15 | 310 | 2.11 | 273.7 | 0.06 | 2.45 | 180.0 | 0.01 | -1.04 | 0.45 | 0.96 | 1.00 |
| 15 | 330 | 2.12 | 298.9 | 0.07 | 2.02 | 180.0 | 0.01 | -1.51 | 0.52 | 0.97 | 1.00 |
| 15 | 350 | 2.13 | 357.1 | 0.08 | 1.66 | 213.6 | 0.02 | 2.61 | 0.65 | 0.98 | 1.00 |
| 25 | 10 | 2.20 | 366.0 | 0.07 | 1.44 | 199.6 | 0.03 | 1.46 | 0.36 | 0.93 | 1.00 |
| 25 | 30 | 2.18 | 366.0 | 0.06 | 1.39 | 193.1 | 0.04 | 1.15 | 0.40 | 0.94 | 1.00 |
| 25 | 50 | 2.18 | 366.0 | 0.05 | 1.40 | 190.7 | 0.04 | 1.12 | 0.42 | 0.97 | 1.00 |
| 25 | 70 | 2.20 | 180.0 | 0.05 | 1.49 | 366.0 | 0.05 | 1.46 | 0.50 | 0.99 | 1.00 |
| 25 | 90 | 2.24 | 366.0 | 0.07 | 1.67 | 187.0 | 0.05 | 1.07 | 0.50 | 0.96 | 1.00 |
| 25 | 110 | 2.21 | 366.0 | 0.07 | 1.87 | 191.2 | 0.05 | 1.18 | 0.53 | 0.96 | 1.00 |
| 25 | 130 | 2.18 | 366.0 | 0.07 | 1.81 | 188.9 | 0.04 | 1.31 | 0.49 | 0.94 | 1.00 |
| 25 | 150 | 2.17 | 366.0 | 0.05 | 1.86 | 182.8 | 0.02 | 1.11 | 0.42 | 0.97 | 1.00 |
| 25 | 170 | 2.22 | 366.0 | 0.06 | 1.80 | 184.7 | 0.03 | 1.11 | 0.40 | 0.96 | 1.00 |
| 25 | 190 | 2.24 | 366.0 | 0.07 | 1.66 | 194.9 | 0.05 | 1.13 | 0.49 | 0.97 | 1.00 |
| 25 | 210 | 2.22 | 366.0 | 0.10 | 1.54 | 191.7 | 0.03 | 1.30 | 0.68 | 0.97 | 1.00 |
| 25 | 230 | 2.21 | 366.0 | 0.11 | 1.50 | 201.9 | 0.01 | 1.89 | 0.71 | 0.97 | 1.00 |
| 25 | 250 | 2.23 | 366.0 | 0.08 | 1.30 | 187.5 | 0.03 | 1.79 | 0.56 | 0.96 | 1.00 |

| | | | | | | | | | | | |
|---|---|---|---|---|---|---|---|---|---|---|---|
| | | | θ | | | 6 | | | | | |
| 25 | 270 | 2.25 | 366.0366.00 | 0.06 | 1.40 | 191.7191.65 | 0.03 | 1.31 | 0.39 | 0.96 | 1.00 |
| 25 | 290 | 2.22 | 366.0366.00 | 0.05 | 1.76 | 192.1192.12 | 0.02 | 1.18 | 0.25 | 0.96 | 1.00 |
| 25 | 310 | 2.19 | 366.0366.00 | 0.05 | 1.87 | 200.5200.51 | 0.01 | 2.19 | 0.34 | 0.99 | 1.00 |
| 25 | 330 | 2.20 | 366.0366.00 | 0.06 | 1.59 | 199.1199.11 | 0.01 | 2.35 | 0.43 | 0.98 | 1.00 |
| 25 | 350 | 2.22 | 366.0366.00 | 0.07 | 1.52 | 206.6206.57 | 0.02 | 1.85 | 0.41 | 0.93 | 1.00 |
| 35 | 10 | 2.23 | 366.0366.00 | 0.08 | 1.49 | 196.8196.78 | 0.02 | 1.82 | 0.39 | 0.86 | 1.00 |
| 35 | 30 | 2.22 | 366.0366.00 | 0.08 | 1.51 | 195.4195.38 | 0.03 | 1.62 | 0.41 | 0.89 | 1.00 |
| 35 | 50 | 2.22 | 366.0366.00 | 0.08 | 1.49 | 198.2198.18 | 0.02 | 1.20 | 0.33 | 0.88 | 1.00 |
| 35 | 70 | 2.23 | 366.0366.00 | 0.08 | 1.50 | 191.7191.65 | 0.03 | 0.94 | 0.39 | 0.90 | 1.00 |
| 35 | 90 | 2.24 | 366.0366.00 | 0.10 | 1.67 | 191.2191.19 | 0.03 | 0.81 | 0.52 | 0.87 | 1.00 |
| 35 | 110 | 2.21 | 366.0366.00 | 0.12 | 1.73 | 199.1199.11 | 0.03 | 0.73 | 0.58 | 0.90 | 1.00 |
| 35 | 130 | 2.20 | 366.0366.00 | 0.12 | 1.72 | 199.6199.58 | 0.02 | 0.73 | 0.61 | 0.93 | 1.00 |
| 35 | 150 | 2.20 | 366.0366.00 | 0.11 | 1.70 | 194.5194.45 | 0.02 | 0.70 | 0.62 | 0.94 | 1.00 |
| 35 | 170 | 2.23 | 366.0366.00 | 0.11 | 1.66 | 202.4202.38 | 0.02 | 0.69 | 0.61 | 0.92 | 1.00 |
| 35 | 190 | 2.24 | 366.0366.00 | 0.12 | 1.64 | 199.6199.58 | 0.03 | 1.03 | 0.62 | 0.92 | 1.00 |
| 35 | 210 | 2.23 | 366.0366.00 | 0.14 | 1.56 | 193.1193.05 | 0.03 | 1.28 | 0.73 | 0.94 | 1.00 |
| 35 | 230 | 2.23 | 366.0366.00 | 0.14 | 1.55 | 197.3197.25 | 0.02 | 1.42 | 0.75 | 0.95 | 1.00 |
| 35 | 250 | 2.24 | 366.0366.00 | 0.13 | 1.49 | 201.4201.44 | 0.02 | 1.58 | 0.70 | 0.94 | 1.00 |
| 35 | 270 | 2.25 | 366.0366.00 | 0.11 | 1.51 | 194.9194.92 | 0.01 | 1.64 | 0.59 | 0.93 | 1.00 |
| 35 | 290 | 2.23 | 366.0366.00 | 0.07 | 1.75 | 200.1200.05 | 0.01 | 1.46 | 0.37 | 0.92 | 1.00 |
| 35 | 310 | 2.21 | 366.0366.00 | 0.07 | 1.84 | 180.0180.00 | 0.02 | 3.13 | 0.42 | 0.92 | 1.00 |
| 35 | 330 | 2.23 | 366.0366.00 | 0.09 | 1.71 | 180.0180.00 | 0.02 | -2.81 | 0.47 | 0.91 | 1.00 |
| 35 | 350 | 2.24 | 366.0366.00 | 0.10 | 1.58 | 185.1185.13 | 0.02 | 3.09 | 0.46 | 0.85 | 1.00 |
| 45 | 10 | 2.20 | 324.5324.5 | 0.19 | 2.02 | 180.0180.0 | 0.05 | -1.61 | 0.84 | 0.91 | 1.00 |

| | | | 1 | | | 0 | | | | | |
|---|---|---|---|---|---|---|---|---|---|---|---|
| 45 | 30 | 2.20 | 315.2~315.19~ | 0.17 | 2.13 | 180.0~180.00~ | 0.04 | -1.48 | 0.81 | 0.89 | 1.00 |
| 45 | 50 | 2.20 | 304.5~304.47~ | 0.16 | 2.22 | 180.0~180.00~ | 0.04 | -1.47 | 0.80 | 0.89 | 1.00 |
| 45 | 70 | 2.20 | 310.5~310.53~ | 0.17 | 2.18 | 180.0~180.00~ | 0.04 | -1.27 | 0.83 | 0.92 | 1.00 |
| 45 | 90 | 2.21 | 311.0~310.99~ | 0.19 | 2.22 | 180.0~180.00~ | 0.04 | -1.13 | 0.85 | 0.90 | 1.00 |
| 45 | 110 | 2.20 | 316.6~316.59~ | 0.20 | 2.16 | 180.0~180.00~ | 0.04 | -1.19 | 0.86 | 0.92 | 1.00 |
| 45 | 130 | 2.19 | 311.9~311.92~ | 0.20 | 2.21 | 180.0~180.00~ | 0.04 | -1.27 | 0.86 | 0.92 | 1.00 |
| 45 | 150 | 2.19 | 328.2~328.24~ | 0.20 | 2.02 | 180.0~180.00~ | 0.04 | -1.29 | 0.88 | 0.94 | 1.00 |
| 45 | 170 | 2.19 | 330.6~330.57~ | 0.20 | 1.99 | 180.0~180.00~ | 0.04 | -1.39 | 0.89 | 0.95 | 1.00 |
| 45 | 190 | 2.19 | 344.6~344.56~ | 0.21 | 1.80 | 180.0~180.00~ | 0.04 | -1.55 | 0.89 | 0.95 | 1.00 |
| 45 | 210 | 2.19 | 353.4~353.41~ | 0.23 | 1.72 | 180.0~180.00~ | 0.04 | -1.40 | 0.91 | 0.95 | 1.00 |
| 45 | 230 | 2.18 | 354.4~354.35~ | 0.24 | 1.68 | 180.0~180.00~ | 0.04 | -1.46 | 0.90 | 0.96 | 1.00 |
| 45 | 250 | 2.20 | 351.1~351.08~ | 0.25 | 1.68 | 180.0~180.00~ | 0.04 | -1.55 | 0.90 | 0.93 | 0.99 |
| 45 | 270 | 2.20 | 348.3~348.29~ | 0.22 | 1.74 | 180.0~180.00~ | 0.04 | -1.60 | 0.88 | 0.94 | 1.00 |
| 45 | 290 | 2.19 | 341.3~341.29~ | 0.18 | 1.88 | 180.0~180.00~ | 0.04 | -1.53 | 0.84 | 0.94 | 1.00 |
| 45 | 310 | 2.18 | 332.9~332.90~ | 0.17 | 2.01 | 180.0~180.00~ | 0.04 | -1.79 | 0.83 | 0.94 | 1.00 |
| 45 | 330 | 2.19 | 327.8~327.77~ | 0.18 | 2.04 | 180.0~180.00~ | 0.04 | -1.67 | 0.84 | 0.93 | 1.00 |
| 45 | 350 | 2.21 | 330.6~330.57~ | 0.19 | 1.98 | 180.0~180.00~ | 0.05 | -1.59 | 0.84 | 0.91 | 1.00 |
| ~55~51 | 10 | 2.20 | 285.4~285.35~ | 0.26 | 2.50 | 180.0~180.00~ | 0.06 | -1.41 | 0.81 | 0.78 | 0.98 |
| ~55~51 | 30 | 2.20 | 279.8~279.76~ | 0.25 | 2.54 | 180.0~180.00~ | 0.05 | -1.55 | 0.83 | 0.78 | 0.98 |
| ~55~51 | 50 | 2.21 | 279.3~279.29~ | 0.24 | 2.57 | 180.0~180.00~ | 0.05 | -1.66 | 0.82 | 0.80 | 0.98 |
| ~55~51 | 70 | 2.21 | 282.1~282.09~ | 0.23 | 2.56 | 180.0~180.00~ | 0.05 | -1.43 | 0.83 | 0.82 | 0.99 |
| ~55~51 | 90 | 2.22 | 280.2~280.23~ | 0.25 | 2.59 | 180.0~180.00~ | 0.05 | -1.38 | 0.84 | 0.79 | 0.99 |
| ~55~51 | 110 | 2.21 | 276.5~276.50~ | 0.25 | 2.65 | 180.0~180.00~ | 0.05 | -1.33 | 0.81 | 0.78 | 0.98 |
| ~55~51 | 130 | 2.20 | 274.6~274.6~ | 0.25 | 2.68 | 180.0~180.0~ | 0.05 | -1.45 | 0.82 | 0.78 | 0.98 |

| | | | | | | | | | | | |
|---|---|---|---|---|---|---|---|---|---|---|---|
| | | | 3 | | | 0 | | | | | |
| 51 | 150 | 2.20 | 278.8 | 0.25 | 2.60 | 180.0 | 0.05 | -1.45 | 0.84 | 0.81 | 0.98 |
| 51 | 170 | 2.20 | 276.0 | 0.26 | 2.64 | 180.0 | 0.05 | -1.55 | 0.82 | 0.78 | 0.98 |
| 51 | 190 | 2.19 | 284.9 | 0.26 | 2.48 | 180.0 | 0.06 | -1.60 | 0.85 | 0.78 | 0.98 |
| 51 | 210 | 2.18 | 302.6 | 0.28 | 2.25 | 180.0 | 0.07 | -1.51 | 0.88 | 0.83 | 0.99 |
| 51 | 230 | 2.17 | 316.6 | 0.31 | 2.05 | 180.0 | 0.08 | -1.57 | 0.90 | 0.83 | 0.99 |
| 51 | 250 | 2.18 | 319.9 | 0.33 | 2.01 | 180.0 | 0.08 | -1.58 | 0.88 | 0.80 | 0.98 |
| 51 | 270 | 2.17 | 323.1 | 0.30 | 2.01 | 180.0 | 0.08 | -1.57 | 0.90 | 0.84 | 0.99 |
| 51 | 290 | 2.18 | 311.0 | 0.27 | 2.14 | 180.0 | 0.07 | -1.73 | 0.88 | 0.86 | 0.98 |
| 51 | 310 | 2.18 | 294.2 | 0.25 | 2.39 | 180.0 | 0.06 | -1.77 | 0.84 | 0.82 | 0.96 |
| 51 | 330 | 2.17 | 291.9 | 0.25 | 2.43 | 180.0 | 0.05 | -1.63 | 0.85 | 0.79 | 0.99 |
| 51 | 350 | 2.19 | 287.7 | 0.26 | 2.47 | 180.0 | 0.06 | -1.58 | 0.84 | 0.78 | 0.98 |

Table 2 Shown are the mean difference per hour and its variance for the OH* equivalent BV frequency measured during the same night for the different latitudinal intervals (-45°±5° until +45°±5°, -51±1°, and -51±1°).

| Latitude | Mean hourly difference [× $10^{-2}$] | Variance [× $10^{-8}$] |
| --- | --- | --- |
| -51 | 0.02 | 1.5 |
| -45 | 0.02 | 1.6 |
| -35 | 0.03 | 6.3 |
| -25 | 0.03 | 4.8 |
| -15 | 0.03 | 4.8 |
| -5 | 0.04 | 6.5 |
| 5 | 0.04 | 6.4 |
| 15 | 0.03 | 4.5 |
| 25 | 0.03 | 4.8 |
| 35 | 0.03 | 7.2 |
| 45 | 0.02 | 1.6 |
| 51 | 0.01 | 1.3 |

**Appendix**

Table 23 Same as table 1 but for the OH*-layer height. Please pay attention to the fact that OH*-layer height is less variable during the year and therefore the fraction of data around the approximation is provided for intervals of +/- 1 % and +/- 2 %.

| Lat. [°] | Lon. [°] | Mean [$10^{-2}$ s$^{-1}$] | 1st oscillation Period [d] | Amp. [$10^{-2}$ s$^{-1}$] | Phase [rad] | 2nd oscillation Period [d] | Amp. [$10^{-2}$ s$^{-1}$] | Phase [rad] | Quality of approxi-mation | Fraction of data +-1% around approx. | +/- 2% around approx. |
|---|---|---|---|---|---|---|---|---|---|---|---|
| -55 -51 | 10 | 86.43 | 340.4 340.36 | 1.30 | -1.51 | 187.9 187.92 | 0.32 | 0.20 | 0.78 | 0.90 | 1.00 |
| -55 -51 | 30 | 86.43 | 358.5 358.54 | 1.45 | -1.57 | 208.9 208.90 | 0.38 | -0.77 | 0.79 | 0.93 | 0.99 |
| -55 -51 | 50 | 86.39 | 361.3 361.34 | 1.43 | -1.58 | 194.9 194.92 | 0.30 | -0.16 | 0.80 | 0.91 | 0.99 |
| -55 -51 | 70 | 86.20 | 347.8 347.82 | 1.27 | -1.46 | 187.9 187.92 | 0.24 | 0.59 | 0.67 | 0.88 | 0.98 |
| -55 -51 | 90 | 86.16 | 355.7 355.74 | 1.34 | -1.45 | 213.6 213.56 | 0.26 | -0.49 | 0.74 | 0.92 | 0.98 |
| -55 -51 | 110 | 86.16 | 362.3 362.27 | 1.52 | -1.51 | 216.8 216.83 | 0.30 | -0.89 | 0.80 | 0.93 | 0.99 |
| -55 -51 | 130 | 86.14 | 366.0 366.00 | 1.69 | -1.51 | 219.2 219.16 | 0.33 | -1.06 | 0.85 | 0.93 | 0.99 |
| -55 -51 | 150 | 86.07 | 354.8 354.81 | 1.47 | -1.41 | 210.8 210.77 | 0.38 | -0.58 | 0.78 | 0.91 | 0.99 |
| -55 -51 | 170 | 86.04 | 355.3 355.28 | 1.43 | -1.47 | 208.0 207.97 | 0.27 | -0.42 | 0.78 | 0.90 | 0.98 |
| -55 -51 | 190 | 86.03 | 351.1 351.08 | 1.41 | -1.40 | 203.3 203.31 | 0.42 | -0.36 | 0.78 | 0.89 | 0.99 |
| -55 -51 | 210 | 86.11 | 355.3 355.28 | 1.44 | -1.47 | 196.3 196.32 | 0.39 | -0.15 | 0.79 | 0.89 | 0.99 |
| -55 -51 | 230 | 86.17 | 350.6 350.62 | 1.44 | -1.57 | 180.0 180.00 | 0.29 | 0.58 | 0.78 | 0.87 | 0.98 |
| -55 -51 | 250 | 86.26 | 350.6 350.62 | 1.48 | -1.55 | 180.9 180.93 | 0.31 | 0.62 | 0.81 | 0.91 | 1.00 |
| -55 -51 | 270 | 86.29 | 338.0 338.03 | 1.29 | -1.40 | 207.0 207.04 | 0.40 | -0.67 | 0.72 | 0.85 | 0.98 |
| -55 -51 | 290 | 86.28 | 334.3 334.30 | 1.37 | -1.44 | 197.7 197.71 | 0.38 | -0.31 | 0.76 | 0.86 | 0.99 |
| -55 -51 | 310 | 86.30 | 348.3 348.29 | 1.40 | -1.59 | 183.3 183.26 | 0.33 | 0.43 | 0.80 | 0.90 | 1.00 |
| -55 -51 | 330 | 86.32 | 343.2 343.16 | 1.34 | -1.56 | 180.0 180.00 | 0.30 | 0.83 | 0.80 | 0.92 | 0.99 |
| -55 -51 | 350 | 86.31 | 295.1 295.14 | 1.15 | -1.08 | 181.9 181.86 | 0.27 | 0.32 | 0.70 | 0.89 | 0.98 |
| -45 | 10 | 86.37 | 257.4 257.3 | 1.04 | -0.40 | 180.0 180.0 | 0.46 | 0.09 | 0.73 | 0.90 | 1.00 |

| | | | | | | | | | | | |
|---|---|---|---|---|---|---|---|---|---|---|---|
| | | | 8 | | | 0 | | | | | |
| -45 | 30 | 86.39 | 274.2 | 1.09 | -0.64 | 180.0 | 0.48 | 0.11 | 0.76 | 0.91 | 1.00 |
| -45 | 50 | 86.38 | 292.4 | 1.07 | -0.80 | 180.0 | 0.51 | 0.32 | 0.77 | 0.91 | 1.00 |
| -45 | 70 | 86.26 | 261.1 | 0.98 | -0.35 | 180.0 | 0.42 | 0.15 | 0.71 | 0.92 | 1.00 |
| -45 | 90 | 86.23 | 273.7 | 1.02 | -0.50 | 180.0 | 0.37 | 0.28 | 0.76 | 0.95 | 1.00 |
| -45 | 110 | 86.28 | 304.9 | 1.11 | -0.94 | 180.5 | 0.46 | 0.38 | 0.82 | 0.96 | 1.00 |
| -45 | 130 | 86.34 | 349.7 | 1.40 | -1.42 | 186.5 | 0.45 | 0.22 | 0.86 | 0.95 | 1.00 |
| -45 | 150 | 86.19 | 304.5 | 1.29 | -0.94 | 180.0 | 0.51 | 0.42 | 0.79 | 0.88 | 0.99 |
| -45 | 170 | 86.06 | 270.0 | 1.19 | -0.46 | 180.0 | 0.49 | 0.22 | 0.78 | 0.90 | 1.00 |
| -45 | 190 | 86.11 | 263.4 | 1.08 | -0.34 | 180.0 | 0.59 | 0.08 | 0.72 | 0.86 | 1.00 |
| -45 | 210 | 86.20 | 272.3 | 1.10 | -0.55 | 180.0 | 0.55 | 0.14 | 0.73 | 0.89 | 1.00 |
| -45 | 230 | 86.24 | 280.7 | 1.01 | -0.70 | 180.0 | 0.44 | 0.27 | 0.73 | 0.91 | 1.00 |
| -45 | 250 | 86.25 | 284.4 | 0.99 | -0.74 | 180.0 | 0.41 | 0.33 | 0.72 | 0.93 | 1.00 |
| -45 | 270 | 86.32 | 272.8 | 1.00 | -0.65 | 180.0 | 0.42 | 0.12 | 0.69 | 0.88 | 1.00 |
| -45 | 290 | 86.21 | 290.5 | 1.22 | -0.92 | 180.0 | 0.45 | 0.30 | 0.75 | 0.87 | 1.00 |
| -45 | 310 | 86.20 | 262.1 | 1.09 | -0.45 | 180.0 | 0.46 | 0.20 | 0.71 | 0.87 | 1.00 |
| -45 | 330 | 86.26 | 252.7 | 0.97 | -0.28 | 180.0 | 0.42 | 0.25 | 0.69 | 0.90 | 1.00 |
| -45 | 350 | 86.30 | 249.0 | 0.95 | -0.15 | 180.0 | 0.44 | 0.20 | 0.70 | 0.92 | 1.00 |
| -35 | 10 | 86.47 | 240.1 | 0.82 | -0.98 | 366.0 | 0.70 | -0.94 | 0.65 | 0.88 | 1.00 |
| -35 | 30 | 86.52 | 259.7 | 0.96 | -0.28 | 180.0 | 0.72 | -0.04 | 0.70 | 0.88 | 1.00 |
| -35 | 50 | 86.55 | 254.1 | 0.95 | -0.13 | 180.0 | 0.72 | -0.04 | 0.67 | 0.87 | 1.00 |
| -35 | 70 | 86.51 | 235.5 | 0.79 | -0.87 | 366.0 | 0.63 | -0.94 | 0.63 | 0.87 | 1.00 |
| -35 | 90 | 86.51 | 240.6 | 0.72 | -0.85 | 366.0 | 0.54 | -0.91 | 0.63 | 0.91 | 1.00 |
| -35 | 110 | 86.61 | 254.6 | 0.85 | -0.23 | 180.0 | 0.53 | -0.03 | 0.67 | 0.90 | 1.00 |
| -35 | 130 | 86.70 | 321.3 | 1.04 | -1.15 | 180.0 | 0.51 | 0.35 | 0.78 | 0.93 | 1.00 |

| | | | | | | | | | | | |
|---|---|---|---|---|---|---|---|---|---|---|---|
| | | | 5 | | | 0 | | | | | |
| -35 | 150 | 86.59 | 271.8 | 1.15 | -0.48 | 180.0 | 0.65 | 0.17 | 0.75 | 0.87 | 1.00 |
| -35 | 170 | 86.33 | 248.5 | 1.05 | 0.01 | 180.0 | 0.73 | 0.04 | 0.71 | 0.88 | 1.00 |
| -35 | 190 | 86.31 | 235.5 | 0.86 | 0.18 | 180.0 | 0.71 | -0.12 | 0.69 | 0.90 | 1.00 |
| -35 | 210 | 86.49 | 243.9 | 0.85 | -0.04 | 180.0 | 0.71 | -0.15 | 0.69 | 0.91 | 1.00 |
| -35 | 230 | 86.55 | 262.1 | 0.78 | -0.35 | 180.0 | 0.61 | -0.05 | 0.63 | 0.89 | 1.00 |
| -35 | 250 | 86.52 | 255.1 | 0.72 | -0.17 | 180.0 | 0.57 | -0.06 | 0.56 | 0.89 | 0.99 |
| -35 | 270 | 86.56 | 270.9 | 0.73 | -0.49 | 180.0 | 0.56 | -0.09 | 0.57 | 0.88 | 0.99 |
| -35 | 290 | 86.33 | 328.2 | 1.12 | -1.20 | 180.0 | 0.63 | 0.47 | 0.78 | 0.90 | 1.00 |
| -35 | 310 | 86.26 | 256.0 | 1.09 | -0.25 | 180.0 | 0.69 | 0.04 | 0.70 | 0.87 | 1.00 |
| -35 | 330 | 86.26 | 216.4 | 0.89 | -0.27 | 366.0 | 0.53 | -0.98 | 0.69 | 0.90 | 1.00 |
| -35 | 350 | 86.35 | 222.9 | 0.79 | -0.43 | 366.0 | 0.51 | -0.96 | 0.61 | 0.88 | 1.00 |
| -25 | 10 | 86.57 | 183.3 | 0.44 | -0.49 | 366.0 | 0.38 | -1.70 | 0.37 | 0.89 | 1.00 |
| -25 | 30 | 86.71 | 366.0 | 0.58 | -1.55 | 196.8 | 0.48 | -0.99 | 0.51 | 0.92 | 1.00 |
| -25 | 50 | 86.74 | 366.0 | 0.60 | -1.60 | 196.3 | 0.52 | -0.99 | 0.46 | 0.86 | 1.00 |
| -25 | 70 | 86.73 | 261.1 | 0.44 | -0.78 | 183.7 | 0.43 | -0.90 | 0.26 | 0.79 | 0.99 |
| -25 | 90 | 86.69 | 264.8 | 0.39 | -1.05 | 186.1 | 0.30 | -1.02 | 0.23 | 0.83 | 0.99 |
| -25 | 110 | 86.69 | 340.4 | 0.44 | -1.59 | 198.7 | 0.32 | -1.19 | 0.28 | 0.85 | 0.99 |
| -25 | 130 | 86.86 | 366.0 | 0.74 | -1.69 | 201.4 | 0.33 | -1.18 | 0.44 | 0.85 | 1.00 |
| -25 | 150 | 86.84 | 366.0 | 0.81 | -1.56 | 187.9 | 0.40 | -0.32 | 0.51 | 0.83 | 0.99 |
| -25 | 170 | 86.62 | 366.0 | 0.42 | -1.78 | 180.0 | 0.41 | 0.05 | 0.36 | 0.87 | 1.00 |
| -25 | 190 | 86.45 | 186.1 | 0.42 | -0.57 | 366.0 | 0.09 | -2.31 | 0.29 | 0.94 | 1.00 |
| -25 | 210 | 86.69 | 184.7 | 0.53 | -0.70 | 366.0 | 0.19 | -1.66 | 0.40 | 0.92 | 1.00 |
| -25 | 230 | 86.84 | 180.0 | 0.55 | -0.59 | 366.0 | 0.39 | -1.60 | 0.44 | 0.89 | 1.00 |
| -25 | 250 | 86.81 | 180.0 | 0.45 | -0.72 | 366.0 | 0.32 | -1.81 | 0.34 | 0.90 | 1.00 |

| | | | | | | | | | | |
|---|---|---|---|---|---|---|---|---|---|---|
| | | | 0 | | | 0 | | | | |
| -25 | 270 | 86.77 | 180.0 | 0.40 | -0.92 | 366.0 | 0.34 | -2.02 | 0.30 | 0.88 | 1.00 |
| -25 | 290 | 86.62 | 366.0 | 0.66 | -1.84 | 195.9 | 0.41 | -1.32 | 0.46 | 0.87 | 1.00 |
| -25 | 310 | 86.52 | 366.0 | 0.68 | -1.67 | 197.3 | 0.48 | -1.17 | 0.48 | 0.86 | 1.00 |
| -25 | 330 | 86.49 | 188.4 | 0.46 | -0.61 | 366.0 | 0.38 | -1.47 | 0.32 | 0.84 | 1.00 |
| -25 | 350 | 86.48 | 187.5 | 0.47 | -0.42 | 366.0 | 0.35 | -1.50 | 0.38 | 0.89 | 1.00 |
| -15 | 10 | 86.33 | 187.5 | 0.88 | -1.79 | 339.0 | 0.33 | -2.81 | 0.66 | 0.92 | 1.00 |
| -15 | 30 | 86.40 | 180.0 | 0.89 | -1.50 | 338.0 | 0.29 | -2.47 | 0.63 | 0.93 | 1.00 |
| -15 | 50 | 86.43 | 180.0 | 0.81 | -1.50 | 366.0 | 0.33 | -2.17 | 0.59 | 0.91 | 1.00 |
| -15 | 70 | 86.35 | 180.0 | 0.56 | -1.56 | 366.0 | 0.35 | -2.02 | 0.37 | 0.84 | 1.00 |
| -15 | 90 | 86.21 | 180.0 | 0.53 | -1.45 | 366.0 | 0.39 | -2.02 | 0.37 | 0.83 | 1.00 |
| -15 | 110 | 86.23 | 180.0 | 0.51 | -1.59 | 340.4 | 0.34 | -2.07 | 0.38 | 0.87 | 1.00 |
| -15 | 130 | 86.46 | 366.0 | 0.73 | -1.73 | 193.1 | 0.39 | -1.77 | 0.42 | 0.82 | 1.00 |
| -15 | 150 | 86.55 | 366.0 | 0.77 | -1.62 | 198.7 | 0.49 | -1.65 | 0.47 | 0.84 | 1.00 |
| -15 | 170 | 86.40 | 184.2 | 0.42 | -1.65 | 366.0 | 0.25 | -2.10 | 0.23 | 0.83 | 1.00 |
| -15 | 190 | 86.29 | 195.9 | 0.67 | -2.15 | 332.4 | 0.15 | -3.07 | 0.42 | 0.87 | 1.00 |
| -15 | 210 | 86.48 | 190.7 | 0.78 | -1.86 | 280.2 | 0.15 | -1.62 | 0.49 | 0.89 | 1.00 |
| -15 | 230 | 86.52 | 180.0 | 0.79 | -1.42 | 366.0 | 0.30 | -1.90 | 0.55 | 0.89 | 1.00 |
| -15 | 250 | 86.42 | 180.5 | 0.95 | -1.66 | 355.7 | 0.25 | -2.56 | 0.64 | 0.92 | 1.00 |
| -15 | 270 | 86.47 | 180.0 | 0.87 | -1.79 | 334.3 | 0.37 | -2.69 | 0.61 | 0.92 | 1.00 |
| -15 | 290 | 86.47 | 180.0 | 0.75 | -1.70 | 348.8 | 0.41 | -2.71 | 0.60 | 0.93 | 1.00 |
| -15 | 310 | 86.29 | 180.0 | 0.81 | -1.72 | 343.2 | 0.34 | -2.24 | 0.59 | 0.90 | 1.00 |
| -15 | 330 | 86.27 | 180.0 | 0.69 | -1.59 | 340.4 | 0.31 | -2.11 | 0.52 | 0.90 | 1.00 |
| -15 | 350 | 86.31 | 180.9 | 0.66 | -1.58 | 339.4 | 0.25 | -2.60 | 0.52 | 0.93 | 1.00 |
| -5 | 10 | 85.59 | 189.3 | 1.31 | -1.83 | 351.6 | 0.48 | -3.09 | 0.78 | 0.89 | 1.00 |

| | | | | | | | | | | |
|---|---|---|---|---|---|---|---|---|---|---|
| | | | 2 | | | 5 | | | | |
| -5 | 30 | 85.57 | 184.7 | 1.42 | -1.69 | 328.2 | 0.37 | -2.91 | 0.81 | 0.90 | 1.00 |
| -5 | 50 | 85.46 | 180.0 | 1.34 | -1.65 | 311.5 | 0.35 | -2.45 | 0.78 | 0.91 | 1.00 |
| -5 | 70 | 85.35 | 180.0 | 1.15 | -1.65 | 313.8 | 0.36 | -2.17 | 0.67 | 0.83 | 1.00 |
| -5 | 90 | 85.32 | 180.0 | 1.07 | -1.54 | 317.5 | 0.33 | -1.93 | 0.64 | 0.85 | 1.00 |
| -5 | 110 | 85.59 | 180.0 | 1.08 | -1.58 | 285.4 | 0.41 | -1.98 | 0.63 | 0.83 | 1.00 |
| -5 | 130 | 85.94 | 180.0 | 0.98 | -1.56 | 309.1 | 0.46 | -1.50 | 0.60 | 0.84 | 1.00 |
| -5 | 150 | 85.99 | 180.0 | 0.82 | -1.42 | 366.0 | 0.52 | -1.75 | 0.56 | 0.82 | 1.00 |
| -5 | 170 | 86.06 | 185.6 | 0.81 | -1.64 | 333.4 | 0.34 | -2.26 | 0.51 | 0.86 | 1.00 |
| -5 | 190 | 86.11 | 184.2 | 0.72 | -1.80 | 320.3 | 0.35 | -2.70 | 0.50 | 0.89 | 1.00 |
| -5 | 210 | 86.00 | 180.0 | 0.87 | -1.62 | 314.7 | 0.29 | -1.80 | 0.55 | 0.88 | 1.00 |
| -5 | 230 | 85.77 | 180.0 | 1.17 | -1.53 | 366.0 | 0.46 | -2.01 | 0.70 | 0.86 | 1.00 |
| -5 | 250 | 85.55 | 180.0 | 1.38 | -1.63 | 366.0 | 0.29 | -1.99 | 0.81 | 0.93 | 1.00 |
| -5 | 270 | 85.67 | 180.0 | 1.18 | -1.62 | 315.7 | 0.35 | -2.24 | 0.78 | 0.95 | 1.00 |
| -5 | 290 | 85.73 | 180.0 | 1.15 | -1.61 | 340.4 | 0.48 | -2.63 | 0.79 | 0.95 | 1.00 |
| -5 | 310 | 85.54 | 180.0 | 1.17 | -1.68 | 341.8 | 0.41 | -2.53 | 0.76 | 0.91 | 1.00 |
| -5 | 330 | 85.49 | 180.0 | 1.12 | -1.57 | 360.4 | 0.41 | -2.51 | 0.76 | 0.93 | 1.00 |
| -5 | 350 | 85.53 | 181.9 | 1.21 | -1.61 | 335.7 | 0.49 | -2.87 | 0.76 | 0.91 | 1.00 |
| 5 | 10 | 85.54 | 201.4 | 1.37 | -2.19 | 366.0 | 0.10 | 2.03 | 0.74 | 0.87 | 0.99 |
| 5 | 30 | 85.37 | 204.2 | 1.46 | -2.21 | 366.0 | 0.10 | 1.30 | 0.74 | 0.87 | 0.99 |
| 5 | 50 | 85.14 | 198.7 | 1.47 | -1.98 | 366.0 | 0.14 | 0.28 | 0.72 | 0.83 | 0.98 |
| 5 | 70 | 85.07 | 194.5 | 1.38 | -1.84 | 366.0 | 0.20 | -0.33 | 0.61 | 0.75 | 0.96 |
| 5 | 90 | 85.40 | 194.5 | 1.25 | -1.79 | 366.0 | 0.14 | -0.91 | 0.61 | 0.78 | 0.99 |
| 5 | 110 | 85.86 | 196.3 | 1.14 | -1.82 | 298.9 | 0.13 | -1.58 | 0.57 | 0.83 | 0.98 |
| 5 | 130 | 86.11 | 194.0 | 0.98 | -1.70 | 366.0 | 0.15 | -0.55 | 0.50 | 0.80 | 0.99 |

| | | | | | | | | | | | |
|---|---|---|---|---|---|---|---|---|---|---|---|
| | | |  | | |  | | | | | |
| 5 | 150 | 85.95 | 187.5 | 1.04 | -1.62 | 366.0 | 0.24 | -0.81 | 0.55 | 0.81 | 0.99 |
| 5 | 170 | 86.26 | 183.7 | 0.94 | -1.35 | 366.0 | 0.31 | -1.95 | 0.58 | 0.85 | 1.00 |
| 5 | 190 | 86.48 | 180.5 | 0.67 | -1.43 | 366.0 | 0.27 | -2.48 | 0.49 | 0.89 | 1.00 |
| 5 | 210 | 86.26 | 181.9 | 0.83 | -1.62 | 366.0 | 0.15 | -1.71 | 0.56 | 0.90 | 1.00 |
| 5 | 230 | 85.86 | 182.8 | 1.21 | -1.71 | 366.0 | 0.16 | -1.41 | 0.73 | 0.91 | 1.00 |
| 5 | 250 | 85.63 | 181.4 | 1.37 | -1.68 | 366.0 | 0.26 | -0.53 | 0.81 | 0.94 | 1.00 |
| 5 | 270 | 85.81 | 180.0 | 1.18 | -1.59 | 366.0 | 0.16 | -1.19 | 0.74 | 0.92 | 1.00 |
| 5 | 290 | 85.82 | 186.1 | 1.17 | -1.74 | 366.0 | 0.15 | -2.76 | 0.74 | 0.91 | 1.00 |
| 5 | 310 | 85.55 | 189.3 | 1.21 | -1.93 | 317.1 | 0.11 | -2.62 | 0.73 | 0.91 | 1.00 |
| 5 | 330 | 85.40 | 184.7 | 1.21 | -1.65 | 366.0 | 0.08 | -1.90 | 0.74 | 0.90 | 1.00 |
| 5 | 350 | 85.44 | 192.6 | 1.29 | -1.96 | 312.9 | 0.08 | -2.49 | 0.76 | 0.92 | 1.00 |
| 15 | 10 | 86.42 | 203.8 | 0.78 | -2.24 | 366.0 | 0.15 | 0.76 | 0.55 | 0.91 | 1.00 |
| 15 | 30 | 86.36 | 203.3 | 0.79 | -2.03 | 366.0 | 0.18 | 0.60 | 0.51 | 0.87 | 1.00 |
| 15 | 50 | 86.13 | 200.5 | 0.89 | -1.81 | 366.0 | 0.31 | 0.21 | 0.50 | 0.82 | 0.99 |
| 15 | 70 | 86.06 | 194.0 | 0.77 | -1.57 | 366.0 | 0.34 | -0.15 | 0.47 | 0.85 | 0.99 |
| 15 | 90 | 86.37 | 184.7 | 0.77 | -1.26 | 366.0 | 0.20 | -0.44 | 0.52 | 0.89 | 1.00 |
| 15 | 110 | 86.66 | 180.0 | 0.67 | -0.77 | 286.8 | 0.15 | 1.32 | 0.39 | 0.86 | 1.00 |
| 15 | 130 | 86.67 | 184.2 | 0.51 | -0.80 | 330.6 | 0.15 | 1.00 | 0.28 | 0.88 | 1.00 |
| 15 | 150 | 86.45 | 180.0 | 0.67 | -1.21 | 366.0 | 0.28 | -0.71 | 0.54 | 0.92 | 1.00 |
| 15 | 170 | 86.64 | 180.0 | 0.76 | -1.17 | 366.0 | 0.39 | -1.52 | 0.62 | 0.94 | 1.00 |
| 15 | 190 | 86.91 | 180.0 | 0.56 | -1.14 | 366.0 | 0.31 | -1.74 | 0.46 | 0.91 | 1.00 |
| 15 | 210 | 86.86 | 180.0 | 0.58 | -0.99 | 267.2 | 0.19 | 0.99 | 0.40 | 0.92 | 1.00 |
| 15 | 230 | 86.55 | 186.5 | 0.79 | -1.55 | 314.3 | 0.19 | 1.10 | 0.60 | 0.93 | 1.00 |
| 15 | 250 | 86.41 | 185.6 | 0.90 | -1.69 | 366.0 | 0.28 | 0.27 | 0.68 | 0.94 | 1.00 |

| | | | | | | | | | | | |
|---|---|---|---|---|---|---|---|---|---|---|---|
| | | | 9 | | | 0 | | | | | |
| 15 | 270 | 86.53 | 180.0 | 0.72 | -1.45 | 361.3 | 0.24 | -0.45 | 0.61 | 0.95 | 1.00 |
| 15 | 290 | 86.53 | 180.0 | 0.82 | -1.16 | 249.9 | 0.13 | 0.98 | 0.58 | 0.95 | 1.00 |
| 15 | 310 | 86.32 | 181.4 | 0.75 | -1.24 | 257.9 | 0.06 | 1.02 | 0.52 | 0.93 | 1.00 |
| 15 | 330 | 86.16 | 184.7 | 0.82 | -1.50 | 366.0 | 0.16 | -0.66 | 0.56 | 0.91 | 1.00 |
| 15 | 350 | 86.34 | 192.1 | 0.78 | -1.82 | 332.4 | 0.12 | 0.55 | 0.62 | 0.95 | 1.00 |
| 25 | 10 | 86.83 | 180.0 | 0.45 | -0.36 | 366.0 | 0.30 | 1.26 | 0.31 | 0.86 | 1.00 |
| 25 | 30 | 86.79 | 180.0 | 0.51 | -0.20 | 326.4 | 0.20 | 1.08 | 0.29 | 0.86 | 1.00 |
| 25 | 50 | 86.70 | 180.0 | 0.48 | -0.19 | 358.1 | 0.17 | 0.70 | 0.29 | 0.89 | 1.00 |
| 25 | 70 | 86.60 | 180.0 | 0.43 | -0.28 | 366.0 | 0.23 | -0.04 | 0.35 | 0.95 | 1.00 |
| 25 | 90 | 86.75 | 180.0 | 0.52 | -0.23 | 336.6 | 0.16 | 0.13 | 0.35 | 0.91 | 1.00 |
| 25 | 110 | 86.84 | 180.0 | 0.67 | -0.09 | 259.7 | 0.26 | 0.40 | 0.39 | 0.89 | 1.00 |
| 25 | 130 | 86.81 | 180.0 | 0.52 | -0.02 | 283.0 | 0.28 | -0.28 | 0.35 | 0.89 | 1.00 |
| 25 | 150 | 86.63 | 366.0 | 0.48 | -1.27 | 180.0 | 0.44 | -0.51 | 0.49 | 0.92 | 1.00 |
| 25 | 170 | 86.71 | 180.0 | 0.56 | -0.59 | 356.7 | 0.39 | -1.21 | 0.52 | 0.94 | 1.00 |
| 25 | 190 | 86.96 | 180.0 | 0.61 | -0.49 | 298.4 | 0.22 | -0.46 | 0.42 | 0.91 | 1.00 |
| 25 | 210 | 87.04 | 180.0 | 0.57 | -0.31 | 297.9 | 0.10 | 1.10 | 0.40 | 0.93 | 1.00 |
| 25 | 230 | 86.95 | 180.0 | 0.60 | -0.58 | 355.7 | 0.35 | 1.17 | 0.56 | 0.95 | 1.00 |
| 25 | 250 | 86.83 | 189.3 | 0.56 | -1.05 | 366.0 | 0.37 | 0.60 | 0.54 | 0.95 | 1.00 |
| 25 | 270 | 86.79 | 180.0 | 0.52 | -0.35 | 366.0 | 0.18 | 0.12 | 0.42 | 0.93 | 1.00 |
| 25 | 290 | 86.76 | 180.0 | 0.63 | -0.25 | 339.0 | 0.13 | -0.94 | 0.44 | 0.91 | 1.00 |
| 25 | 310 | 86.71 | 180.0 | 0.52 | -0.18 | 358.5 | 0.20 | -1.49 | 0.43 | 0.96 | 1.00 |
| 25 | 330 | 86.68 | 180.0 | 0.46 | -0.43 | 366.0 | 0.14 | -0.94 | 0.35 | 0.94 | 1.00 |
| 25 | 350 | 86.77 | 180.0 | 0.51 | -0.57 | 352.0 | 0.25 | 1.25 | 0.34 | 0.89 | 1.00 |
| 35 | 10 | 86.58 | 180.0 | 0.68 | 0.57 | 366.0 | 0.63 | 1.32 | 0.65 | 0.93 | 1.00 |

| | | | | | | | | | | | |
|---|---|---|---|---|---|---|---|---|---|---|---|
| | | | 0 | | | 0 | | | | | |
| 35 | 30 | 86.56 | 180.0 | 0.72 | 0.69 | 366.0 | 0.48 | 1.29 | 0.63 | 0.94 | 1.00 |
| 35 | 50 | 86.54 | 180.0 | 0.66 | 0.62 | 366.0 | 0.40 | 1.23 | 0.55 | 0.93 | 1.00 |
| 35 | 70 | 86.44 | 180.0 | 0.63 | 0.58 | 366.0 | 0.31 | 0.98 | 0.50 | 0.92 | 1.00 |
| 35 | 90 | 86.49 | 180.0 | 0.73 | 0.57 | 366.0 | 0.29 | 1.17 | 0.55 | 0.91 | 1.00 |
| 35 | 110 | 86.52 | 180.0 | 0.77 | 0.63 | 366.0 | 0.13 | 1.16 | 0.55 | 0.91 | 1.00 |
| 35 | 130 | 86.57 | 180.0 | 0.71 | 0.59 | 291.4 | 0.22 | 0.39 | 0.50 | 0.94 | 1.00 |
| 35 | 150 | 86.50 | 180.0 | 0.59 | 0.47 | 317.1 | 0.27 | -0.18 | 0.41 | 0.93 | 1.00 |
| 35 | 170 | 86.58 | 180.0 | 0.62 | 0.37 | 312.9 | 0.23 | 0.37 | 0.41 | 0.92 | 1.00 |
| 35 | 190 | 86.72 | 180.0 | 0.56 | 0.39 | 335.7 | 0.21 | 1.14 | 0.37 | 0.90 | 1.00 |
| 35 | 210 | 86.83 | 180.0 | 0.60 | 0.38 | 366.0 | 0.39 | 1.14 | 0.50 | 0.92 | 1.00 |
| 35 | 230 | 86.82 | 180.0 | 0.60 | 0.27 | 366.0 | 0.51 | 1.17 | 0.58 | 0.95 | 1.00 |
| 35 | 250 | 86.72 | 366.0 | 0.55 | 1.04 | 180.0 | 0.56 | 0.14 | 0.62 | 0.94 | 1.00 |
| 35 | 270 | 86.63 | 180.0 | 0.58 | 0.23 | 366.0 | 0.40 | 0.89 | 0.51 | 0.94 | 1.00 |
| 35 | 290 | 86.52 | 180.0 | 0.72 | 0.49 | 366.0 | 0.21 | 0.39 | 0.54 | 0.92 | 1.00 |
| 35 | 310 | 86.45 | 180.0 | 0.73 | 0.55 | 366.0 | 0.25 | 0.67 | 0.60 | 0.94 | 1.00 |
| 35 | 330 | 86.53 | 180.0 | 0.64 | 0.54 | 366.0 | 0.37 | 1.03 | 0.59 | 0.95 | 1.00 |
| 35 | 350 | 86.64 | 180.0 | 0.62 | 0.46 | 366.0 | 0.59 | 1.31 | 0.62 | 0.93 | 1.00 |
| 45 | 10 | 86.34 | 366.0 | 0.97 | 1.35 | 180.0 | 0.72 | 0.80 | 0.78 | 0.94 | 1.00 |
| 45 | 30 | 86.36 | 366.0 | 0.86 | 1.33 | 180.9 | 0.72 | 0.77 | 0.78 | 0.96 | 1.00 |
| 45 | 50 | 86.37 | 366.0 | 0.79 | 1.34 | 182.3 | 0.66 | 0.82 | 0.74 | 0.96 | 1.00 |
| 45 | 70 | 86.35 | 180.0 | 0.67 | 0.91 | 365.1 | 0.65 | 1.35 | 0.74 | 0.96 | 1.00 |
| 45 | 90 | 86.35 | 180.0 | 0.69 | 0.79 | 366.0 | 0.65 | 1.40 | 0.72 | 0.96 | 1.00 |
| 45 | 110 | 86.30 | 180.0 | 0.71 | 0.86 | 364.1 | 0.61 | 1.44 | 0.67 | 0.94 | 1.00 |
| 45 | 130 | 86.29 | 180.0 | 0.59 | 0.89 | 361.3 | 0.55 | 1.27 | 0.63 | 0.96 | 1.00 |

| | | | | | | | | | | | |
|---|---|---|---|---|---|---|---|---|---|---|---|
| | | | θ | | | 4 | | | | | |
| 45 | 150 | 86.25 | 366.0 | 0.55 | 1.21 | 180.0 | 0.53 | 1.00 | 0.57 | 0.93 | 1.00 |
| 45 | 170 | 86.24 | 366.0 | 0.65 | 1.31 | 180.0 | 0.59 | 0.97 | 0.65 | 0.91 | 1.00 |
| 45 | 190 | 86.27 | 366.0 | 0.88 | 1.37 | 180.0 | 0.67 | 0.99 | 0.73 | 0.90 | 1.00 |
| 45 | 210 | 86.33 | 366.0 | 1.08 | 1.39 | 180.5 | 0.71 | 0.88 | 0.78 | 0.92 | 1.00 |
| 45 | 230 | 86.35 | 366.0 | 1.12 | 1.38 | 181.9 | 0.72 | 0.82 | 0.80 | 0.92 | 1.00 |
| 45 | 250 | 86.31 | 366.0 | 0.96 | 1.30 | 180.5 | 0.69 | 0.81 | 0.80 | 0.94 | 1.00 |
| 45 | 270 | 86.27 | 366.0 | 0.73 | 1.21 | 180.0 | 0.70 | 0.89 | 0.74 | 0.96 | 1.00 |
| 45 | 290 | 86.24 | 180.0 | 0.74 | 0.85 | 358.1 | 0.58 | 1.15 | 0.70 | 0.95 | 1.00 |
| 45 | 310 | 86.14 | 180.0 | 0.69 | 0.88 | 354.4 | 0.65 | 1.21 | 0.66 | 0.95 | 1.00 |
| 45 | 330 | 86.24 | 366.0 | 0.83 | 1.23 | 180.0 | 0.63 | 0.78 | 0.75 | 0.95 | 1.00 |
| 45 | 350 | 86.35 | 366.0 | 0.90 | 1.32 | 180.0 | 0.69 | 0.73 | 0.80 | 0.96 | 1.00 |
| 51 | 10 | 86.38 | 366.0 | 1.20 | 1.34 | 180.0 | 0.48 | 0.98 | 0.76 | 0.91 | 1.00 |
| 51 | 30 | 86.40 | 366.0 | 1.11 | 1.33 | 180.0 | 0.46 | 0.96 | 0.73 | 0.89 | 1.00 |
| 51 | 50 | 86.44 | 366.0 | 1.00 | 1.34 | 182.8 | 0.44 | 0.99 | 0.72 | 0.91 | 1.00 |
| 51 | 70 | 86.43 | 366.0 | 0.88 | 1.34 | 181.4 | 0.45 | 1.08 | 0.69 | 0.93 | 0.99 |
| 51 | 90 | 86.41 | 366.0 | 0.92 | 1.40 | 180.0 | 0.40 | 0.98 | 0.69 | 0.92 | 0.99 |
| 51 | 110 | 86.28 | 366.0 | 0.91 | 1.35 | 180.0 | 0.42 | 1.06 | 0.68 | 0.93 | 1.00 |
| 51 | 130 | 86.18 | 366.0 | 0.96 | 1.40 | 180.0 | 0.36 | 1.25 | 0.68 | 0.89 | 1.00 |
| 51 | 150 | 86.10 | 366.0 | 1.08 | 1.39 | 180.0 | 0.35 | 1.27 | 0.70 | 0.89 | 0.99 |
| 51 | 170 | 86.04 | 366.0 | 1.31 | 1.43 | 180.0 | 0.36 | 1.32 | 0.75 | 0.88 | 0.99 |
| 51 | 190 | 86.02 | 366.0 | 1.53 | 1.39 | 180.0 | 0.46 | 1.34 | 0.80 | 0.85 | 0.99 |
| 51 | 210 | 85.97 | 366.0 | 1.74 | 1.43 | 180.0 | 0.50 | 1.27 | 0.83 | 0.88 | 1.00 |
| 51 | 230 | 85.99 | 366.0 | 1.70 | 1.42 | 180.0 | 0.55 | 1.11 | 0.84 | 0.87 | 1.00 |
| 51 | 250 | 86.04 | 366.0 | 1.45 | 1.43 | 180.0 | 0.53 | 1.25 | 0.81 | 0.89 | 0.99 |

| | | | | | | | | | | | |
|---|---|---|---|---|---|---|---|---|---|---|---|
| | | | 0 | | | 0 | | | | | |
| 51 | 270 | 86.08 | 366.0 | 1.21 | 1.38 | 180.0 | 0.54 | 1.30 | 0.73 | 0.88 | 0.99 |
| 51 | 290 | 86.15 | 366.0 | 1.10 | 1.26 | 180.0 | 0.47 | 1.33 | 0.70 | 0.89 | 0.99 |
| 51 | 310 | 86.09 | 366.0 | 1.15 | 1.26 | 180.0 | 0.36 | 1.16 | 0.70 | 0.89 | 0.99 |
| 51 | 330 | 86.16 | 366.0 | 1.18 | 1.28 | 180.0 | 0.35 | 1.03 | 0.74 | 0.89 | 0.99 |
| 51 | 350 | 86.27 | 366.0 | 1.15 | 1.32 | 180.0 | 0.45 | 0.83 | 0.74 | 0.90 | 1.00 |

Table 34 Same as table 1 but for the FWHM. Please pay attention to the fact that FWHM is more variable during the year and therefore the fraction of data around the approximation is provided for intervals of +/- 7.5 % and +/- 15 %.

| Lat. | Lon. | Mean | 1st oscillation | | | 2nd oscillation | | | Quality of | Fraction of data | |
| --- | --- | --- | --- | --- | --- | --- | --- | --- | --- | --- | --- |
| [°] | [°] | [$10^{-2}$ s$^{-1}$] | Period [d] | Amp. [$10^{-2}$ s$^{-1}$] | Phase [rad] | Period [d] | Amp. [$10^{-2}$ s$^{-1}$] | Phase [rad] | approxi-mation | +-7.5% | +/- 15% |
| | | | | | | | | | | around approx. | |
| -55 -51 | 10 | 7.50 | 366.0 366.00 | 0.74 | 1.78 | 191.7 191.65 | 0.60 | 1.22 | 0.51 | 0.68 | 0.97 |
| -55 -51 | 30 | 7.48 | 366.0 366.00 | 0.82 | 1.63 | 191.7 191.65 | 0.60 | 1.23 | 0.60 | 0.71 | 0.96 |
| -55 -51 | 50 | 7.54 | 366.0 366.00 | 0.68 | 1.58 | 189.3 189.32 | 0.47 | 1.37 | 0.48 | 0.70 | 0.94 |
| -55 -51 | 70 | 7.57 | 366.0 366.00 | 0.67 | 1.61 | 194.0 193.98 | 0.38 | 1.20 | 0.32 | 0.68 | 0.96 |
| -55 -51 | 90 | 7.53 | 366.0 366.00 | 0.67 | 1.69 | 194.5 194.45 | 0.36 | 1.25 | 0.36 | 0.63 | 0.95 |
| -55 -51 | 110 | 7.51 | 366.0 366.00 | 0.79 | 1.67 | 193.5 193.52 | 0.48 | 1.23 | 0.55 | 0.69 | 0.96 |
| -55 -51 | 130 | 7.58 | 366.0 366.00 | 0.77 | 1.69 | 194.0 193.98 | 0.45 | 1.13 | 0.49 | 0.66 | 0.95 |
| -55 -51 | 150 | 7.65 | 366.0 366.00 | 0.72 | 1.60 | 193.5 193.52 | 0.46 | 1.16 | 0.46 | 0.67 | 0.94 |
| -55 -51 | 170 | 7.69 | 366.0 366.00 | 0.75 | 1.61 | 191.7 191.65 | 0.51 | 1.15 | 0.53 | 0.68 | 0.98 |
| -55 -51 | 190 | 7.66 | 366.0 366.00 | 0.78 | 1.58 | 191.2 191.19 | 0.60 | 1.23 | 0.58 | 0.72 | 0.97 |
| -55 -51 | 210 | 7.68 | 366.0 366.00 | 0.85 | 1.63 | 190.7 190.72 | 0.61 | 1.08 | 0.62 | 0.73 | 0.97 |
| -55 -51 | 230 | 7.76 | 366.0 366.00 | 0.80 | 1.68 | 189.3 189.32 | 0.48 | 1.22 | 0.65 | 0.80 | 0.98 |
| -55 -51 | 250 | 7.71 | 366.0 366.00 | 0.94 | 1.73 | 189.8 189.79 | 0.49 | 1.37 | 0.66 | 0.76 | 0.96 |
| -55 -51 | 270 | 7.70 | 366.0 366.00 | 0.97 | 1.73 | 188.9 188.86 | 0.53 | 1.19 | 0.64 | 0.67 | 0.95 |
| -55 -51 | 290 | 7.75 | 366.0 366.00 | 0.95 | 1.70 | 188.4 188.39 | 0.53 | 1.03 | 0.66 | 0.71 | 0.96 |
| -55 -51 | 310 | 7.66 | 366.0 366.00 | 0.87 | 1.67 | 193.5 193.52 | 0.59 | 0.73 | 0.65 | 0.77 | 0.96 |
| -55 -51 | 330 | 7.59 | 366.0 366.00 | 0.76 | 1.72 | 188.9 188.86 | 0.51 | 0.96 | 0.61 | 0.76 | 0.98 |
| -55 -51 | 350 | 7.57 | 366.0 366.00 | 0.68 | 1.93 | 189.3 189.32 | 0.40 | 1.26 | 0.46 | 0.69 | 0.94 |
| -45 | 10 | 7.55 | 366.0 366.00 | 0.43 | 1.69 | 190.7 190.72 | 0.33 | 0.84 | 0.28 | 0.64 | 0.95 |
| -45 | 30 | 7.57 | 366.0 366.00 | 0.51 | 1.55 | 193.5 193.52 | 0.33 | 1.05 | 0.36 | 0.72 | 0.98 |

| | | | | | | | | | | | |
|---|---|---|---|---|---|---|---|---|---|---|---|
| -45 | 50 | 7.57 | 366.0 | 0.50 | 1.52 | 199.6 | 0.29 | 1.09 | 0.32 | 0.74 | 0.98 |
| -45 | 70 | 7.66 | 366.0 | 0.45 | 1.54 | 199.1 | 0.24 | 0.80 | 0.25 | 0.68 | 0.96 |
| -45 | 90 | 7.61 | 366.0 | 0.51 | 1.75 | 200.5 | 0.18 | 0.68 | 0.29 | 0.71 | 0.96 |
| -45 | 110 | 7.58 | 366.0 | 0.62 | 1.76 | 203.3 | 0.24 | 0.92 | 0.39 | 0.75 | 0.98 |
| -45 | 130 | 7.55 | 366.0 | 0.63 | 1.75 | 199.6 | 0.28 | 0.93 | 0.45 | 0.75 | 0.98 |
| -45 | 150 | 7.72 | 366.0 | 0.47 | 1.85 | 204.2 | 0.16 | 1.10 | 0.29 | 0.73 | 0.99 |
| -45 | 170 | 7.76 | 366.0 | 0.44 | 1.77 | 201.9 | 0.16 | 0.82 | 0.27 | 0.80 | 0.97 |
| -45 | 190 | 7.67 | 366.0 | 0.55 | 1.84 | 196.8 | 0.24 | 0.88 | 0.40 | 0.80 | 0.98 |
| -45 | 210 | 7.69 | 366.0 | 0.61 | 1.74 | 196.3 | 0.30 | 0.88 | 0.47 | 0.79 | 0.99 |
| -45 | 230 | 7.71 | 366.0 | 0.52 | 1.79 | 198.7 | 0.22 | 0.67 | 0.37 | 0.80 | 0.98 |
| -45 | 250 | 7.66 | 366.0 | 0.58 | 1.92 | 199.1 | 0.22 | 0.64 | 0.38 | 0.74 | 0.97 |
| -45 | 270 | 7.66 | 366.0 | 0.55 | 1.86 | 192.1 | 0.37 | 0.77 | 0.39 | 0.70 | 0.96 |
| -45 | 290 | 7.76 | 366.0 | 0.62 | 1.82 | 180.0 | 0.32 | 0.82 | 0.51 | 0.76 | 0.99 |
| -45 | 310 | 7.76 | 366.0 | 0.46 | 1.69 | 187.5 | 0.34 | 0.86 | 0.42 | 0.79 | 0.98 |
| -45 | 330 | 7.65 | 366.0 | 0.45 | 1.69 | 188.9 | 0.36 | 0.60 | 0.40 | 0.77 | 0.99 |
| -45 | 350 | 7.58 | 366.0 | 0.41 | 1.79 | 193.5 | 0.26 | 0.70 | 0.26 | 0.72 | 0.96 |
| -35 | 10 | 7.39 | 366.0 | 0.25 | 2.07 | 180.0 | 0.21 | -0.38 | 0.10 | 0.60 | 0.90 |
| -35 | 30 | 7.41 | 366.0 | 0.28 | 1.90 | 180.0 | 0.21 | -0.94 | 0.13 | 0.63 | 0.92 |
| -35 | 50 | 7.43 | 366.0 | 0.27 | 2.05 | 180.0 | 0.26 | -1.33 | 0.15 | 0.69 | 0.90 |
| -35 | 70 | 7.47 | 180.0 | 0.24 | -1.09 | 366.0 | 0.20 | 1.97 | 0.10 | 0.68 | 0.91 |
| -35 | 90 | 7.48 | 366.0 | 0.21 | 1.84 | 180.0 | 0.15 | -0.67 | 0.07 | 0.62 | 0.92 |
| -35 | 110 | 7.40 | 366.0 | 0.31 | 1.72 | 180.0 | 0.12 | -0.76 | 0.13 | 0.66 | 0.93 |
| -35 | 130 | 7.33 | 366.0 | 0.32 | 1.83 | 180.0 | 0.16 | -0.17 | 0.16 | 0.69 | 0.94 |
| -35 | 150 | 7.42 | 331.0 | 0.27 | 2.54 | 180.0 | 0.17 | -1.31 | 0.14 | 0.69 | 0.95 |

| -35 | 170 | 7.54 | 180.0180.00 | 0.25 | -1.01 | 366.0366.00 | 0.21 | 2.79 | 0.15 | 0.74 | 0.94 |
|---|---|---|---|---|---|---|---|---|---|---|---|
| -35 | 190 | 7.44 | 366.0366.00 | 0.37 | 2.33 | 180.0180.00 | 0.17 | -1.05 | 0.24 | 0.76 | 0.96 |
| -35 | 210 | 7.30 | 366.0366.00 | 0.42 | 2.13 | 180.0180.00 | 0.03 | -0.63 | 0.29 | 0.78 | 0.97 |
| -35 | 230 | 7.36 | 351.1351.08 | 0.18 | 2.24 | 180.0180.00 | 0.10 | -1.13 | 0.08 | 0.76 | 0.97 |
| -35 | 250 | 7.38 | 322.2322.18 | 0.19 | -2.90 | 180.0180.00 | 0.15 | -0.71 | 0.09 | 0.73 | 0.95 |
| -35 | 270 | 7.49 | 180.0180.00 | 0.29 | 0.30 | 366.0366.00 | 0.24 | 3.06 | 0.16 | 0.66 | 0.93 |
| -35 | 290 | 7.69 | 359.9359.94 | 0.43 | 2.11 | 180.0180.00 | 0.27 | 0.06 | 0.33 | 0.74 | 0.96 |
| -35 | 310 | 7.73 | 366.0366.00 | 0.32 | 1.85 | 180.0180.00 | 0.17 | -0.31 | 0.18 | 0.72 | 0.95 |
| -35 | 330 | 7.67 | 180.0180.00 | 0.29 | -0.60 | 366.0366.00 | 0.15 | 1.96 | 0.12 | 0.69 | 0.93 |
| -35 | 350 | 7.45 | 180.0180.00 | 0.26 | -0.23 | 366.0366.00 | 0.15 | 1.37 | 0.08 | 0.60 | 0.89 |
| -25 | 10 | 7.48 | 180.0180.00 | 0.15 | 0.02 | 366.0366.00 | 0.11 | 2.18 | 0.05 | 0.68 | 0.96 |
| -25 | 30 | 7.53 | 366.0366.00 | 0.24 | 2.21 | 180.0180.00 | 0.11 | -1.54 | 0.11 | 0.71 | 0.96 |
| -25 | 50 | 7.48 | 366.0366.00 | 0.19 | 2.41 | 180.0180.00 | 0.13 | -1.55 | 0.07 | 0.67 | 0.97 |
| -25 | 70 | 7.49 | 180.0180.00 | 0.18 | -1.13 | 366.0366.00 | 0.15 | 2.38 | 0.07 | 0.66 | 0.94 |
| -25 | 90 | 7.59 | 366.0366.00 | 0.15 | 1.61 | 180.0180.00 | 0.10 | -1.37 | 0.04 | 0.65 | 0.95 |
| -25 | 110 | 7.50 | 363.7363.67 | 0.25 | 1.43 | 239.7239.67 | 0.10 | 0.96 | 0.09 | 0.69 | 0.96 |
| -25 | 130 | 7.34 | 365.5365.53 | 0.35 | 1.49 | 180.0180.00 | 0.12 | -1.49 | 0.17 | 0.61 | 0.95 |
| -25 | 150 | 7.46 | 180.0180.00 | 0.16 | -1.25 | 366.0366.00 | 0.14 | 1.70 | 0.05 | 0.65 | 0.93 |
| -25 | 170 | 7.55 | 180.0180.00 | 0.17 | -0.51 | 366.0366.00 | 0.13 | -2.77 | 0.07 | 0.71 | 0.96 |
| -25 | 190 | 7.47 | 366.0366.00 | 0.22 | 2.75 | 180.0180.00 | 0.11 | 0.18 | 0.12 | 0.78 | 0.98 |
| -25 | 210 | 7.20 | 366.0366.00 | 0.26 | 2.27 | 180.0180.00 | 0.09 | 0.73 | 0.18 | 0.81 | 0.98 |
| -25 | 230 | 7.17 | 238.7238.74 | 0.12 | -1.71 | 180.0180.00 | 0.07 | 0.88 | 0.05 | 0.72 | 0.97 |
| -25 | 250 | 7.33 | 263.9263.91 | 0.21 | -0.48 | 180.0180.00 | 0.20 | -0.30 | 0.13 | 0.72 | 0.96 |
| -25 | 270 | 7.51 | 219.6219.62 | 0.27 | 0.05 | 366.0366.00 | 0.12 | -3.00 | 0.14 | 0.68 | 0.96 |

| | | | | | | | | | | | |
|---|---|---|---|---|---|---|---|---|---|---|---|
| -25 | 290 | 7.69 | 359.9 | 0.21 | 2.87 | 180.0 | 0.17 | 0.58 | 0.13 | 0.75 | 0.98 |
| -25 | 310 | 7.68 | 366.0 | 0.30 | 2.34 | 236.4 | 0.09 | 0.81 | 0.12 | 0.75 | 0.98 |
| -25 | 330 | 7.59 | 180.0 | 0.25 | -0.52 | 225.2 | 0.19 | 1.56 | 0.04 | 0.69 | 0.95 |
| -25 | 350 | 7.47 | 180.0 | 0.37 | -0.14 | 244.8 | 0.27 | 1.19 | 0.10 | 0.67 | 0.95 |
| -15 | 10 | 7.73 | 180.0 | 0.19 | 1.94 | 365.5 | 0.18 | 1.34 | 0.18 | 0.85 | 1.00 |
| -15 | 30 | 7.75 | 366.0 | 0.21 | 1.69 | 186.5 | 0.20 | 1.85 | 0.18 | 0.84 | 0.99 |
| -15 | 50 | 7.74 | 180.0 | 0.13 | 2.50 | 346.9 | 0.12 | 2.16 | 0.07 | 0.79 | 0.99 |
| -15 | 70 | 7.75 | 366.0 | 0.13 | 1.82 | 180.0 | 0.13 | 2.54 | 0.07 | 0.75 | 0.98 |
| -15 | 90 | 7.94 | 180.0 | 0.20 | 1.99 | 366.0 | 0.14 | 1.56 | 0.11 | 0.76 | 0.98 |
| -15 | 110 | 7.88 | 180.0 | 0.23 | 2.16 | 361.3 | 0.21 | 1.43 | 0.19 | 0.80 | 0.98 |
| -15 | 130 | 7.79 | 366.0 | 0.38 | 1.33 | 197.3 | 0.26 | 1.96 | 0.28 | 0.75 | 1.00 |
| -15 | 150 | 7.84 | 292.4 | 0.26 | 1.13 | 180.0 | 0.13 | 2.85 | 0.16 | 0.78 | 0.98 |
| -15 | 170 | 8.02 | 334.8 | 0.18 | 0.36 | 180.0 | 0.05 | 2.75 | 0.08 | 0.79 | 1.00 |
| -15 | 190 | 7.92 | 366.0 | 0.19 | 1.63 | 198.7 | 0.17 | 2.12 | 0.15 | 0.84 | 0.99 |
| -15 | 210 | 7.65 | 366.0 | 0.38 | 1.63 | 204.7 | 0.20 | 1.70 | 0.31 | 0.83 | 1.00 |
| -15 | 230 | 7.62 | 366.0 | 0.18 | 1.13 | 184.7 | 0.14 | 1.27 | 0.12 | 0.83 | 0.99 |
| -15 | 250 | 7.75 | 305.9 | 0.14 | -0.82 | 180.0 | 0.03 | 1.34 | 0.07 | 0.83 | 1.00 |
| -15 | 270 | 7.77 | 366.0 | 0.17 | -2.64 | 186.5 | 0.15 | 2.69 | 0.14 | 0.86 | 1.00 |
| -15 | 290 | 7.85 | 366.0 | 0.22 | 2.42 | 180.9 | 0.16 | 2.68 | 0.18 | 0.84 | 1.00 |
| -15 | 310 | 7.81 | 181.9 | 0.25 | 2.88 | 366.0 | 0.22 | 1.78 | 0.25 | 0.83 | 0.99 |
| -15 | 330 | 7.72 | 274.2 | 0.21 | 1.78 | 180.9 | 0.11 | 2.45 | 0.18 | 0.90 | 1.00 |
| -15 | 350 | 7.71 | 356.7 | 0.23 | 0.67 | 180.0 | 0.16 | 1.92 | 0.18 | 0.84 | 1.00 |
| -5 | 10 | 7.73 | 280.2 | 0.11 | 1.90 | 180.0 | 0.11 | -2.84 | 0.06 | 0.72 | 0.99 |
| -5 | 30 | 7.74 | 180.0 | 0.14 | -2.86 | 254.6 | 0.09 | -2.60 | 0.04 | 0.78 | 0.98 |

| | | | | | | | | | | | |
|---|---|---|---|---|---|---|---|---|---|---|---|
| -5 | 50 | 7.73 | 366.0 | 0.10 | -1.62 | 180.0 | 0.06 | -3.00 | 0.02 | 0.73 | 0.98 |
| -5 | 70 | 7.72 | 303.1 | 0.26 | -1.50 | 180.0 | 0.17 | 2.63 | 0.16 | 0.78 | 0.98 |
| -5 | 90 | 7.82 | 283.5 | 0.37 | -1.40 | 180.0 | 0.19 | 2.41 | 0.23 | 0.75 | 0.97 |
| -5 | 110 | 7.81 | 238.7 | 0.40 | -0.49 | 180.0 | 0.25 | 2.87 | 0.24 | 0.75 | 0.97 |
| -5 | 130 | 7.91 | 211.7 | 0.39 | 0.94 | 366.0 | 0.19 | -2.81 | 0.28 | 0.77 | 0.99 |
| -5 | 150 | 8.05 | 228.0 | 0.43 | 0.19 | 180.0 | 0.32 | -2.90 | 0.32 | 0.79 | 0.98 |
| -5 | 170 | 8.09 | 249.5 | 0.28 | 0.26 | 180.0 | 0.26 | -2.59 | 0.23 | 0.79 | 1.00 |
| -5 | 190 | 8.13 | 180.0 | 0.35 | 3.14 | 366.0 | 0.08 | 0.76 | 0.21 | 0.81 | 0.99 |
| -5 | 210 | 8.09 | 200.1 | 0.39 | 2.56 | 366.0 | 0.24 | 1.08 | 0.29 | 0.74 | 0.98 |
| -5 | 230 | 8.03 | 180.0 | 0.37 | -3.07 | 366.0 | 0.19 | 0.72 | 0.24 | 0.75 | 0.99 |
| -5 | 250 | 7.90 | 180.0 | 0.46 | -2.68 | 366.0 | 0.19 | -1.23 | 0.32 | 0.77 | 0.97 |
| -5 | 270 | 7.70 | 180.0 | 0.46 | -2.77 | 366.0 | 0.13 | -1.64 | 0.32 | 0.77 | 0.98 |
| -5 | 290 | 7.70 | 180.0 | 0.34 | -2.76 | 304.0 | 0.14 | -2.01 | 0.18 | 0.74 | 0.98 |
| -5 | 310 | 7.70 | 184.7 | 0.29 | -2.80 | 289.6 | 0.04 | 2.71 | 0.17 | 0.80 | 0.98 |
| -5 | 330 | 7.73 | 180.0 | 0.23 | -2.48 | 366.0 | 0.15 | 0.77 | 0.13 | 0.75 | 0.99 |
| -5 | 350 | 7.74 | 276.5 | 0.14 | 1.44 | 180.0 | 0.13 | -2.76 | 0.08 | 0.72 | 0.98 |
| 5 | 10 | 7.85 | 366.0 | 0.30 | -0.94 | 180.0 | 0.28 | -2.58 | 0.22 | 0.75 | 0.98 |
| 5 | 30 | 7.85 | 180.0 | 0.25 | -2.60 | 366.0 | 0.24 | -1.22 | 0.18 | 0.72 | 0.97 |
| 5 | 50 | 7.77 | 366.0 | 0.35 | -1.38 | 180.0 | 0.16 | -3.02 | 0.18 | 0.69 | 0.96 |
| 5 | 70 | 7.74 | 325.4 | 0.49 | -1.36 | 180.0 | 0.16 | 2.54 | 0.29 | 0.68 | 0.95 |
| 5 | 90 | 7.77 | 329.6 | 0.60 | -1.38 | 180.0 | 0.23 | 2.53 | 0.41 | 0.67 | 0.97 |
| 5 | 110 | 7.86 | 332.4 | 0.68 | -1.57 | 180.0 | 0.24 | 2.57 | 0.52 | 0.77 | 0.99 |
| 5 | 130 | 8.00 | 338.5 | 0.58 | -1.76 | 180.0 | 0.24 | 2.55 | 0.54 | 0.85 | 1.00 |
| 5 | 150 | 8.02 | 333.8 | 0.60 | -1.60 | 180.0 | 0.34 | 2.82 | 0.52 | 0.81 | 0.99 |

| | | | | | | | | | | | |
|---|---|---|---|---|---|---|---|---|---|---|---|
| 5 | 170 | 7.89 | 364.1  | 0.51 | -1.63 | 180.0  | 0.28 | -2.91 | 0.48 | 0.83 | 1.00 |
| 5 | 190 | 7.99 | 348.8  | 0.46 | -1.70 | 180.0  | 0.19 | 2.88 | 0.39 | 0.83 | 1.00 |
| 5 | 210 | 8.19 | 245.3  | 0.34 | -0.21 | 180.0  | 0.30 | -2.90 | 0.30 | 0.85 | 1.00 |
| 5 | 230 | 8.12 | 180.0  | 0.33 | -3.10 | 366.0  | 0.27 | -1.37 | 0.27 | 0.80 | 0.99 |
| 5 | 250 | 7.93 | 180.0  | 0.42 | -2.81 | 366.0  | 0.27 | -1.32 | 0.35 | 0.78 | 0.98 |
| 5 | 270 | 7.79 | 180.0  | 0.44 | -2.89 | 366.0  | 0.29 | -1.52 | 0.38 | 0.78 | 0.99 |
| 5 | 290 | 7.77 | 366.0  | 0.39 | -1.64 | 180.0  | 0.34 | -3.04 | 0.39 | 0.78 | 0.99 |
| 5 | 310 | 7.70 | 366.0  | 0.20 | -1.84 | 180.0  | 0.20 | -2.76 | 0.15 | 0.79 | 0.98 |
| 5 | 330 | 7.74 | 180.0  | 0.23 | -2.73 | 366.0  | 0.12 | -1.03 | 0.11 | 0.72 | 0.98 |
| 5 | 350 | 7.77 | 366.0  | 0.23 | -1.08 | 180.0  | 0.20 | -2.66 | 0.16 | 0.74 | 0.98 |
| 15 | 10 | 7.86 | 351.1  | 0.36 | -1.36 | 180.0  | 0.08 | 1.70 | 0.24 | 0.79 | 0.98 |
| 15 | 30 | 7.94 | 359.0  | 0.43 | -1.45 | 184.7  | 0.12 | 3.01 | 0.33 | 0.77 | 0.99 |
| 15 | 50 | 7.90 | 352.0  | 0.43 | -1.47 | 180.5  | 0.18 | 2.86 | 0.38 | 0.84 | 1.00 |
| 15 | 70 | 7.82 | 353.0  | 0.52 | -1.57 | 180.0  | 0.11 | 2.27 | 0.41 | 0.80 | 1.00 |
| 15 | 90 | 7.78 | 354.4  | 0.50 | -1.41 | 180.0  | 0.18 | 3.05 | 0.40 | 0.79 | 0.99 |
| 15 | 110 | 7.89 | 366.0  | 0.55 | -1.55 | 180.0  | 0.17 | -2.91 | 0.36 | 0.72 | 0.98 |
| 15 | 130 | 7.96 | 363.7  | 0.50 | -1.65 | 180.5  | 0.15 | 3.06 | 0.28 | 0.71 | 0.96 |
| 15 | 150 | 7.87 | 331.0  | 0.37 | -1.50 | 180.0  | 0.08 | 2.31 | 0.30 | 0.85 | 0.99 |
| 15 | 170 | 7.54 | 366.0  | 0.37 | -1.41 | 180.0  | 0.05 | -2.52 | 0.26 | 0.81 | 0.99 |
| 15 | 190 | 7.60 | 366.0  | 0.55 | -1.51 | 221.5  | 0.07 | -1.57 | 0.40 | 0.76 | 0.98 |
| 15 | 210 | 7.83 | 350.6  | 0.60 | -1.57 | 180.0  | 0.13 | 1.77 | 0.45 | 0.76 | 0.99 |
| 15 | 230 | 7.85 | 348.3  | 0.47 | -1.43 | 180.0  | 0.13 | 1.87 | 0.38 | 0.82 | 0.99 |
| 15 | 250 | 7.78 | 366.0  | 0.19 | -2.21 | 205.6  | 0.07 | -2.86 | 0.08 | 0.80 | 0.99 |
| 15 | 270 | 7.74 | 180.0  | 0.09 | 2.90 | 366.0  | 0.04 | -1.95 | 0.02 | 0.77 | 0.98 |

| | | | | | | | | | | | |
|---|---|---|---|---|---|---|---|---|---|---|---|
| 15 | 290 | 7.78 | 282.1 | 0.29 | -0.62 | 180.0 | 0.15 | 2.88 | 0.24 | 0.83 | 0.99 |
| 15 | 310 | 7.87 | 331.5 | 0.41 | -1.38 | 180.0 | 0.14 | 2.02 | 0.39 | 0.88 | 1.00 |
| 15 | 330 | 7.78 | 310.5 | 0.41 | -1.22 | 180.0 | 0.13 | 1.53 | 0.40 | 0.86 | 1.00 |
| 15 | 350 | 7.77 | 344.6 | 0.43 | -1.33 | 180.0 | 0.13 | 1.20 | 0.39 | 0.85 | 0.99 |
| 25 | 10 | 7.54 | 366.0 | 0.15 | -1.79 | 206.1 | 0.07 | -1.43 | 0.03 | 0.62 | 0.91 |
| 25 | 30 | 7.57 | 366.0 | 0.23 | -1.81 | 189.3 | 0.12 | -2.24 | 0.07 | 0.68 | 0.92 |
| 25 | 50 | 7.55 | 366.0 | 0.24 | -1.92 | 198.7 | 0.11 | -2.29 | 0.08 | 0.66 | 0.95 |
| 25 | 70 | 7.46 | 180.0 | 0.19 | -1.81 | 366.0 | 0.14 | -2.08 | 0.09 | 0.71 | 0.95 |
| 25 | 90 | 7.44 | 366.0 | 0.21 | -1.00 | 180.0 | 0.21 | -2.02 | 0.11 | 0.67 | 0.93 |
| 25 | 110 | 7.57 | 366.0 | 0.30 | -0.82 | 185.1 | 0.24 | -1.89 | 0.18 | 0.70 | 0.93 |
| 25 | 130 | 7.64 | 366.0 | 0.36 | -1.08 | 187.9 | 0.12 | -1.61 | 0.19 | 0.75 | 0.96 |
| 25 | 150 | 7.48 | 366.0 | 0.26 | -0.80 | 198.2 | 0.10 | -1.66 | 0.14 | 0.79 | 0.99 |
| 25 | 170 | 7.21 | 366.0 | 0.37 | -0.95 | 206.1 | 0.21 | -1.63 | 0.18 | 0.66 | 0.95 |
| 25 | 190 | 7.20 | 366.0 | 0.49 | -1.19 | 199.1 | 0.24 | -1.44 | 0.32 | 0.71 | 0.96 |
| 25 | 210 | 7.35 | 366.0 | 0.55 | -1.45 | 199.1 | 0.12 | -1.24 | 0.37 | 0.69 | 0.99 |
| 25 | 230 | 7.39 | 366.0 | 0.49 | -1.53 | 192.1 | 0.12 | -0.96 | 0.32 | 0.70 | 0.96 |
| 25 | 250 | 7.36 | 180.0 | 0.21 | -0.88 | 366.0 | 0.20 | -2.23 | 0.11 | 0.67 | 0.94 |
| 25 | 270 | 7.36 | 202.4 | 0.20 | -1.60 | 365.5 | 0.04 | -2.00 | 0.05 | 0.63 | 0.93 |
| 25 | 290 | 7.50 | 366.0 | 0.14 | -0.72 | 208.4 | 0.09 | -1.22 | 0.03 | 0.67 | 0.95 |
| 25 | 310 | 7.56 | 240.6 | 0.19 | -0.71 | 366.0 | 0.17 | -0.68 | 0.12 | 0.74 | 0.98 |
| 25 | 330 | 7.49 | 243.4 | 0.29 | -0.21 | 180.0 | 0.11 | 0.70 | 0.16 | 0.65 | 0.96 |
| 25 | 350 | 7.44 | 276.0 | 0.20 | -0.72 | 180.0 | 0.11 | 0.75 | 0.08 | 0.63 | 0.92 |
| 35 | 10 | 7.37 | 366.0 | 0.19 | -1.41 | 204.7 | 0.17 | -1.16 | 0.06 | 0.61 | 0.93 |
| 35 | 30 | 7.38 | 366.0 | 0.24 | -1.46 | 200.5 | 0.18 | -1.21 | 0.09 | 0.67 | 0.92 |

| | | | | | | | | | | | |
|---|---|---|---|---|---|---|---|---|---|---|---|
| 35 | 50 | 7.33 | 366.0  | 0.19 | -1.41 | 204.2  | 0.12 | -1.54 | 0.05 | 0.62 | 0.91 |
| 35 | 70 | 7.28 | 366.0  | 0.22 | -1.26 | 197.7  | 0.20 | -1.70 | 0.08 | 0.64 | 0.92 |
| 35 | 90 | 7.28 | 366.0  | 0.32 | -0.96 | 191.7  | 0.25 | -1.67 | 0.15 | 0.61 | 0.92 |
| 35 | 110 | 7.40 | 366.0  | 0.48 | -0.93 | 187.0  | 0.22 | -1.30 | 0.30 | 0.72 | 0.95 |
| 35 | 130 | 7.47 | 366.0  | 0.48 | -0.88 | 180.0  | 0.21 | -0.58 | 0.35 | 0.75 | 0.96 |
| 35 | 150 | 7.39 | 366.0  | 0.46 | -0.83 | 192.6  | 0.17 | -1.22 | 0.30 | 0.73 | 0.96 |
| 35 | 170 | 7.22 | 366.0  | 0.45 | -0.92 | 201.9  | 0.19 | -1.56 | 0.26 | 0.67 | 0.96 |
| 35 | 190 | 7.19 | 366.0  | 0.58 | -1.10 | 201.0  | 0.22 | -1.44 | 0.34 | 0.68 | 0.96 |
| 35 | 210 | 7.24 | 366.0  | 0.65 | -1.27 | 195.4  | 0.23 | -1.50 | 0.44 | 0.71 | 0.97 |
| 35 | 230 | 7.25 | 366.0  | 0.66 | -1.30 | 198.2  | 0.24 | -1.53 | 0.43 | 0.70 | 0.96 |
| 35 | 250 | 7.26 | 366.0  | 0.48 | -1.43 | 199.6  | 0.28 | -1.63 | 0.27 | 0.65 | 0.95 |
| 35 | 270 | 7.30 | 181.9  | 0.24 | -1.08 | 366.0  | 0.18 | -1.17 | 0.13 | 0.71 | 0.95 |
| 35 | 290 | 7.45 | 180.0  | 0.20 | -0.58 | 355.7  | 0.14 | -0.43 | 0.09 | 0.72 | 0.96 |
| 35 | 310 | 7.42 | 366.0  | 0.27 | -1.05 | 180.0  | 0.20 | 0.07 | 0.17 | 0.73 | 0.96 |
| 35 | 330 | 7.30 | 366.0  | 0.32 | -1.20 | 180.0  | 0.18 | 0.59 | 0.17 | 0.65 | 0.94 |
| 35 | 350 | 7.25 | 366.0  | 0.29 | -1.30 | 198.2  | 0.16 | -0.46 | 0.11 | 0.57 | 0.92 |
| 45 | 10 | 7.47 | 337.1  | 0.61 | -1.08 | 180.0  | 0.29 | 1.17 | 0.48 | 0.78 | 0.97 |
| 45 | 30 | 7.44 | 339.9  | 0.53 | -1.10 | 180.0  | 0.30 | 1.11 | 0.42 | 0.72 | 0.97 |
| 45 | 50 | 7.42 | 260.2  | 0.43 | -0.82 | 366.0  | 0.31 | -0.73 | 0.38 | 0.70 | 0.97 |
| 45 | 70 | 7.38 | 320.3  | 0.52 | -0.82 | 180.0  | 0.25 | 1.24 | 0.39 | 0.74 | 0.96 |
| 45 | 90 | 7.35 | 314.3  | 0.60 | -0.69 | 180.0  | 0.26 | 1.24 | 0.43 | 0.74 | 0.94 |
| 45 | 110 | 7.46 | 311.5  | 0.60 | -0.63 | 180.0  | 0.24 | 1.05 | 0.45 | 0.74 | 0.98 |
| 45 | 130 | 7.50 | 294.7  | 0.56 | -0.35 | 180.0  | 0.25 | 0.86 | 0.45 | 0.78 | 0.97 |
| 45 | 150 | 7.51 | 334.3  | 0.63 | -0.83 | 180.0  | 0.28 | 1.06 | 0.53 | 0.80 | 0.98 |

| | | | | | | | | | | | |
|---|---|---|---|---|---|---|---|---|---|---|---|
| 45 | 170 | 7.43 | 322.2  | 0.59 | -0.77 | 180.0  | 0.31 | 1.14 | 0.52 | 0.78 | 0.97 |
| 45 | 190 | 7.41 | 346.0  | 0.73 | -1.11 | 180.0  | 0.32 | 1.09 | 0.59 | 0.79 | 0.98 |
| 45 | 210 | 7.43 | 362.7  | 0.83 | -1.29 | 180.0  | 0.27 | 1.23 | 0.62 | 0.73 | 0.99 |
| 45 | 230 | 7.49 | 366.0  | 0.96 | -1.39 | 180.0  | 0.23 | 1.03 | 0.63 | 0.72 | 0.98 |
| 45 | 250 | 7.53 | 366.0  | 0.93 | -1.48 | 180.0  | 0.19 | 0.78 | 0.58 | 0.69 | 0.96 |
| 45 | 270 | 7.67 | 366.0  | 0.67 | -1.31 | 206.6  | 0.21 | -1.30 | 0.44 | 0.72 | 0.98 |
| 45 | 290 | 7.66 | 366.0  | 0.54 | -1.26 | 188.4  | 0.20 | -0.33 | 0.39 | 0.75 | 0.99 |
| 45 | 310 | 7.59 | 366.0  | 0.62 | -1.27 | 180.0  | 0.24 | 0.70 | 0.45 | 0.73 | 0.98 |
| 45 | 330 | 7.52 | 359.0  | 0.70 | -1.22 | 180.0  | 0.33 | 1.10 | 0.53 | 0.73 | 0.99 |
| 45 | 350 | 7.44 | 349.2  | 0.67 | -1.20 | 180.0  | 0.30 | 1.32 | 0.48 | 0.68 | 0.97 |
| 51 | 10 | 7.40 | 262.5  | 0.93 | -0.16 | 180.0  | 0.31 | 0.82 | 0.58 | 0.64 | 0.95 |
| 51 | 30 | 7.38 | 249.5  | 0.73 | -0.39 | 366.0  | 0.37 | -0.88 | 0.57 | 0.66 | 0.93 |
| 51 | 50 | 7.36 | 253.2  | 0.59 | -0.40 | 366.0  | 0.34 | -0.63 | 0.47 | 0.69 | 0.92 |
| 51 | 70 | 7.27 | 248.5  | 0.73 | 0.16 | 180.0  | 0.28 | 0.73 | 0.49 | 0.54 | 0.80 |
| 51 | 90 | 7.25 | 245.3  | 0.59 | -0.06 | 366.0  | 0.27 | -0.69 | 0.41 | 0.64 | 0.93 |
| 51 | 110 | 7.31 | 232.2  | 0.57 | 0.26 | 366.0  | 0.20 | -0.80 | 0.41 | 0.71 | 0.94 |
| 51 | 130 | 7.32 | 230.8  | 0.65 | 0.30 | 366.0  | 0.21 | -0.81 | 0.47 | 0.68 | 0.95 |
| 51 | 150 | 7.34 | 230.8  | 0.62 | 0.26 | 366.0  | 0.22 | -0.71 | 0.46 | 0.70 | 0.95 |
| 51 | 170 | 7.31 | 227.6  | 0.71 | 0.29 | 366.0  | 0.25 | -0.85 | 0.55 | 0.71 | 0.95 |
| 51 | 190 | 7.32 | 234.5  | 0.76 | 0.03 | 366.0  | 0.32 | -0.93 | 0.57 | 0.67 | 0.94 |
| 51 | 210 | 7.41 | 259.3  | 0.88 | -0.11 | 180.0  | 0.29 | 1.10 | 0.58 | 0.63 | 0.94 |
| 51 | 230 | 7.54 | 290.5  | 1.00 | -0.64 | 180.0  | 0.37 | 1.21 | 0.67 | 0.63 | 0.95 |
| 51 | 250 | 7.63 | 327.3  | 1.07 | -1.16 | 180.0  | 0.37 | 1.27 | 0.68 | 0.69 | 0.95 |
| 51 | 270 | 7.65 | 347.8  | 1.05 | -1.36 | 180.0  | 0.42 | 0.86 | 0.66 | 0.69 | 0.95 |

| | | | | | | | | | | | |
|---|---|---|---|---|---|---|---|---|---|---|---|
| 5551 | 290 | 7.69 | 347.4347.35 | 0.91 | -1.28 | 180.0180.00 | 0.38 | 0.93 | 0.61 | 0.73 | 0.95 |
| 5551 | 310 | 7.61 | 329.6329.64 | 0.95 | -1.00 | 180.0180.00 | 0.45 | 1.03 | 0.66 | 0.70 | 0.96 |
| 5551 | 330 | 7.60 | 317.5317.52 | 0.86 | -0.82 | 180.0180.00 | 0.43 | 1.12 | 0.60 | 0.64 | 0.95 |
| 5551 | 350 | 7.44 | 266.7266.71 | 0.97 | -0.24 | 180.0180.00 | 0.35 | 0.86 | 0.61 | 0.62 | 0.94 |